# From Complexity to Clarity: Analytical Expressions of Deep Neural Network Weights via Clifford Algebra and Convexity

**Mert Pilanci**                                                                *pilanci@stanford.edu*
*Department of Electrical Engineering*
*Stanford University*
*www.stanford.edu/∼pilanci*

**Reviewed on OpenReview:** *https: // openreview. net/ forum? id= 6XPwSwEWsV*

## Abstract

In this paper, we introduce a novel analysis of neural networks based on geometric (Clifford) algebra and convex optimization. We show that optimal weights of deep ReLU neural networks are given by the wedge product of training samples when trained with standard regularized loss. Furthermore, the training problem reduces to convex optimization over wedge product features, which encode the geometric structure of the training dataset. This structure is given in terms of signed volumes of triangles and parallelotopes generated by data vectors. The convex problem finds a small subset of samples via $\ell_1$ regularization to discover only relevant wedge product features. Our analysis provides a novel perspective on the inner workings of deep neural networks and sheds light on the role of the hidden layers.

## 1 Introduction

While there has been a lot of progress in developing deep neural networks (DNNs) to solve practical machine learning problems (Krizhevsky et al., 2012; LeCun et al., 2015; OpenAI, 2023), the inner workings of neural networks is not well understood. A foundational theory for understanding how neural networks work is still lacking despite extensive research over several decades. In this paper, we provide a novel analysis of neural networks based on geometric algebra and convex optimization. We show that weights of deep ReLU neural networks learn the wedge product of a subset of training samples when trained by minimizing standard regularized loss functions. Furthermore, the training problem for two-layer and three-layer networks reduces to convex optimization over wedge product features, which encode the geometric structure of the training dataset. This structure is given in terms of signed volumes of triangles and parallelotopes generated by data vectors. By the addition of an additional ReLU layer, the wedge products are iterated to yield a richer discrete dictionary of wedge features. Our analysis provides a novel perspective on the inner workings of deep neural networks and sheds light on the role of the hidden layers.

### 1.1 Prior work

The quest to understand the internal workings of neural networks (NNs) has led to numerous theoretical and empirical studies over the years. A striking discovery is the phenomenon of "neural collapse," observed when the representations of individual classes in the penultimate layer of a deep neural network tend to a point of near-indistinguishability (Papyan et al., 2020). Despite this insightful finding, the underlying mechanism that enables this collapse is yet to be fully understood. Linearizations and infinite-width approximations have been proposed to explain the inner workings of neural networks (Jacot et al., 2018; Chizat et al., 2019; Radhakrishnan et al., 2023). However, these approaches often simplify the rich non-linear interactions inherent in deep networks, potentially missing out on the full spectrum of dynamics and behaviors exhibited during training and inference.

Infinite dimensional convex neural networks were introduced in Bengio et al. (2005), offering insights into their structure. Following work analyzed optimization and approximation properties of infinite convex neural

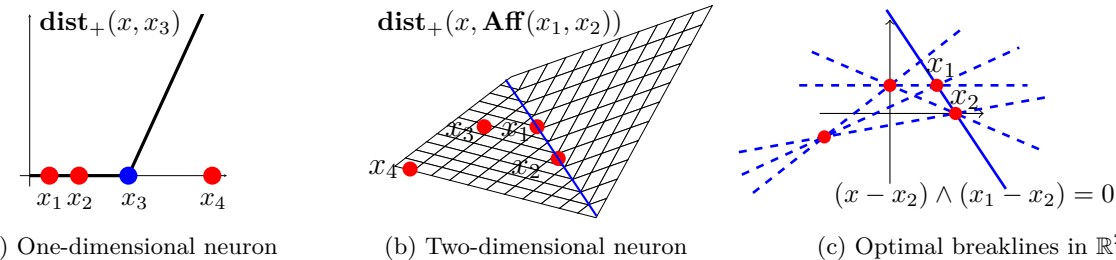

Figure 1: An illustration of the geometric interpretation of optimal ReLU neurons. The breaklines/breakpoints pass through a subset of *special* training samples.

networks (Bach, 2017). Although these works advanced the understanding of convexity in infinite dimensional NNs, they also highlight the computational challenges inherent in training infinite dimensional models, including solving a finite dimensional non-convex problems to add a single neuron (Bach, 2017). On the other hand, it has been noted that the activation patterns of deep ReLU networks exhibit a structured yet poorly understood simplicity. The work Hanin and Rolnick (2019) highlighted that the actual number of activation regions a ReLU network learns in practice is significantly smaller than the theoretical maximum.

Previous studies (Fisher and Jerome, 1975; Mammen and Van De Geer, 1997) have investigated splines in relation to $\ell_1$ extremal problems, demonstrating the adaptability of spline models to local data characteristics. More recent work (Balestriero et al., 2018) connected deep networks to spline theory, showing that many deep learning architectures could be interpreted as max-affine spline operators. Approximation properties of ReLU and squared ReLU networks with regularization were studied in Klusowski and Barron (2018). The authors in Savarese et al. (2019) considered infinitely wide univariate ReLU networks and showed that the minimum squared $\ell_2$ norm fit is given by a linear spline interpolation. Another line of work (Parhi and Nowak, 2021; 2022) developed a variational framework to analyze functions learned by deep ReLU networks, revealing that these functions belong to a space similar to classical bounded variation-type spaces. In a similar spirit, connections to kernel Banach spaces via representer theorems were developed in Bartolucci et al. (2023). The work Unser (2019) introduced a general representer theorem that connects deep learning with splines and sparsity. In contrast, our results provide an independent and novel perspective on the optimal weights of a deep neural networks through geometric algebra, and may shed light into the spline theory of deep networks. In particular, the relation between linear splines and one-dimensional ReLU networks discovered in Savarese et al. (2019) is generalized to arbitrary dimension and depth in our work. The key ingredient in our analysis is the use of wedge products, which are not present in any work analyzing neural networks to the best of our knowledge.

The relationship between neural networks and geometric structures has been another area of research focus. Convex optimization and convex geometry viewpoint of neural networks has been extensively studied in recent work (Pilanci and Ergen, 2020; Ergen and Pilanci, 2021a; Bartan and Pilanci, 2021a; Ergen and Pilanci, 2021b;c; Wang and Pilanci, 2022; Wang et al., 2021; Ergen et al., 2022a; Lacotte and Pilanci, 2020; Bartan and Pilanci, 2021b). However, previous works have focused mostly on computational aspects of convex reformulations. This work provides an entirely new set of convex reformulations, which are different from the ones in the literature. Two main advantages of our approach are that our results hold for arbitrary depth and dimension, and that they provide a geometric interpretation and novel closed-form formulas, which can be used to polish any existing deep neural network.

## 1.2 Summary of results

The work in this paper diverges from other approaches by using Clifford's geometric algebra and convex analysis to characterize the structure of optimal neural network weights. We show that the optimal weights for deep ReLU neural networks can be found via the closed-form formula, $x_{j_1} \wedge \ldots \wedge x_{j_k}$, known as a k-blade in geometric algebra, with $\wedge$ signifying the wedge product. For each individual neuron, this expression involves a *special* subset of training samples indexed by $(j_1, \ldots, j_k)$, which may vary across neurons. Surprisingly,

the entire network training procedure can be reinterpreted as a purely discrete problem that identifies this unique subset for every neuron. Moreover, we show that this problem can be cast as a Lasso variable selection procedure over wedge product features, algebraically encoding the geometric structure of the training dataset.

**The Unexpected Neuron Functionality**

The conventional belief suggests that artificial neurons optimize their response by aligning with the relevant input samples (Carter et al., 2019), a notion inspired by the direction-sensitive neurons observed within the visual cortex (Hubel and Wiesel, 1968). However, this interpretation hits a dead end in large DNNs, with numerous neurons responding to unrelated features, making it nearly impossible to understand the specific role of individual neurons (O'Mahony et al., 2023). Contrary to this conventional wisdom, our findings reveal that ReLU neurons are, in fact, orthogonal to a specific set of data points, due to the properties of the wedge product. As a consequence, these neurons yield a null output for this distinct subset of training samples, diverging completely from the anticipated alignment. This outcome underscores a nuanced understanding; rather than merely aligning with input samples, the neurons assess the oriented distance relative to the affine hull encapsulated by the special subset of training samples. This concept is visually explained in Figure 1 (a-b), where ReLU activation in 1D and 2D is interpreted as an oriented distance function. The optimal breaklines of the ReLU neurons, i.e., $\{x : w^T x + b = 0\}$, intersect with a select group of *special* training samples, expressed using a simple equation involving wedge products, leading to a zero output from the neurons for these instances, as illustrated in Figure 1(c). This result challenges the traditional interpretations of the role of hidden layers in DNNs, and provides a fresh perspective on the inner workings of deep neural networks. We show for the first time that geometric algebra provides the right set of concepts and tools to work with such oriented distances, enabling the transformation of the problem into a simple convex formulation.

**Decoding DNNs with Geometric Algebra**

Our results show that within a deep neural network, when an input sample $x$ is multiplied with a trained neuron, it yields the product $x^T \star (x_{j_1} \wedge \ldots \wedge x_{j_k})$. Leveraging geometric algebra, this product can be shown to be equal to the signed distance between $x$ and the linear span of the point set $x_{j_1}, \ldots, x_{j_k}$, scaled by the length of the neuron. This allows the neurons to measure the oriented distance between the input sample and the affine hull of the special subset of samples (see Figure 1 for an illustration). Rectified Linear Unit (ReLU) activations transform negative distances, representing inverse orientations, to zero. When this operation is extended across a collection of neurons within a layer, the layer's output effectively translates the input sample into a coordinate system defined by the affine hulls of a special subset of training samples. Consequently, each layer is fundamentally tied solely to a specific subset of the training data. This subset can be identified by examining the weights of a trained network. Furthermore, with access to these training samples, the entirety of the network weights can be reconstructed using the wedge product formula. Moreover, when this operation is repeated through additional ReLU layers, our analysis reveals a geometric regularity within the space partitioning of the network, highlighting a consistent pattern of translations and interactions with dual vectors (see Figure 7). This geometric elucidation sheds fresh light on the mysterious roles played by the hidden layers in DNNs.

## 1.3 Notation

We use lower-case letters for vectors and upper-case letters for matrices. The notation $[n]$ represents integers from 1 to $n$. We use a multi-index notation to simplify indexing matrices and tensors. Specifically, $j = (j_1, ..., j_k)$ is a multi-index and $\sum_j$ denotes the summation operator over all indices included in the multi-index $j$, i.e., $\sum_{j_1} \cdots \sum_{j_k}$. Given a matrix $K \in \mathbb{R}^{n \times p}$, an ordinary index $i \in [n]$, and a multi-index $j = (j_1, ..., j_k)$ where $j_i \in [d_i] \ \forall i \in [k]$, the notation $K_{ij}$ denotes the $(i, v(j_1, ..., j_k))$-th entry of this matrix, where $v$ represents the function that maps indices $(j_1, ..., j_k)$ to a one-dimensional index according to a column-major ordering. Formally, we can define $v(j_1, ..., f_d) := (j_1 - 1) + (j_2 - 1)d_1 + (j_3 - 1)d_1 d_2 + ... + (j_n - 1)d_1 d_2 ... d_{k-1}$. We allow the use of multi-index and ordinary indices together, e.g., $\sum_j a_j v_{j_1} \cdots v_{j_d} = \sum_{j_1, \cdots j_d} a_{(j_1, \cdots, j_d)} v_{j_1} \cdots v_{j_d}$. The notation $K_{i\cdot}$ denotes the $i$-th row of $K$. The notation $K_{\cdot j}$ denotes the $j$-th

column of $K$. The $p$-norm of a $d$-dimensional vector $w$ for some $p \in (0, \infty)$ is represented by $\|w\|_p$, and it is defined as $\|w\|_p \triangleq (\sum_{i=1}^d |w_i|^p)^{1/p}$. In addition, we use the notation $\|w\|_0$ to denote the number of non-zero entries of $w$ and $\|w\|_\infty$ to denote the maximum absolute value of the entries of $w$. We use the notation $\mathcal{B}_p^d$ for the unit $p$-norm ball in $\mathbb{R}^d$ given by $\{w \in \mathbb{R}^d : \|w\|_p \leq 1\}$. The notation $\mathbf{dist}(\mathbf{x}, \mathcal{Y})$ is used for the minimum Euclidean distance between a vector $\mathbf{x} \in \mathbb{R}^d$ and a subset $\mathcal{Y} \subseteq \mathbb{R}^d$. $\mathbf{Vol}(\mathcal{C})$ denotes $d$-volume of a subset $\mathcal{C} \subseteq \mathbb{R}^d$. The notation $(\cdot)_+$ is used the positive part of a real number. When applied to a scalar multiple of a pseudoscalar in a geometric algebra $\mathbb{G}^d$ such as $\alpha \mathbf{I} \in \mathbb{G}^d$, where $\mathbf{I}^2 = \pm 1$, the notation $(\alpha \mathbf{I})_+ = (\alpha)_+ \in \mathbb{R}$ represents the positive part of the scalar component of $\mathbf{I}$. We use the notation $\mathbf{Vol}_+(\cdot) = (\mathbf{Vol}(\cdot))_+$ to denote the positive part of signed volumes. We extend this notation to other functions, e.g., $\det_+(\cdot)$ denotes the positive part of the determinant, and $\mathbf{dist}_+(\cdot, \cdot)$ denotes the positive part of the Euclidean distance. $\mathbf{diam}(S)$ denotes the Euclidean diameter of a subset $S \subseteq \mathbb{R}^d$. $\mathbf{Span}(S)$ and $\mathbf{Aff}(S)$ denote the linear span and affine hull of a set of vectors $S$ respectively. We overload scalar functions to apply to vectors and matrices element-wise. For instance $(Xw)_+$ denotes the ReLU activation applied to each entry of $Xw$. We use $\gtrsim$ to denote the inequality up to a constant factor.

## 2 Setting and Methodology

### 2.1 Preliminaries

Consider a deep neural network

$$f(x) = \sigma(W^{(L)} \cdots \sigma(W^{(1)}x + b^{(1)}) \cdots + b^{(L)}), \tag{1}$$

where $\sigma : \mathbb{R} \to \mathbb{R}$ is a non-linear activation function, $W^{(1)}, \ldots, W^{(L)}$ are trainable weight matrices, $b^{(1)}, \ldots, b^{(L)}$ are trainable bias vectors and $x \in \mathbb{R}^d$ is the input. The activation function $\sigma(\cdot)$ operates on each element individually.

**Training two-layer ReLU networks**

We will begin by examining the regularized training objective for a two-layer neural network with ReLU activation function and $m$ hidden neurons.

$$p^* \triangleq \min_{W^{(1)}, W^{(2)}, b^{(1)}, b^{(2)}} \ell\Big( \sum_{j=1}^m \sigma(XW_j^{(1)} + 1_n b_j^{(1)}) W_j^{(2)} + b^{(1)}, y \Big) + \lambda \sum_{j=1}^m \|W_j^{(1)}\|_p^2 + \|W_j^{(2)}\|_p^2, \tag{2}$$

where $X \in \mathbb{R}^{n \times d}$ is the training data matrix, $W^{(1)} \in \mathbb{R}^{d \times m}$, $W^{(2)} \in \mathbb{R}^{m \times c}$ and $b^{(1)} \in \mathbb{R}^m$, $b^{(2)} \in \mathbb{R}$ are trainable weights, $\ell(\cdot, y)$ is a convex loss function, $y \in \mathbb{R}^n$ is a vector containing the training labels, and $\lambda > 0$ is the regularization parameter. Here we use the $p$-norm in the regularization of the weights. Initially, we will assume an output dimension of $c = 1$, and we will extend this to arbitrary values of $c$. Typical loss functions used in practice include squared loss, logistic loss, cross-entropy loss and hinge loss, which are convex functions.

When $p = 2$, the objective (2) reduces to the standard weight decay regularized NN problem

$$p^* = \min_{\theta, b} \ell\big( f_{\theta, b}(X), y \big) + \lambda \|\theta\|_2^2. \tag{3}$$

Here, $\theta$ is a vector containing the weights $W_1$ and $W_2$ in vectorized form and

$$f_{\theta, b}(X) \triangleq \sum_{j=1}^m \sigma(XW_j^{(1)} + 1_n b_j^{(1)}) W_j^{(2)} + 1_n b^{(2)}, \tag{4}$$

where $1_n$ is a vector of ones of length $n$. When $b$ is set to zero, we refer to the neurons in the first layer as the *bias-free* neurons. When $b$ is not set to zero, we refer to the neurons in the first layer as the *biased* neurons. Note that the bias terms are excluded from the regularization.

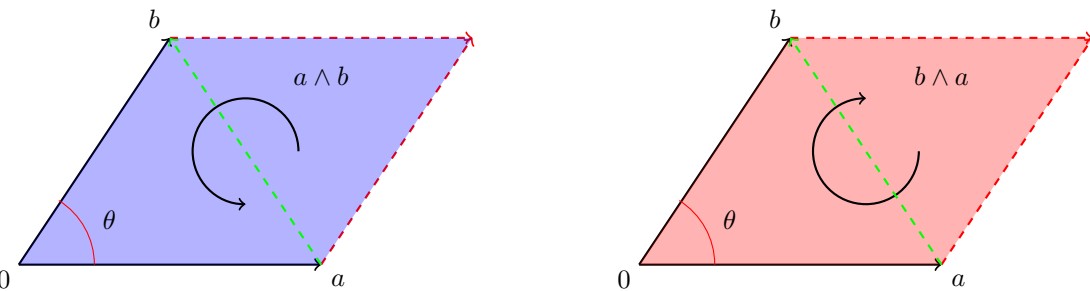

Figure 2: Wedge product of two-dimensional vectors.

**Training deep ReLU networks**

We will extend our analysis by also considering the objective in (3) using the deeper network model show in (1). It will be shown by a simple induction that the structural results for two-layer networks applies to blocks of layers in arbitrarily deep ReLU networks.

**Augmented data matrix**

In order to simplify our expressions for the case of $p = 1$, we augment the set of $n$ training data vectors of dimension $d$ by including $d$ additional vectors from the standard basis of $\mathbb{R}^d$ and let $\tilde{n} = n + d$. Specifically, we define the augmented data samples as $\{x_i\}_{i=1}^{n+d} = \{x_i\}_{i=1}^{n} \cup \{e_i\}_{i=1}^{d}$, where $x_i \in \mathbb{R}^d$ represents the original training data points for $1 \leq i \leq n$ and $e_i$ is the $i$-th standard basis vector in $\mathbb{R}^d$ for $i \geq n$. We define $\tilde{X} = [x_1, \cdots, x_{n+d}]^T$.

## 2.2 Geometric Algebra

Clifford's Geometric Algebra (GA) is a mathematical framework that extends the classical vector and linear algebra and provides a unified language for expressing geometric constructions and ideas (Artin, 2016). GA has found applications in classical and relativistic physics, quantum mechanics, electromagnetics, computer graphics, robotics and numerous other fields (Doran and Lasenby, 2003; Dorst et al., 2012). GA enables encoding geometric transformations in a form that is highly intuitive and convenient. More importantly, GA unifies several mathematical concepts, including complex numbers, quaternions, and tensors and provides a powerful toolset.

We consider GA over a $d$-dimensional Euclidean space, denoted as $\mathbb{G}^d$. The fundamental object in $\mathbb{G}^d$ is the *multivector*, $M = \langle M \rangle_0 + \langle M \rangle_1 + \ldots + \langle M \rangle_d$, which is a sum of vectors, bivectors, trivectors, and so forth. Here, $\langle M \rangle_k$ denotes the k-vector part of $M$. For instance, in the two-dimensional space $\mathbb{G}^2$, a multivector can be written as $M = a + v + B$, where $a$ is a scalar, $v$ is a vector, and $B$ is a bivector. In this case, the basis elements are not just the canonical vectors $e_1, e_2$ but also the grade-2 element $e_1 e_2$.

A key operation in GA is the *geometric product*, denoted by juxtaposition of operands: $ab$. For vectors $a$ and $b$, the geometric product can be expressed as $ab = a \cdot b + a \wedge b$, where $\cdot$ denotes the dot product and $\wedge$ denotes the wedge (or outer) product.

The dot product $a \cdot b$ is a scalar representing the projection of $a$ onto $b$. The wedge product $a \wedge b$ is a bivector representing the oriented area spanned by $a$ and $b$. In the geometric algebra $\mathbb{G}^d$, higher-grade entities (trivectors, 4-vectors, etc.) can be constructed by taking the wedge product of a vector with a bivector, a trivector, and so on. A $k$-blade is a $k$-vector that can be expressed as the wedge product of $k$ vectors. For example, the bivector $a \wedge b$ is a 2-blade.

Figure 2 shows two important properties of the wedge product in $\mathbb{R}^2$:

1. Wedge product of two vectors represents the signed area of the paralleogram spanned by the two vectors. In the left figure, $a \wedge b$ is represented by the blue parallelogram. When $a, b$ are two-dimensional vectors, the

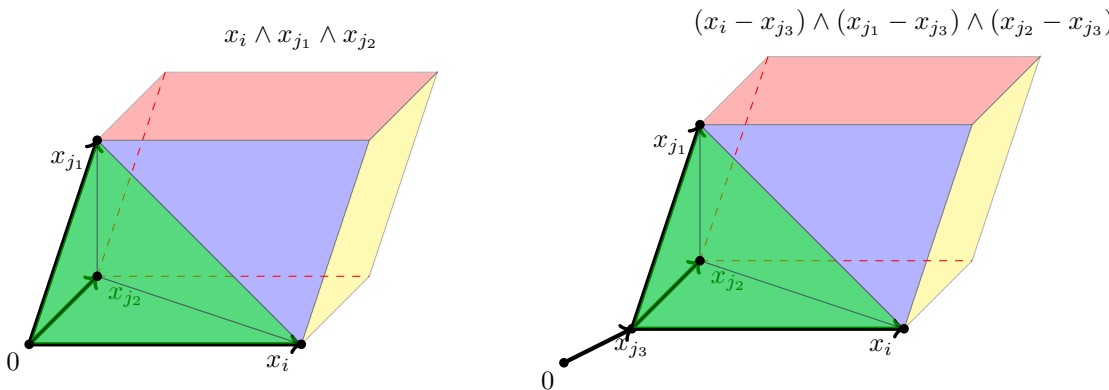

Figure 3: Wedge product representation of optimal neural networks in $\mathbb{R}^3$.

magnitude of $a \wedge b$ is equal to this area and is given by $\left| \det \begin{bmatrix} a_1 & b_1 \\ a_2 & b_2 \end{bmatrix} \right| = |a_1 b_2 - b_1 a_2|$. The sign of the area is determined by the orientation of the vectors: the area is positive when $a$ can be rotated counter-clockwise to $b$ and negative otherwise.

2. The wedge product is anti-commutative: In the right figure, $b \wedge a$ is represented by the red parallelogram. It has the same area as $a \wedge b = -b \wedge a$, but an opposite orientation. As a result, we have $a \wedge a = -a \wedge a = 0$.

By considering the half of the parallelogram, the area of the triangle in $\mathbb{R}^2$ formed by 0, $a$ and $b$ shown by the dashed line in Figure 2 is given by the magnitude of $\frac{1}{2} a \wedge b$. The signed area of a generic triangle in $\mathbb{R}^2$ formed by three arbitrary vectors $a$, $b$ and $c$ is given by

$$\frac{1}{2}(a - c) \wedge (b - c) = \frac{1}{2}\big(a \wedge b - a \wedge c - c \wedge b + c \wedge c\big) = \frac{1}{2}\big(a \wedge b + b \wedge c + c \wedge a\big),$$

which is a consequence of the distributive property of the wedge product. Therefore, the signed area of the triangle is half the sum of the wedge products for each adjacent pair in the counter-clockwise sequence $a \to b \to c \to a$ encircling the triangle. In higher dimensions, the scalar part of the wedge product represents signed the volume of a parallelotope spanned by the vectors (see Figure 3).

The metric signature in $\mathbb{G}^d$ over the Euclidean space is characterized by all positive signs, indicating that all unit basis vectors are mutually orthogonal and have unit Euclidean norm. Let us call the standard (Kronecker) basis in $d$-dimensional Euclidean space $e_1, \ldots, e_d$, which satisfies $e_i^2 = 1 \, \forall i$ and $e_i e_j = -e_j e_i \, \forall i \neq j$. The wedge product $\mathbf{I} \triangleq e_1 \wedge \cdots \wedge e_d = e_1 \ldots e_d$ represents the highest grade element and is defined to be the unit pseudoscalar. The inverse of $\mathbf{I}$ is defined as $\mathbf{I}^{-1} = e_d \ldots e_1$, and satisfies $\mathbf{I}^{-1}\mathbf{I} = 1$. Squaring $\mathbf{I}$, we obtain $\mathbf{I}^2 = (e_1 e_2)^2 = e_1 e_2 e_1 e_2 = -1$, analogous to the unit imaginary scalar in complex numbers, which is a subalgebra of $\mathbb{G}^2$.

The $\ell_2$ norm of a multivector is defined as the square root of the sum of the squares of the scalar coefficients of its basis $k$-vectors. For a multivector $M$ it holds that $\|M\|_2^2 = \langle M^\dagger M \rangle_0$, where $M^\dagger$ is the reversion of $M$. $M^\dagger$ is analogous to complex conjugation and is defined by three properties: (i) $(MN)^\dagger = N^\dagger M^\dagger$, (ii) $(M + N)^\dagger = M^\dagger + N^\dagger$, (iii) $M^\dagger = M$ when $M$ is a vector. For instance, $(e_1 e_2 e_3 e_4)^\dagger = e_4 e_3 e_2 e_1$ and $(e_1 + e_2 e_3)^\dagger = e_1 + e_3 e_2$. We define the $\ell_1$ norm of a multivector as the sum of the absolute values of each component. The definition of inner and wedge products can be naturally extended to multivectors. In particular, the inner-product between two $k$-vectors $M = \alpha_1 \wedge \cdots \wedge \alpha_k$ and $N = \beta_1 \wedge \cdots \beta_k$ is defined by the Gram determinant $M \cdot N \triangleq \det(\langle \alpha_i, \beta_j \rangle_{i,j=1}^k)$.

**Hodge dual**

A $k$-blade can be viewed as a $k$-dimensional oriented parallelogram. For each such parallelogram, we may associate a $(d - k)$-dimensional orthogonal complement. This duality between $k$-vectors and $(d - k)$-vectors

is established through the Hodge star operator $\star$. For every pair of $k$-vectors $M, N \in \mathbb{G}^d$, there exists a unique $(d-k)$-vector $\star M \in \mathbb{G}^d$ with the property that

$$\star M \wedge N = (M \cdot N)\, e_1 \wedge \cdots \wedge e_d = (M \cdot N)\, \mathbf{I}.$$

We may also express the Hodge dual of a $k$-vector $M$ as $\star M = M\mathbf{I}^{-1}$, where $\mathbf{I}^{-1}$ is the inverse of the unit pseudoscalar. This linear transformation from $d$-vectors to $d-k$ vectors defined by $M \to \star M$ is the Hodge star operator. An example in $\mathbb{G}^3$ is $\star e_1 = e_1 \mathbf{I}^{-1} = e_1 e_3 e_2 e_1 = e_3 e_2$.

**Generalized cross product and wedge products**

The usual cross product of two vectors is only defined in $\mathbb{R}^3$. However, the wedge product can be used to define a generalized cross product in any dimension. The generalized cross product in higher dimensions is an operation that takes in $d-1$ vectors in an $\mathbb{R}^d$ and outputs a vector that is orthogonal to all of these vectors. The generalized cross product of the vectors $v_1, v_2, \ldots, v_{n-1}$ can be defined via the Hodge dual of their wedge product as $\times(v_1, v_2, \ldots, v_{n-1}) \triangleq \star(v_1 \wedge v_2 \wedge \ldots \wedge v_{n-1})..$ It holds that $\times(v_1, v_2, \ldots, v_{n-1}) \cdot v_i = 0$ for all $i = 1, \ldots, n-1$. The signed distance of a vector $x$ to the linear span of a collection of vectors $x_1, ..., x_{d-1}$ can be expressed via the generalized cross product and wedge products as

$$\mathbf{dist}(x, \mathbf{Span}(x_1, \ldots, x_{d-1})) = \frac{\times(x_1, \ldots, x_{d-1})^T x}{\| \times (x_1, \ldots, x_{d-1})\|_2} = \frac{\star(x \wedge x_1 \wedge \cdots \wedge x_{d-1})}{\|x_1 \wedge \cdots \wedge x_{d-1}\|_2}.$$

This formulation stems from an intuitive geometric principle: the ratio of the volume of a parallelotope $\mathcal{P}(x, x_1, \ldots, x_{d-1})$, which is spanned by the vectors $x, x_1, \ldots, x_{d-1}$, to the volume of its base $\mathcal{P}(x_1, \ldots, x_{d-1})$, the parallelotope formed by $x_1, \ldots, x_{d-1}$ alone. This ratio effectively captures the height of the parallelotope relative to its base, which corresponds to the distance of $x$ from the subspace spanned by $x_1, \ldots, x_{d-1}$. Note that the Hodge dual $\star$ transforms the $d$-vector in the numerator into a scalar. We provide a subset of other important properties of the generalized cross product in Section 6.1 of the Appendix.

## 2.3 Convex duality

Convexity and duality plays a key role in the analysis of optimization problems (Boyd and Vandenberghe, 2004). Here, we show how the convex dual of a non-convex neural network training problem can be used to analyze optimal weights.

**Convex duals of neural network problems**

The non-convex optimization problem (2) has a convex dual formulation derived in recent work (Pilanci and Ergen, 2020; Ergen and Pilanci, 2021a) given by

$$p^* \geq d^* \triangleq \max_{v \in \mathbb{R}^n} \; -\ell^*(v, y) \quad \text{s.t.} \quad |v^T \sigma(Xw)| \leq \lambda, \, \forall w \in \mathcal{B}_p^d, \tag{5}$$

where we take the network to be bias-free (see Section 8.2 for biased neurons). Here, $\ell^*(\cdot, y)$ is the convex conjugate of the loss function $\ell(\cdot, y)$ defined as $\ell^*(v, y) \triangleq \sup_{q \in \mathbb{R}^n} v^T q - \ell(q, y)$ and $\mathcal{B}_p^d$ is the unit $p$-norm ball in $\mathbb{R}^d$. Moreover, it was shown in Pilanci and Ergen (2020) that when $\sigma$ is the ReLU activation, strong duality holds, i.e., $p^* = d^*$, when the number of neurons $m$ exceeds a critical threshold. The value of this threshold can be determined from an optimal solution of the dual problem in (5). This result was extended to deeper ReLU networks in Ergen and Pilanci (2021c) and to deep threshold activation networks in Ergen et al. (2022b).

Our main strategy to analyze the optimal weights is based on analyzing the extreme points of the dual constraint set in (5). In the following section, we present our main results.

## 3 Theoretical Results

### 3.1 One-dimensional data

We start with the simplest case where the training data is one-dimensional, i.e., $d = 1$ and the number of training points, $n$ is arbitrary.

**Theorem 1.** *For all values of the regularization norm $p \in [1, \infty)$, the two-layer neural network problem in (2) can be recast as the following $\ell_1$-regularized convex optimization problem*

$$\min_{\substack{z \in \mathbb{R}^{2n} \\ t \in \mathbb{R}}} \ell\left(Kz + 1_n t, y\right) + \lambda \|z\|_1 \,, \tag{6}$$

*where the entries of the matrix $K$ are given by*

$$K_{ij} \triangleq \begin{cases} (x_i - x_j)_+ & 1 \le j \le n \\ (x_{j-n} - x_i)_+ & n < j \le 2n, \end{cases} \tag{7}$$

*and the number of neurons obey $m \ge \|z^*\|_0$. An optimal network can be constructed as*

$$f(x) = \sum_{j=1}^{n} z_j^*(x - x_j)_+ + \sum_{j=1}^{n} z_{j+n}^*(x_j - x)_+ + t^*, \tag{8}$$

*where $z^*$ and $t^*$ are optimizers of (6).*

*Remark.* It will be revealed in the following sections that the term $(x_i - x_j)_+$, as present in (7), stands for the wedge product $\left(\begin{bmatrix} x_i \\ 1 \end{bmatrix} \wedge \begin{bmatrix} x_j \\ 1 \end{bmatrix}\right)_+$, yielding the positive part of the signed length of the interval $[x_i, x_j]$. Appending 1 to the vectors is due to the presence of bias in the neurons. This quantity can also be seen as a directional distance we denote as $\mathbf{dist}_+(x_i, x_j)$, which will be generalized to higher dimensions in the sequel. As we delve into higher dimensional NN problems, the above wedge product expression will be substituted by the positive part of the signed volume of higher dimensional simplices such as triangles and parallelograms.

*Remark.* This result is a refinement of the linear spline characterization of one-dimensional infinitely wide ReLU NNs (Savarese et al., 2019; Parhi and Nowak, 2020). The linear spline dictionary is given by the collection of ramp functions $\{(x - x_j)_+, (x_j - x)_+\}_{j=1}^{n}$, which is well-known in adaptive regression splines (Friedman, 1991). Our work is the first to recognize this dictionary via wedge products, characterize it as a finite dimensional Lasso problem, and associate it to volume forms. This enables us to generalize the result to higher dimensions and arbitrarily deep ReLU networks. It is important to note that unlike the existing literature on infinite neural networks (Bach, 2017; Bengio et al., 2005), our characterization of the dictionaries is discrete rather than continuous, enabling standard convex programming.

An important feature of the optimal network in (8) is that the break points are located only at the training data points. In other words, the prediction $f(x)$ is a piecewise linear function whose slope only may change at the training points with at most $n$ breakpoints. However, since the optimal $z^*$ is sparse, the number of pieces is at most $\|z^*\|_0$, and can be smaller than $n$.

We note that convex programs for deep networks trained for one-dimensional data were considered in Ergen et al. (2022b); Zeger et al. (2024). In Ergen and Pilanci (2021a), it was shown that the optimal solution to (3) may not be unique and may contain break points at locations other than the training data points. However, Theorem 1 reveals that at least one optimal solution is in the form of (8). In addition, it is shown that the solution is unique when the bias terms are regularized (Boursier and Flammarion, 2023) and a skip connection is included. We discuss uniqueness in Section 5.

### 3.2 Two-dimensional data

We now consider the case $d = 2$ and use the two-dimensional geometric algebra $\mathbb{G}^2$. Interestingly, we will observe that the volume of the interval $[x_i, x_j]$ appearing above generalizes to the volume of a triangle. We

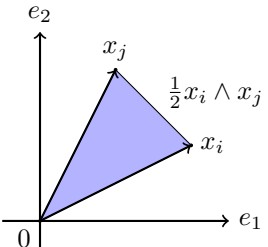 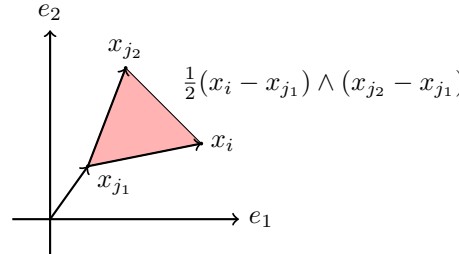

Figure 4: Illustration of the matrix $K$ for neural networks without bias (left) and with bias (right) via the triangular area defined in the convex program from Theorem 2 in $\mathbb{R}^2$.

will observe that the $\ell_1$ and $\ell_2$-regularized problems exhibit certain differences from one another. We start with the $\ell_1$-regularized problem $p = 1$, since the form of the convex program is simpler to state due to the polyhedral nature of the $\ell_1$ norm. In the case of $p = 2$, we require a mild regularity condition on the dataset to handle the curvature of the $\ell_2$ norm.

### 3.2.1 $\ell_1$ regularization - neurons without biases

**Theorem 2.** *For $p = 1$ and $d = 2$, the two-layer neural network problem without biases can be recast as the following convex $\ell_1$-regularized optimization problem*

$$\min_{z \in \mathbb{R}^{\tilde{n}}} \ell\big(Kz, y\big) + \lambda \|z\|_1. \tag{9}$$

*Here, the matrix $K \in \mathbb{R}^{\tilde{n} \times \tilde{n}}$ is defined as $K_{ij} = \kappa(x_i, x_j)$ where*

$$\kappa(x, x') := \frac{(x \wedge x')_+}{\|x'\|_1} = \frac{2\mathbf{Vol}_+(\triangle(0, x, x'))}{\|x'\|_1}, \tag{10}$$

*provided that the number of neurons satisfy $m \geq \|z^*\|_0$. Here, $\triangle(0, x_i, x_j)$ denotes the triangle formed by the path $0 \to x_i \to x_j$, and $\mathbf{Vol}_+(\cdot)$ denotes the positive part of the signed area of this triangle.*

*An optimal network can be constructed as follows:*

$$f(x) = \sum_{j=1}^{\tilde{n}} z_j^* \kappa(x, x_j),$$

*where $z^*$ is an optimal solution to (9). The optimal first layer neurons are given by a scalar multiple of the generalized cross product, $\times x_j = \star x_j$, with breaklines $x \wedge x_j = 0$, corresponding to non-zero $z_j^*$ for $j \in [\tilde{n}]$.*

*Remark.* The optimal hidden neurons given by the generalized cross products, $\times x_j$, are orthogonal to the training data points $x_j$ for $j = 1, ..., \tilde{n}$. Therefore, the breaklines of each ReLU neuron pass through the origin and some data point $x_j$. Note that $(x \wedge x_j)_+ = (x_1 x_{j2} - x_2 x_{j1})_+$ is a ReLU ridge function.

We provide an illustration the matrix $K$ in Figure 4. The signed area of the triangle formed by the path $0 \to x_i \to x_j$, denoted by the wedge product $\frac{1}{2} x_i \wedge x_j$ is positive when this path is ordered counterclockwise and negative otherwise. In this figure, the positive part of this signed area given by $\mathbf{Vol}_+(\triangle(0, x_i, x_j)) = \frac{1}{2}(x_i \wedge x_j)_+$ is non-zero only when $x_i$ is to the right of the line passing through the origin and $x_j$. When bias terms are added to the neurons, the matrix $K$ changes as shown in the right panel of Figure 4 as shown in Theorem 3 (see also Section 8.2 in the Supplementary Material).

### 3.2.2 $\ell_2$ regularization (weight decay) - neurons with biases

Now we show that similar results holds for the $\ell_2$ regularization (weight decay) for a near-optimal solution, also demonstrate the case with biases. We first quantify near-optimality as follows:

**Definition 1** (Near-optimal solutions). We call a set of parameters that achieve the cost $\hat{p}$ in the two-layer NN objective (3) $\epsilon$-optimal if

$$p^* \leq \hat{p} \leq (1 + \epsilon)p^*, \tag{11}$$

where $p^*$ is the global optimal value of (3) and $\epsilon > 0$.

**Definition 2** (Range dispersion in $\mathbb{R}^2$). We call that a two-dimensional dataset is $\epsilon$-dispersed for some $\epsilon \in (0, \frac{1}{2}]$ if

$$|\theta_{i+1} - \theta_i| \pmod{\pi} \leq \epsilon\pi \quad \forall i \in [n], \tag{12}$$

where $\theta_i$ are the angles of the vector $x_i$ with respect to the horizontal axis, i.e.,

$$x_i = \|x_i\|_2 \big[ \cos(\theta_i) \ \sin(\theta_i) \big]^T$$

sorted in increasing order. We call the dataset locally $\epsilon$-dispersed if the above condition holds for the set of differences $\{x_i - x_j\}_{i=1}^n$ for all $j \in [n]$.

Range dispersion measures the diversity of the normal planes corresponding to the training data. Local dispersion holds when the data centered at any training sample is dispersed. In the left panel of Figure 5, we illustrate the range dispersion condition for a two-dimensional dataset.

**Theorem 3.** *Suppose that the training set $\{x_1, \ldots, x_n\}$ is locally $\epsilon$-dispersed. For $p = 2$ and $d = 2$, an $\epsilon$-optimal network can be found via the following convex optimization problem*

$$\min_{\substack{z \in \mathbb{R}^{\binom{\tilde{n}}{2}} \\ t \in \mathbb{R}}} \ell\big(Kz + \mathbf{1}t, y\big) + \lambda\|z\|_1, \tag{13}$$

*when the number of neurons obey $m \geq \|z^*\|_0$. Here, the matrix $K \in \mathbb{R}^{\tilde{n} \times \binom{\tilde{n}}{2}}$ is defined as $K_{ij} := \kappa(x_i, x_{j_1}, x_{j_2})$ for $j = (j_1, j_2)$, where*

$$\kappa(x, x', x'') = \frac{\big(x \wedge x' + x' \wedge x'' + x'' \wedge x\big)_+}{\|x' \wedge x''\|_2} = \frac{2\mathbf{Vol}_+(\triangle(x, x', x''))}{\|x' - x''\|_2} = \mathbf{dist}_+(x, \mathbf{Aff}(x', x'')),$$

*for $j = (j_1, j_2)$. The $\epsilon$-optimal neural network can be constructed as*

$$f(x) = \sum_{j=(j_1, j_2)} z_j^* \kappa(x, x_{j_1}, x_{j_2}),$$

*where $z^*$ is an optimal solution to (13). The optimal hidden neurons and biases are given by a scalar multiple of $\star(x_{j_1} - x_{j_2})$, and $-\star(x_{j_2} \wedge (x_{j_1} - x_{j_2}))$ respectively, with breaklines $(x - x_{j_2}) \wedge (x_{j_1} - x_{j_2}) = 0$ for $j = (j_1, j_2)$ corresponding to non-zero $z_j^*$.*

*Remark.* We note that the form of the near-optimal NN for $p = 2$ is near identical to the $p = 1$ case, for which the result is exact. This discrepancy is due to the polyhedral nature of the dual problem with $\ell_1$ regularization (see Figure 40 of Appendix II). In Section 4, we present numerical evidence that the decision regions of optimal NNs with $p = 1$ and $p = 2$ are near identical for small values of $\lambda$.

### 3.3 Arbitrary dimensions

Now we consider the generic case where $d$ and $n$ are arbitrary and we use the $d$-dimensional geometric algebra $\mathbb{G}^d$. Suppose that $X \in \mathbb{R}^{n \times d}$ is a training data matrix such that $\mathbf{rank}(X) = d$ without loss of generality. Otherwise, we can reduce the dimension of the problem to $\mathbf{rank}(X)$ using Singular Value Decomposition (see Lemma 23 in the Supplementary Material), hence $d$ can be regarded as the rank of the data. Since many datasets encountered in machine learning problems are close to low rank, this method can be used to reduce the number of variables in the convex programs we will introduce in this section.

### 3.3.1 $\ell_1$ regularization - neurons without biases

**Theorem 4.** *The 2-layer neural network problem in* (2) *when* $p = 1$ *and biases set to zero is equivalent to the following convex Lasso problem*

$$\min_z \ell(Kz, y) + \lambda\|z\|_1, \tag{14}$$

*provided that the number of neurons satisfy* $m \geq \|z^*\|_0$. *The matrix* $K$ *is defined as* $K_{ij} = \kappa(x_i, x_{j_1}, ..., x_{j_{d-1}})$ *for* $j = (j_1, ..., j_{d-1})$, *where*

$$\kappa(x, u_1, ..., u_{d-1}) = \frac{x^T \times (x_{j_1}, \cdots, x_{j_{d-1}})}{\| \times (x_{j_1}, \cdots, x_{j_{d-1}})\|_2} = \frac{\mathbf{Vol}_+(\mathcal{P}(x, u_1, ..., u_{d-1}))}{\| \star (u_1 \wedge ... \wedge u_{d-1})\|_1}, \tag{15}$$

*the multi-index* $j = (j_1, ..., j_{d-1})$ *indexes over all combinations of* $d-1$ *rows* $x_{j_1}, ..., x_{j_{d-1}} \in \mathbb{R}^d$ *of* $\tilde{X} \in \mathbb{R}^{\tilde{n} \times d}$. *An optimal neural network can be constructed as follows:*

$$f(x) = \sum_{j = (j_1, ..., j_{d-1})} z_j^* \kappa(x, x_{j_1}, ..., x_{j_{d-1}}),$$

*where* $z^*$ *is an optimal solution to* (14). *The optimal neurons are given by a scalar multiple of the generalized cross-product* $\times(x_{j_1}, \cdots, x_{j_{d-1}}) = \star(x_{j_1} \wedge \cdots \wedge x_{j_{d-1}})$, *with breaklines* $x \wedge x_{j_1} \wedge \cdots \wedge x_{j_{d-1}} = 0$, *corresponding to non-zero* $z_j^*$ *for* $j = (j_1, \cdots, j_{d-1})$.

We recall that $\mathcal{P}(x, u_1, ..., u_{d-1})$ denotes the parallelotope formed by the vectors $x, u_1, ..., u_{d-1}$, and the positive part of the signed volume of this parallelotope is given by $\mathbf{Vol}_+(\mathcal{P}(x, u_1, ..., u_{d-1}))$.

*Remark.* The optimal neurons are orthogonal to $d - 1$ data points, i.e., $\times(x_{j_1}, \cdots, x_{j_{d-1}}) \cdot x_i = 0$ for all $i \in \{j_1, \cdots, j_{d-1}\}$. Therefore, the hidden ReLU neuron is activated on a halfspace defined by the hyperplane that passes through data points $x_{j_1}, \cdots, x_{j_{d-1}}$.

The proof of this theorem can be found in Section 9.2 of the Supplementary Material.

*Remark.* We note that the combinations can be taken over $d - 1$ linearly independent rows of $X$ since otherwise the volume is zero and corresponding weights can be set to zero. Moreover, the permutations of the indices $x_{j_1}, ..., x_{j_{d-1}}$ may only change the sign of the volume $\mathbf{Vol}(\mathcal{P}(x_i, x_{j_1}, ..., x_{j_{d-1}}))$. Therefore, it is sufficient to consider each subset that contain $d - 1$ linearly independent data points and compute $\mathbf{Vol}_+(\pm\mathcal{P}(x_i, x_{j_1}, ..., x_{j_{d-1}}))$ for each subset. The cost of enumerating over all size $d$ subsets is $O(\binom{n}{d})$.

### 3.3.2 $\ell_2$ regularization - neurons with biases

We begin by defining a parameter that sets an upper limit on the diameter of the chambers in the arrangement generated by the rows of the training data matrix $X$.

**Definition 3.** We define the Maximum Chamber Diameter, denoted as $\mathscr{D}(X)$, using the following equation:

$$\mathscr{D}(X) := \max_{\substack{w, v \in \mathbb{R}^d, \|w\|_2 = \|v\|_2 = 1 \\ \text{sign}(Xw) = \text{sign}(Xv)}} \|w - v\|_2. \tag{16}$$

Here $w$ and $v$ are unit-norm vectors in $\mathbb{R}^d$, such that the sign of the inner-product with the data rows are the same.

We call a dataset $\epsilon$-dispersed if $\mathscr{D}(X) \leq \epsilon$, and locally $\epsilon$-dispersed when the dataset centered at any training sample is $\epsilon$ dispersed, i.e., $\mathscr{D}(X - 1x_j^T) \leq \epsilon \; \forall j \in [n]$.

*Remark.* The quantity $\mathscr{D}(X)$ is a generalization of the 2D range dispersion in Definition 2 to arbitrary dimensions, and captures the diversity of the ranges of the hyperplanes whose normals are training points $\{x_i\}_{i=1}^n$. We prove in Section 3.5 that when the data is randomly generated, e.g., i.i.d. from a Gaussian distribution, the maximum chamber diameter $\mathscr{D}(X)$ is bounded by $(\frac{d}{n})^{1/4}$ with probability that approaches 1 exponentially fast. In Lemma 8 of Section 3.5, we show that this implies Gaussian data is $\epsilon$-dispersed and locally $\epsilon$-dispersed when as $n \gtrsim \epsilon^{-4}d$ with high probability.

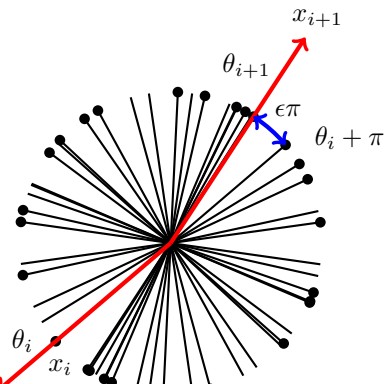 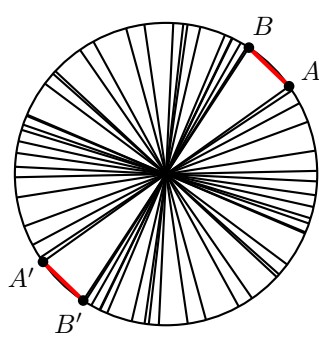

Figure 5: (left) An illustration of the angular dispersion condition. The angle between the span of two consecutive vectors $x_i$ and $x_{i+1}$ is bounded by $\epsilon\pi$. (right) The maximum chamber diameter of this line arrangement is the Euclidean distance between $A$ and $B$, i.e., $\mathscr{D}(X) = \|A - B\|$.

**Theorem 5.** *Consider the following convex optimization problem*

$$\hat{p}_\lambda := \min_z \ell(Kz, y) + \lambda\|z\|_1. \tag{17}$$

*The matrix $K$ is defined as $K_{ij} = \kappa(x_i, x_{j_1}, ..., x_{j_{d-1}})$ for bias-free and $K_{ij} = \kappa_b(x_i, x_{j_1}, ..., x_{j_{d-1}})$ for biased neurons where*

$$\kappa(x, u_1, ..., u_{d-1}) = \frac{(x \wedge u_1 \wedge \cdots \wedge u_{d-1})_+}{\|u_1 \wedge ... \wedge u_{d-1}\|_2} = \mathbf{dist}_+(x, \mathbf{Span}(u_1, ..., u_{d-1}))$$

$$\kappa_b(x, u_1, ..., u_d) = \frac{((x - u_d) \wedge (u_1 - u_d) \wedge \cdots \wedge (u_{d-1} - u_d))_+}{\|(u_1 - u_d) \wedge ... \wedge (u_{d-1} - u_d)\|_2} = \mathbf{dist}_+(x, \mathbf{Aff}(u_1, ..., d))$$

*and the multi-index $j = (j_1, ..., j_{d-1})$ is indexing over all combinations of $d - 1$ rows $x_{j_1}, ..., x_{j_{d-1}} \in \mathbb{R}^d$ of $X \in \mathbb{R}^{n \times d}$. When the maximum chamber diameter satisfies $\mathscr{D}(X) \leq \epsilon$ for bias-free neurons or $\mathscr{D}(X - 1x_j^T) \leq \epsilon \, \forall j \in [n]$ for biased neurons, for some $\epsilon \in (0, 1)$, we have the following approximation bounds*

$$p^* \leq \hat{p}_\lambda \leq \frac{1}{1 - \epsilon}p^*, \tag{18}$$

$$\hat{p}_{(1-\epsilon)\lambda} \leq p^* \leq \hat{p}_\lambda \leq p^* + \frac{\epsilon}{1 - \epsilon}\lambda R^*, \tag{19}$$

*Here, $p^*$ is the value of the optimal NN objective in (2) (with or without bias terms) and $R^*$ is the corresponding weight decay regularization term of an optimal NN. Bias-free and biased networks that achieves the cost $\hat{p}_\lambda$ in (2) are*

$$f(x) = \sum_{j=(j_1,...,j_{d-1})} z_j^* \kappa(x_{j_1}, ..., x_{j_{d-1}}) \quad and \quad f(x) = \sum_{j=(j_1,...,j_{d-1})} z_j^* \kappa_b(x_{j_1}, ..., x_{j_{d-1}}),$$

*respectively, where $z^*$ is an optimal solution to (17).*

*Remark.* The above result shows that the convex optimization produces a near-optimal solution when the chamber diameter is small, which is expected when the number of data points is large.

## 3.4 Illustrative Calculations in Geometric Algebra

Here we illustrate applications of the closed-form geometric algebra expressions in $\mathbb{R}^2$. Consider a dataset with two training samples of dimension two given by

$$x_1 = \begin{bmatrix} 1 \\ 1 \end{bmatrix} = e_1 + e_2 \quad \text{and} \quad x_2 = \begin{bmatrix} 0 \\ 2 \end{bmatrix} = 2e_2.$$

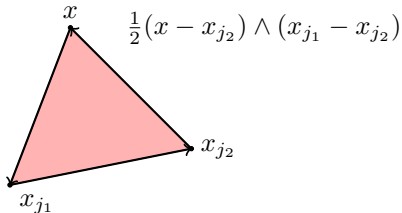

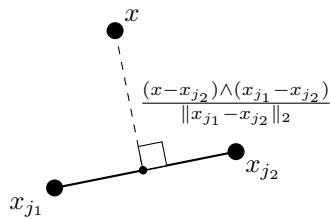

(a) The area of triangle formed by the points $x$, $x_{j_1}$ and $x_{j_2}$ is given by a wedge product.

(b) Distance to affine hull is given by a wedge product normalized by Euclidean distance.

Figure 6: Wedge product representation of distance to affine hull (a) and triangular area (b).

Theorem 4 for $p = 1$ implies that an optimal neuron corresponding to a sample $x$ is given by the dual $\star x := x\mathbf{I}^{-1}$. We recall that $\mathbf{I} = e_1e_2\ldots e_d = e_1e_2$ when $d = 2$, and $\mathbf{I}^{-1} = e_2e_1$. Hence, we have $\star x = xe_2e_1$ in $\mathbb{G}^2$. Next, recall that for $p = 1$, we augment the training set with two standard basis vectors of $\mathbb{R}^2$ and obtain $\{\tilde{x}_i\}_{i=1}^4 = \{e_1 - e_2, 2e_1, e_1, e_2\}$. For each training sample, we calculate these neurons as follows

$$w_1 = \star\tilde{x}_1 = (e_1 + e_2)e_2e_1 = \underbrace{e_1e_2}_{-e_2e_1}\, e_1 + \underbrace{e_2e_2}_{1}\, e_1 = -e_2\underbrace{e_1e_1}_{1} + e_1 = -e_2 + e_1$$

and

$$w_2 = \star\tilde{x}_2 = 2e_2e_2e_1 = 2e_1.$$

We find two additional neurons from the augmented training data points $e_1$ and $e_2$ as $w_3 = e_2$ and $w_4 = -e_1$ respectively using the same process. This yields $m = 4$ neurons, which are orthogonal to $\{\tilde{x}_i\}_{i=1}^4$.

Theorem 2 provides the optimal ReLU network with no biases as

$$f(x) = \sum_j (x^T w_j)_+ \alpha_j = \sum_{j=1}^4 \frac{\mathbf{Vol}(\triangle(0, x, \tilde{x}_j))_+}{\|\tilde{x}_j\|_1}\alpha_j,$$

where $\{\alpha_j\}_{j=1}^m$ are determined via the Lasso problem $\arg\min_\alpha \sum_{i=1}^n (f(x_i) - y_i)^2 + \lambda\|\alpha\|_1$ as described earlier in (9).

Let us illustrate some variants of this network which has no bias terms and trained with $p$-regularization where $p = 1$. For the optimal network with bias terms, we replace the volume terms with $\mathbf{Vol}(\triangle(x, \tilde{x}_{j_1}, \tilde{x}_{j_2}))$. This volume term has the loop based expansion

$$2\mathbf{Vol}(\triangle(x, \tilde{x}_{j_1}, \tilde{x}_{j_2})) = (x - \tilde{x}_{j_2}) \wedge (\tilde{x}_{j_1} - \tilde{x}_{j_2}) = x \wedge \tilde{x}_{j_1} - x \wedge \tilde{x}_{j_2} - \tilde{x}_{j_2} \wedge \tilde{x}_{j_1} - \underbrace{\tilde{x}_{j_2} \wedge \tilde{x}_{j_2}}_{0}$$

$$= x \wedge \tilde{x}_{j_1} + \tilde{x}_{j_1} \wedge \tilde{x}_{j_2} + \tilde{x}_{j_2} \wedge x.$$

In other words, the volume of the triangle formed by vertices $x, \tilde{x}_{j_1}, \tilde{x}_{j_2}$ can be expanded as the wedge product of consecutive pairs of vectors taken over the loop $x \to \tilde{x}_{j_1} \to \tilde{x}_{j_2} \to x$. This is illustrated in Figure 6a. Let us now illustrate the role of regularization. For the $p$-regularized network with $p = 2$, Theorem 3 implies that we replace the volume terms with

$$\mathbf{dist}(x, \mathbf{Aff}(x_{j_1}, x_{j_2})) = \frac{(x - x_{j_2}) \wedge (x_{j_1} - x_{j_2}))}{\|x_{j_1} - x_{j_2}\|_2} = \frac{\mathbf{Vol}(\triangle(x, x_{j_1}, x_{j_2}))}{\mathbf{dist}(x_{j_1}, x_{j_2})}.$$

This is illustrated in Figure 6b. The change from the volume to the distance to affine hull is a consequence of the division by $\|x_{j_1} - x_{j_2}\|_2$, which is the length of the base (the line segment $\mathbf{Aff}(x_{j_1}, x_{j_2})$) of the triangle $\triangle(x, x_{j_1}, x_{j_2})$.

### 3.5 Data isometry and chamber diameter

Results from high dimensional probability can be used to establish bounds on the chamber diameter of a hyperplane arrangement generated by a random collection of training points. A similar analysis was considered in Plan and Vershynin (2014) for the purpose of dimension reduction. We first show that the chamber diameter is small when the training dataset satisfies an isometry condition.

**Lemma 6** (Isometry implies small chamber diameter). *Suppose that the following condition holds*

$$(1 - \epsilon)\|w\|_2 \leq \frac{1}{\alpha n}\|Xw\|_1 \leq (1 + \epsilon)\|w\|_2, \quad \forall w \in \mathbb{R}^d, \tag{20}$$

*where $\alpha \in \mathbb{R}$ is fixed. Then, the chamber diameter $\mathscr{D}(X)$ defined in (16) is bounded by $4\sqrt{\epsilon}$.*

Next, we show that the isometry condition is satisfied when the training dataset is generated from a random distribution. Consequently, we obtain a bound on the chamber diameter of the hyperplane arrangement generated by the random dataset.

**Lemma 7** (Random datasets have small chamber diameter). *Suppose that $x_1, \cdots, x_n \sim \mathcal{N}(0, I_d)$ are $n$ random vectors sampled from the standard $d$-dimensional multivariate normal distribution and let $X = [x_1, \cdots, x_n]^T$. Then, the $\ell_2$ diameter $\mathscr{D}(X)$ satisfies*

$$\mathscr{D}(X) \leq 9\left(\frac{d}{n}\right)^{1/4}, \tag{21}$$

*with probability at least $1 - 2e^{-d/2}$.*

*Remark.* We note that the constant 9 in the above lemma can be improved to 1 by using the more refined analysis due to Gordon (Gordon, 1985). Moreover, the result can be extended to sub-Gaussian data distributions. However, we use Lemma 6 since it is simpler and suffices for our purposes.

**Lemma 8** (Random datasets are dispersed and locally dispersed). *Suppose that $x_1, \cdots, x_n \sim \mathcal{N}(0, I_d)$ are $n$ random vectors sampled from the standard $d$-dimensional multivariate normal distribution and let $X = [x_1, \cdots, x_n]^T$. Suppose that $n \gtrsim \epsilon^{-4}d$. Then, the dataset $X$ is $\epsilon$-dispersed, i.e., $\mathscr{D}(X) \leq \epsilon$ and locally $\epsilon$-dispersed, i.e., $\mathscr{D}(X - 1x_j^T) \leq \epsilon \; \forall j \in [n]$ with high probability.*

#### 3.5.1 Dvoretzky's Theorem

In this section, we present a connection to Dvoretzky's theorem (Dvoretzky, 1959), a fundamental result in functional analysis and high-dimensional convex geometry (Vershynin, 2011).

**Theorem 9** (Dvoretzky's Theorem). *(Geometric version) Let $\mathcal{C}$ be a symmetric convex body in $\mathbb{R}^n$. For any $\epsilon > 0$, there exists an intersection $\mathcal{C}_S \triangleq \mathcal{C} \cap S$ of $\mathcal{C}$ by a subspace $S \subseteq \mathbb{R}^n$ of dimension $k(n, \epsilon) \to \infty$ as $n \to \infty$ such that*

$$(1 - \epsilon)B_2 \subseteq \mathcal{C}_S \subseteq (1 + \epsilon)B_2$$

*where $B_2$ is the $n$-dimensional Euclidean unit ball.*

The above shows that there exists a $k$-dimensional linear subspace such that the intersection of the convex body $\mathcal{C}$ with this subspace is approximately spherical; that is, it is contained in a ball of radius $1 + \epsilon$ and contains a ball of radius $1 - \epsilon$.

If we represent the linear subspace in Theorem 9 via the range of the matrix $X$ and let $\mathcal{C}$ be the $\ell_1$ ball, it is straightforward to show that Dvoretzky's theorem reduces to the isometry condition (6) up to a scalar normalization. Therefore, the isometry condition (6) can be interpreted as a condition to guarantee that the $\ell_1$ ball is near-spherical when restricted to the range of the training data matrix.

### 3.6 Deep neural networks

Consider the deep neural network of $L$ layers composed of sequential two-layer blocks considered in Section (2.1) as follows

$$f_\theta(x) = W^{(L)}\sigma(W^{(L-1)} \cdots W^{(3)}\sigma(W^{(2)}\sigma(W^{(1)}x)\cdots)), \quad \theta \triangleq (W^{(1)}, \cdots, W^{(L)}).$$

Theorems 12 and 13 in this section extend Theorem 4 to three-layer ReLU networks and derive their corresponding convex Lasso formulation over a larger discrete wedge product dictionary. We first illustrate that our results apply to neural networks of arbitrary depth.

### 3.6.1 $\ell_p$ regularization

Suppose that the number of layers, $L$, is even and consider the non-convex training problem

$$p^* \triangleq \min_\theta \ell(f_\theta(X), y) + \lambda \sum_{\ell=1}^{L/2} \sum_{j=1}^{m} \|W_{j\cdot}^{(2\ell-1)}\|_p^2 + \|W_{\cdot j}^{(2\ell)}\|_p^2, \tag{22}$$

where $X \in \mathbb{R}^{n \times d}$ is the training data matrix, $y \in \mathbb{R}^n$ is a vector containing the training labels, l and $\lambda > 0$ is the regularization parameter. Here, $f(X)$ represents the output of the deep NN over the training data matrix $X$ given by $f_\theta(X) = [f_\theta(x_1) \cdots f_\theta(x_n)]^T$. Note that the $\ell_p$ norms in the regularization terms are taken over the columns of odd layer weight matrices and rows of even weight matrices, which is consistent with the two-layer network objective in (2). When $p = 2$, the regularization term is the squared Frobenius norm of the weight matrices which reduces to the standard weight decay regularization term. We assume that the number of neurons in each layer is sufficiently large to meet the conditions required for applying Theorem 4 to two consecutive layers.

**Theorem 10** (Structure of the optimal weights for $\ell_1$ regularization)**.** *The weights of an optimal solution of* (22) *for $p = 1$ are given by*

$$W_j^{(1)} = \alpha_j^{(1)} \star (x_{j_1^{(1)}} \wedge \cdots \wedge x_{j_{d-1}^{(1)}}), \quad and$$
$$W_j^{(\ell)} = \alpha_j^{(\ell)} \star (\tilde{x}_{j_1^{(\ell)}}^{(\ell)} \wedge \cdots \wedge \tilde{x}_{j_{d-1}^{(\ell)}}^{(\ell)}), \quad for \ \ell = 3, 5, ..., L-1, \tag{23}$$

*where $\alpha_j^{(\ell)}$ are scalar weights and $\tilde{x}_i^{(\ell)} \triangleq \sigma(W^{(\ell-1)} \cdots \sigma(W^{(1)} x_i) \cdots) \forall i$. Here, $W^{(\ell)}$ are optimal weights of the $\ell$-th layer for the problem* (22)*, and $j_k^{(\ell)} \in [n]$ are certain indices.*

### 3.6.2 $\ell_2$ regularization

Consider the training problem (22) with $p = 2$, which simplifies to

$$p^* \triangleq \min_\theta \ \ell(f_\theta(X), y) + \lambda \sum_{\ell=1}^{L} \|W^{(\ell)}\|_F^2. \tag{24}$$

**Theorem 11** (Structure of the optimal weights for $\ell_2$ regularization)**.** *Consider an approximation of the optimal solution of* (24) *given by*

$$W_j^{(1)} = \alpha_j^{(1)} \star (x_{j_1^{(1)}} \wedge \cdots \wedge x_{j_{d-1}^{(1)}}), \quad and$$
$$W_j^{(\ell)} = \alpha_j^{(\ell)} \star (\tilde{x}_{j_1^{(\ell)}}^{(\ell)} \wedge \cdots \wedge \tilde{x}_{j_{d-1}^{(\ell)}}^{(\ell)}), \quad for \ \ell = 2, 3, ..., L, \tag{25}$$

*where $\alpha_j^{(\ell)}$ are scalar weights, $\tilde{x}_i^{(\ell)} \triangleq \sigma(W^{(\ell-1)} \cdots \sigma(W^{(1)} x_i) \cdots)$ and $j_k^{(\ell)} \in [n]$ are certain indices. The above weights provide the same loss as the optimal solution of* (24)*. Moreover, the regularization term is only a factor $2/(1-\epsilon)$ larger than the optimal regularization term, where $\epsilon$ is an uppper-bound on the chamber diameters $\mathscr{D}(X^\ell)$ for $\ell = 0, ..., L-2$. Here, $X^\ell = \sigma(\cdots \sigma(XW^{(1)}) \cdots W^{(\ell-1)})$ are the $\ell$-th layer activations of the network given by the weights* (25)*.*

### 3.6.3 Interpretation of the optimal weights

A fully transparent interpretation of how deep networks build representations can be given using our results. We have shown that each optimal neuron followed by a ReLU activation measures the positive distance of an

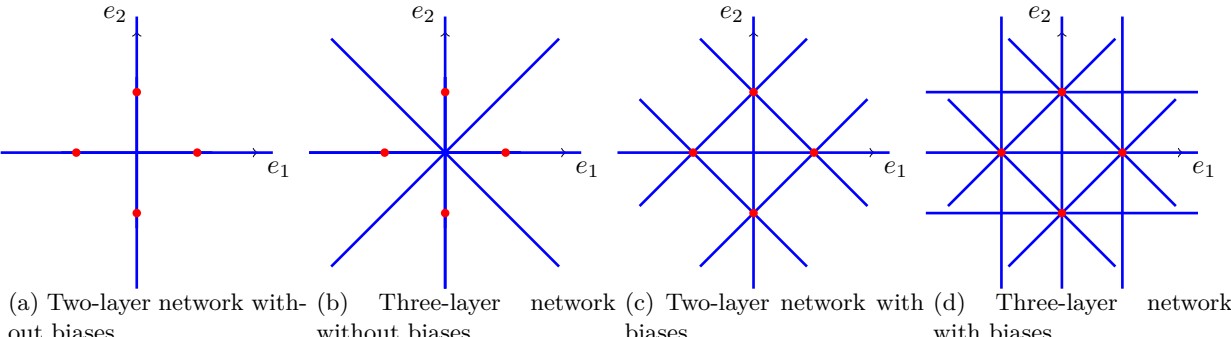

(a) Two-layer network without biases

(b) Three-layer network without biases

(c) Two-layer network with biases

(d) Three-layer network with biases

Figure 7: Optimal space partitioning of two-layer and three-layer ReLU networks predicted by Theorems 4 and 13 for $p = 1$. The blue lines represent the breaklines of optimal neurons. The red dots represent the training data points. Theorem 13 is provided in Section 3.6.

input sample to the linear span (or affine hull, in the presence of bias terms) generated by a unique subset of training points using the formula

$$(x^T \star (x_{j_1} \wedge \ldots \wedge x_{j_k}))_+ = \mathbf{dist}_+\big(x, \mathbf{Span}(x_{j_1}, ..., x_{j_{d-1}})\big).$$

The ReLU activation serves as a crucial orientation determinant in this context. By nullifying negative signed distances, it effectively establishes a directionality in the space. Geometrically speaking, it delineates the specific side of the affine hull relevant for a particular input sample. In intermediate layers, the formula is applied to the activations of the previous layer, which are themselves signed distances to affine hulls of subsets of training data.

Since each layer of the network consists of a number of neurons, the activations of the network transforms the input data into a series of distances to these unique affine hulls as

$$\big[\mathbf{dist}_+\big(x, \mathbf{Span}(x_{j_1^{(1)}}, ..., x_{j_k^{(1)}})\big), \ldots, \mathbf{dist}_+\big(x, \mathbf{Span}(x_{j_1^{(m)}}, ..., x_{j_k^{(m)}})\big)\big].$$

Moreover, the information encapsulated within the weights of the network can be succinctly represented by the indices $j_1^{(1)}, \ldots, j_k^{(1)}, \ldots, j_1^{(m)}, \ldots, j_k^{(m)}$. These indices highlight the pivotal training samples that effectively determine the geometric orientation of each neuron. The formula essentially implies that the deep neural network's behavior and decisions are intrinsically tied to specific subsets of the training data, denoted by these critical indices.

This interpretation not only offers a geometric perspective on neural networks but also explains the pivotal role hidden layers play. Each hidden layer is a series of coordinate transformations, represented by the affine hulls of various data point subsets. As the data progresses through the network, it gets transformed and re-encoded, with every neuron contributing to this transformation based on its unique geometric connection to the training dataset.

### 3.6.4 Three-layer networks with bias-free first layer

We first consider the case when the first layer neurons $W_j^{(1)}$ are size $d \times 1 \; \forall j \in [m]$ for some arbitrary $d$, and the first layer neurons are bias-free, i.e., $b_j^{(1)} = 0 \, \forall j \in [m]$. We have the following theorem.

**Theorem 12.** *The three-layer neural network problem when $p = 1$, $W_j^{(1)} \in \mathbb{R}^{d \times 1}, b_j^{(1)} = 0 \, \forall j \in [m]$, set to zero is equivalent to the following convex Lasso problem*

$$\min_{z,b} \ell\big(K^{(1)} z_1 + K^{(2)} z_2 + 1_n b, y\big) + \lambda \|z\|_1. \tag{26}$$

The matrices $K^{(1)}$ and $K^{(2)}$ are given by

$$K_{ij}^{(1)} := \frac{\left(\left(x_i \wedge \tilde{x}_{j_1} \wedge ... \wedge \tilde{x}_{j_{d-1}}\right)_+ - \left(x_{j_0} \wedge \tilde{x}_{j_1} \wedge ... \wedge \tilde{x}_{j_{d-1}}\right)_+\right)_+}{\|\tilde{x}_{j_1} \wedge ... \wedge \tilde{x}_{j_{d-1}}\|_1} \tag{27}$$

$$= \frac{\left(\mathbf{Vol}_+\left(\mathcal{P}(x_i, \tilde{x}_{j_1}, ..., \tilde{x}_{j_{d-1}})\right) - \mathbf{Vol}_+(\mathcal{P}(x_{j_0}, \tilde{x}_{j_1}, ..., \tilde{x}_{j_{d-1}}))\right)_+}{\|\tilde{x}_{j_1} \wedge ... \wedge \tilde{x}_{j_{d-1}}\|_1} \tag{28}$$

and

$$K_{ij}^{(2)} := \frac{\left(\mathbf{Vol}_+\left(\mathcal{P}(x_{j_0}, \tilde{x}_{j_1}, ..., \tilde{x}_{j_{d-1}})\right) - \mathbf{Vol}_+(\mathcal{P}(x_i, \tilde{x}_{j_1}, ..., \tilde{x}_{j_{d-1}}))\right)_+}{\|\tilde{x}_{j_1} \wedge ... \wedge \tilde{x}_{j_{d-1}}\|_1} \tag{29}$$

where $j = (j_0, j_1, ..., j_{d-1})$. The multi-index $(j_1, ..., j_{d-1})$ indexes all combinations of $d-1$ vectors from the set $\left\{\{x_i\}_{i=1}^n, \{x_i - x_j\}_{i=1,j=1}^n, \{e_k\}_{k=1}^d\right\}$ and $j_0 \in [n]$. Each optimal first layer neuron weight $w \in \mathbb{R}^d$ satisfy equalities of the form

$$x_i^T w = 0 \tag{30}$$

$$(x_i - x_j)^T w = 0 \tag{31}$$

$$e_k^T w = 0, \tag{32}$$

for a certain set of $i, j \in [n], k \in [d]$. An optimal network can be constructed as follows:

$$f(x) = \sum_j z_{1j}^* \frac{\left(\left(x \wedge \tilde{x}_{j_1} \wedge ... \wedge \tilde{x}_{j_{d-1}}\right)_+ - \left(x_{j_0} \wedge \tilde{x}_{j_1} \wedge ... \wedge \tilde{x}_{j_{d-1}}\right)_+\right)_+}{\|\tilde{x}_{j_1} \wedge ... \wedge \tilde{x}_{j_{d-1}}\|_1} \tag{33}$$

$$+ \sum_j z_{2j}^* \frac{\left(\left(x_{j_0} \wedge \tilde{x}_{j_1} \wedge ... \wedge \tilde{x}_{j_{d-1}}\right)_+ - \left(x \wedge \tilde{x}_{j_1} \wedge ... \wedge \tilde{x}_{j_{d-1}}\right)_+\right)_+}{\|\tilde{x}_{j_1} \wedge ... \wedge \tilde{x}_{j_{d-1}}\|_1}, \tag{34}$$

where $z^*$ is an optimal solution to (26).

*Remark.* In addition to the optimal neurons for the two-layer case (Theorem 4), here we obtain additional neurons orthogonal to a subset of the data points and their pairwise differences. See Figure 7(b) for an illustration.

### 3.6.5 Three-layer networks with biased neurons

We now consider the case when the first layer neurons $W_j^{(1)}$ are size $d \times 1$ $\forall j \in [m]$ for some arbitrary dimension $d$, and the all the three layers contain trainable bias terms. We have the following theorem.

**Theorem 13.** *Consider the three-layer neural network problem when $p = 1$ and $W_j^{(1)} \in \mathbb{R}^{d \times 1}$ $\forall j \in [m]$. Each optimal first layer neuron weight-bias pair $(w, b) \in \mathbb{R}^d \times \mathbb{R}$ satisfy equalities of the form*

$$(x_i - x_\ell)^T w = 0 \tag{35}$$

$$(x_i - x_j)^T w = 0 \tag{36}$$

$$e_k^T w = 0 \tag{37}$$

$$x_\ell^T w + b = 0, \tag{38}$$

*for a certain set of $i, j, \ell \in [n], k \in [d]$.*

*Remark.* In addition to the optimal neurons for the two-layer case (Theorem 16), here we obtain additional neurons whose breaklines are translations of the affine hull of a subset of the data points to certain other data points. See Figure 7(d) for an illustration.

*Remark.* We note that the optimization problem and optimal networks take a similar form as in Theorem 12, except that the $x_i$ are replaced by $x_i - x_\ell$ and the bias term $b = -x_\ell^T w$ is added.

### 3.7 Space partitioning of optimal deep networks

We now illustrate the optimal two-layer neurons predicted by Theorems 4 and compare them with optimal three-layer neurons (see Theorems 13-12 in the Section 3.6) as regularization tends to zero for $p = 1$ unless stated otherwise. Consider the two-dimensional training data $\{x_1 = (1,0), x_2 = (0,1), x_3 = (-1,0), x_4 = (0,-1)\}$ shown in Figure 7.

In panel (a), we consider a two-layer ReLU network without biases. Two optimal neurons are $(w_1^T x)_+$, $(w_2^T x)_+$, given by Theorem 2. Their breaklines, $w_1^T x = 0$ and $w_2^T x = 0$, are plotted as blue lines, and pass through the origin and data points, since the optimal neurons are scalar multiples of the Hodge duals of 1-blades formed by data points.

In panel (b), we consider a three-layer ReLU network without biases, and we display all four optimal first layer neurons given by Theorem 12. In addition to the neurons with breaklines $w_1^T x = 0$ and $w_2^T x = 0$, we also have $w_3^T x = 0$ and $w_4^T x = 0$ which are translations of the affine hulls, $\mathbf{Aff}(x_1, x_2)$ and $\mathbf{Aff}(x_2, x_3)$, to the origin.

In panel (c), we consider a two-layer ReLU network with biases regularized with $p = 2$, and we display all six optimal neurons given by Theorem 3. Their breaklines pass between each pair of samples, since the optimal neurons are scalar multiples of the Hodge duals of 1-blades formed by the differences of data points.

In panel (d), we consider a three-layer ReLU network with biases, and we display all 12 optimal first layer neurons given by Theorem 13. In addition to the breaklines that pass between each pair of samples, we also have translations of all possible affine combinations of size two, e.g., $\mathbf{Aff}(x_1, x_2)$, $\mathbf{Aff}(x_1, x_3)$,..., to every data point.

### 3.8 Vector-output Neural Networks

Consider the vector-output neural network problem in (2) given by

$$p_v^* \triangleq \min_{W^{(1)}, W^{(2)}, b} \ell\Big(\sum_{j=1}^{m} \sigma(XW_j^{(1)} + 1b_j)W_j^{(2)}, Y\Big) + \lambda \sum_{j=1}^{m} \|W_j^{(1)}\|_p^2 + \|W_j^{(2)}\|_p^2. \tag{39}$$

Here, the matrix $Y \in \mathbb{R}^{n \times c}$ contains the $c$-dimensional training labels, and $W^{(1)} \in \mathbb{R}^{d \times m}$, $W^{(2)} \in \mathbb{R}^{m \times c}$, and $b \in \mathbb{R}^m$ are trainable weights. We have the following extension of the convex progam (14) for vector-output neural networks.

$$\hat{p}_v \triangleq \min_{Z \in \mathbb{R}^{p \times c}} \ell(KZ, Y) + \lambda \sum_{j=1} \|Z_j\|_2, \tag{40}$$

where $Z_j$ is the $j$-th column of the matrix $Z$.

**Theorem 14.** *Define the matrix $K$ as follows*

$$K_{ij} = \frac{(x_i \wedge x_{j_1} \wedge \cdots \wedge x_{j_{d-1}})_+}{\|x_{j_1} \wedge ... \wedge x_{j_{d-1}}\|_p},$$

*where the multi-index $j = (j_1, ..., j_{d-1})$ is over all combinations of $r$ rows and $r = \mathbf{rank}(X)$. It holds that*

- *when $p = 1$, the convex problem (40) is equivalent to the non-convex problem (39), i.e., $p_v^* = \hat{p}_v$.*

- *when $p = 2$, the convex problem (40) is a $\frac{1}{1-\epsilon}$ approximation of the non-convex problem (39), i.e., $p_v^* \leq \hat{p}_v \leq \frac{1}{1-\epsilon} p_v^*$, where $\epsilon \in (0,1)$ is an upper-bound on the maximum chamber diameter $\mathscr{D}(X)$.*

*An neural network achieving the above approximation bound can be constructed as follows:*

$$f(x) = \sum_j Z_j^* \frac{(x_i \wedge x_{j_1} \wedge \cdots \wedge x_{j_{d-1}})_+}{\|x_{j_1} \wedge ... \wedge x_{j_{d-1}}\|_p}, \tag{41}$$

*where $Z^*$ is an optimal solution to (40).*

# 4 Numerical Results

In this section, we introduce and examine a numerical procedure to take advantage of the closed-form formulas in refining neural network parameters and producing a geometrically interpretable network.

## 4.1 Refining neural network weights via geometric algebra

We apply the characterization of Theorem 4, which states that the hidden neurons are scalar multiples of $\star(x_{j_1} \wedge \cdots \wedge x_{j_{d-1}})$, Additionally, they are orthogonal to the $r-1$ training data points specified by $x_{j_1}, \cdots, x_{j_{d-1}}$, where $r$ represents the rank of the training data matrix.

The inherent challenge lies in identifying the specific subset of the $r-1$ training points needed to form each neuron. Fortunately, this subset can be estimated when we have access to approximate neuron weights, typically acquired using standard non-convex heuristics such as stochastic gradient descent (SGD) or variants such as Adam and AdamW (Kingma and Ba, 2014; Loshchilov and Hutter, 2018). After obtaining an approximate weight vector for each neuron, we can gauge which subsets of training data are nearly orthogonal to the neuron. This is achieved by evaluating the inner-products between the neuron weight and all training vectors, subsequently selecting the $r-1$ entries of the smallest magnitude. This refinement, which we term the *polishing* process, is delineated as follows for each neuron $w_1, ..., w_m$:

For each $j \in [m]$ (optional: Append 1 to the training samples to account for the neuron bias term)

1. Calculate the inner-product magnitudes: $|x_i^T w_j|$ for each $i \in [n]$.

2. Identify the $r-1$ training vectors with the minimal inner-product magnitude, denoted as $x_{j_1}, \cdots, x_{j_{d-1}}$.

3. Update the neuron using: $w_j \leftarrow \star(x_{j_1} \wedge \cdots \wedge x_{j_{d-1}}) = \times(x_{j_1}, \ldots, x_{j_{d-1}})$. As a result, we have $w_j \perp x_{j_1}, \ldots, x_{j_{d-1}}$. This can be done by solving the linear system $w_j^T x_{j_i} = 0$ for $i = 1, ..., r-1$, or finding a minimal left singular vector of the matrix $[x_{j_1}, \ldots, x_{j_{d-1}}]$, and normalizing $w_j$ such that $\|w_j\|_p = 1$.

4. Optimize the weights of the following layer(s).

5. Optimize the scaling factors between consecutive layers (see Supplementary Material 9.13.1).

As a result, each neuron is assigned a closed-form symbolic expression, which only depends on a small subset of training samples.

### 4.1.1 Computational complexity of polishing

For a dataset of $n$ samples of dimension $d$, the cost of applying the polishing process to a layer of $m$ neurons is given by $O(mnd) + O(md^\omega)$, where $\omega$ is the matrix multiplication exponent, e.g., $\gamma = 3$ using classical solvers (Gloub and Van Loan, 1996) and $\omega \leq 2.376$ using fast matrix multiplication. We note that the latter class of algorithms are not practical for realistic sizes. However, $O(md^\omega)$ can be reduced to $O(\sqrt{\kappa} md^2 \log(1/\epsilon))$ for $\epsilon$-approximate solutions of the linear system, where $\kappa$ is the condition number. The computational cost is dominated by the calculation of inner products ($O(mnd)$) for large $n$ and the $d \times d$ linear system solve ($O(md^\omega)$) for large $d$.

### 4.1.2 Toy spiral dataset

Figure 8 for the 2D spiral dataset and a two-layer neural network optimized with squared loss. In the initial panel of this figure, the training curve of a two-layer ReLU neural network from (2) is depicted, considering $p = 2$ and weight decay regularization set at $\beta = 10^{-5}$. The dataset, divided into two classes represented by blue and red crosses, is showcased in the second panel. By resorting to the dual formulation in (5), the global optimum value is computed. Notably, while SGD is far from the global optimum, the *polishing* process enhances the neurons, leading to a marked improvement in the objective value—evidenced by the solid line in the left panel. A comparative visualization of the decision region pre and post-polishing is presented in the subsequent panels, highlighting the enhanced data distribution fit due to the polishing.

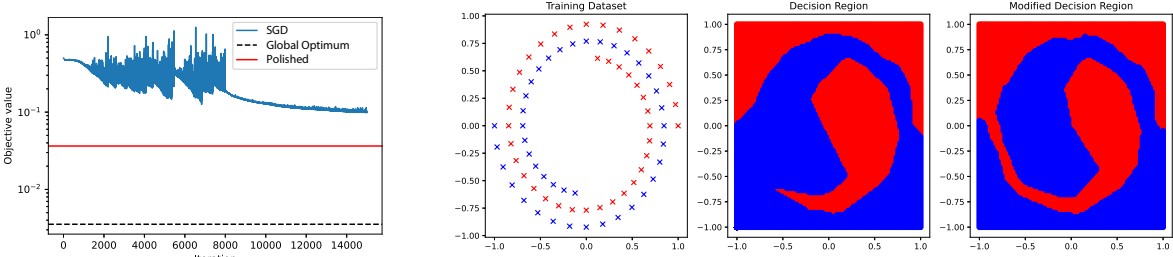

Figure 8: Comparison of SGD and polishing via geometric algebra in the spiral dataset.

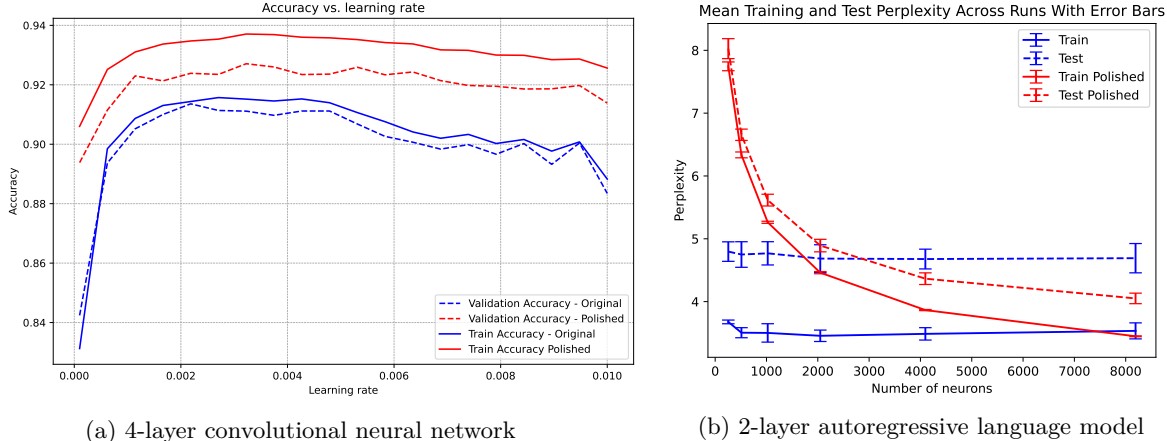

(a) 4-layer convolutional neural network      (b) 2-layer autoregressive language model

Figure 9: Comparison of AdamW and polishing CIFAR image classification (a) character-level MLP trained on a small subset of Wikipedia (b). Section 7.1 in the Supplementary Material contains additional numerical results, including a comprehensive hyperparameter search.

### 4.1.3 Real datasets

To illustrate the polishing strategy, we present three examples in Figures 9 and Figure 10.

In Figure 9 (a), we investigate binary image classification on the CIFAR dataset (Krizhevsky and Hinton, 2009). A four-layer convolutional network composed of two convolutional layers with $3 \times 3 \times 32$ filters and two fully connected layers with 512 hidden neurons is trained to distinguish class 0 (airplane) from class 2 (bird). We train it via AdamW optimizer using default hyperparameters and varying learning rate using 20 epochs and a batch size of 2048. After training, we implement the proposed polishing process on the first layer weights. Next, we re-train the second-layer weights while first layer weights are fixed via convex optimization. The resulting average train/validation accuracies are plotted over 5 independent trials to account for the randomness in optimization. We observe that the polishing process improves both the training and test accuracy. In Section 7.1.1, we provide additional results with different hyperparameters.

In Figure 9 (b), repeat the same polishing strategy for a small character-based autoregressive language model. We train a two-layer ReLU network to predict the next character in a sequence of characters from a small subset of Wikipedia consisting of first 650000 characters from the article titled 'Neural network (machine learning)' and other articles linked from the same page. We use the AdamW optimizer with a learning rate of $10^{-4}$ and a batch size of 8192. The block size is set to 16 characters. We apply polishing to the first layer weights. Next, we re-optimize the final layer weights while the first layer weights are fixed via convex optimization. The resulting average train/validation accuracies, along with 1-standard deviation error bars, are plotted over 8 independent trials to account for the randomness in optimization. We observe a significant improvement in perplexity after polishing when the number of neurons is large enough. In Section 7.1.5, we provide additional results with different hyperparameters.

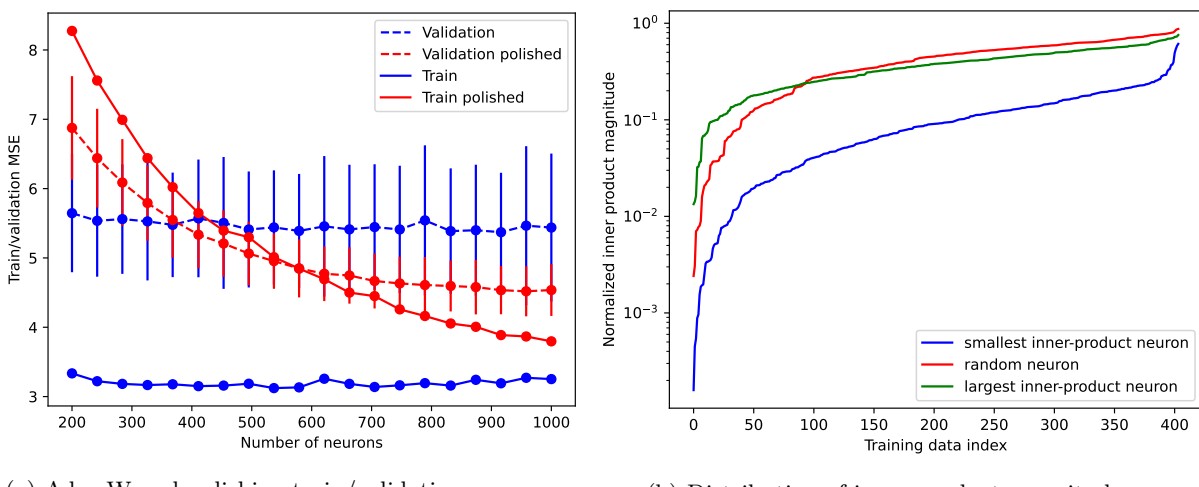

(a) AdamW and polishing train/validation errors

(b) Distribution of inner-product magnitudes

Figure 10: AdamW and polishing via geometric algebra in the Boston Housing dataset.

In Figure 10, we demonstrate the polishing strategy applied within a tabular learning context using the Boston Housing dataset. This dataset consists of 506 samples, each with 13 features representing various attributes of housing in Boston. We use a two-layer neural network with varying number of hidden neurons in the first layer, trained to predict the median value of owner-occupied homes. The network is trained using the AdamW optimizer with a learning rate of $10^{-2}$ and a batch size of 16 over 100 epochs. After training, we apply the proposed polishing process to the first layer, followed by re-optimizing the second-layer weights. The resulting train and validation MSE as well as one standard deviation error bars are plotted over 100 independent trials to account for the randomness in optimization. As shown in Figure 10 (a), the polishing process consistently results in improved performance when the number of neurons are sufficiently large, similar to previous cases, demonstrating its robustness across different types of datasets. In Figure 10 (b), we present the distribution of the magnitude of inner-products between the weight vectors of the AdamW-trained first layer before polishing. This plot shows that many inner products are small, on the order of $10^{-2}$ to $10^{-3}$, when the vectors are normalized to have a unit Euclidean norm. In Figure 39 in Section 7.6 of the Appendix, we present additional plots for traditional methods, showing that they perform worse compared to the polishing process. The work Adlam et al. (2020) reports the validation accuracy of kernel ridge regression, NN ensembles and Bayesian NNs, which all underperform compared to our approach. In the Supplementary Material (Section 7.1), we provide a detailed analysis of the effect of changing the hyperparameters and optimizers, including the learning rate, momentum parameters ($\beta_1$ and $\beta_2$ in Adam and AdamW), batch sizes, number of epochs, and also present additional results with fully connected networks and other binary classification tasks. We observe that the polishing process consistently improves the quality of the weights, leading to a significant improvement in the accuracy of the network while making the neurons fully interpretable as oriented distance functions via geometric algebra.

## 4.2 Comparison of $\ell_2$ and $\ell_1$ regularized neural networks

In this section we compare the predictions of optimal neural networks with $p$-regularized neurons, specifically focusing on $p = 2$ and $p = 1$. In Figure 11, we compare the optimal decision boundaries of $\ell_1$ and $\ell_2$ regularized NNs on the XOR dataset with $n = 4$ samples by solving the dual convex problem in (6). It can be seen that both models yield the same decision region. Moreover, two distinct breaklines of 4 optimal neurons are plotted. These can be identified as the affine hulls of a subset of training points.

In Figure 12, we compare $\ell_1$ and $\ell_2$ regularized neural networks on the spiral dataset across varying levels of regularization strength $\lambda$ by solving the convex Lasso formulations. It can be observed that the optimal

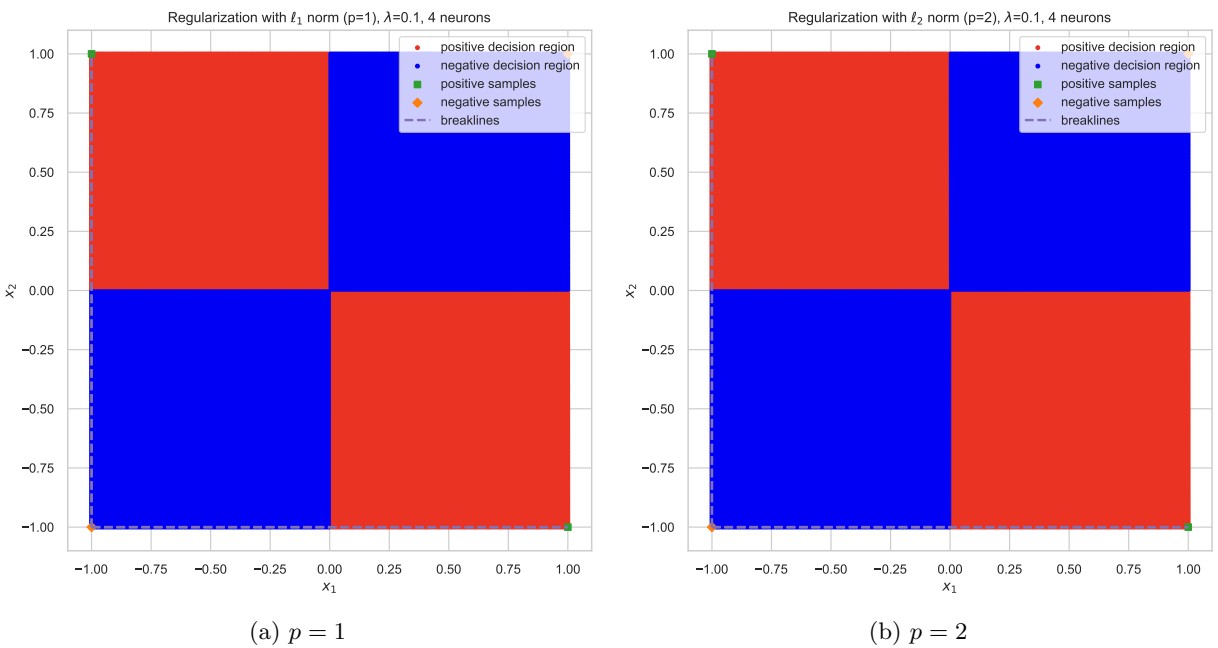

Figure 11: Comparison of $\ell_1$ and $\ell_2$ regularized neural networks on the XOR dataset. The breaklines of the optimal neurons are shown as dashed lines, representing the affine hulls $\mathbf{Aff}\left(\begin{bmatrix} -1 \\ -1 \end{bmatrix}, \begin{bmatrix} 1 \\ -1 \end{bmatrix}\right)$ and $\mathbf{Aff}\left(\begin{bmatrix} -1 \\ -1 \end{bmatrix}, \begin{bmatrix} -1 \\ 1 \end{bmatrix}\right)$ which pass through a subset of the training data points.

decision regions differ for large values of $\lambda$, whereas they become nearly identical for small values of $\lambda$. This is expected since with small $\lambda$, the network is forced to interpolate the data points regardless of the type of the regularization norm.

## 5    Discussion

In this work, we have presented an analysis that uncovers a deep connection between Clifford's geometric algebra and optimal ReLU neural networks. By demonstrating that optimal weights of such networks are intrinsically tied to the wedge product of training samples, our results enable an understanding of how neural networks build representations as explicit functions of the training data points. Moreover, these closed-form functions not only provide a theoretical lens to understand neural networks, but also has the potential to guide new architectures and training algorithms that directly harness these geometric insights.

**Computational complexity of global optimization**

The computational complexity of the polishing process is dominated by step 3, which involves solving a linear system or finding a minimal left singular vector. This can be done in $O(n^2 r)$ time using the QR decomposition or the SVD. This process is repeated for each neuron, resulting in a total complexity of $O(n^2 rm)$, where $m$ is the number of neurons. In contrast, the complexity of training the neural network to global optimal using the convex programs derived in Theorems 4 is $O(\binom{n}{r} n^2)$, which is tractable for small $r$. Note that for convolutional neural networks, the rank is bounded by the spatial size of the filter, which is a small constant (Ergen and Pilanci, 2024). Another application where the data is inherently low rank is Neural Radiance Fields (Mildenhall et al., 2021). The exponential complexity in $r$ can not be improved unless $P = NP$ (Pilanci and Ergen, 2020; Wang and Pilanci, 2023). However, the convex programs can be well-approximated by sampling the wedge products, in a similar manner to the randomized sampling employed in convex formulations of NNs (Ergen and Pilanci, 2024; 2023; Mishkin et al., 2022). Recent work showed that random sampling of polynomially many variables in the convex program (5) provides a strong approximation with only logarithmic gap to the global optimum (Kim and Pilanci, 2024). Another work

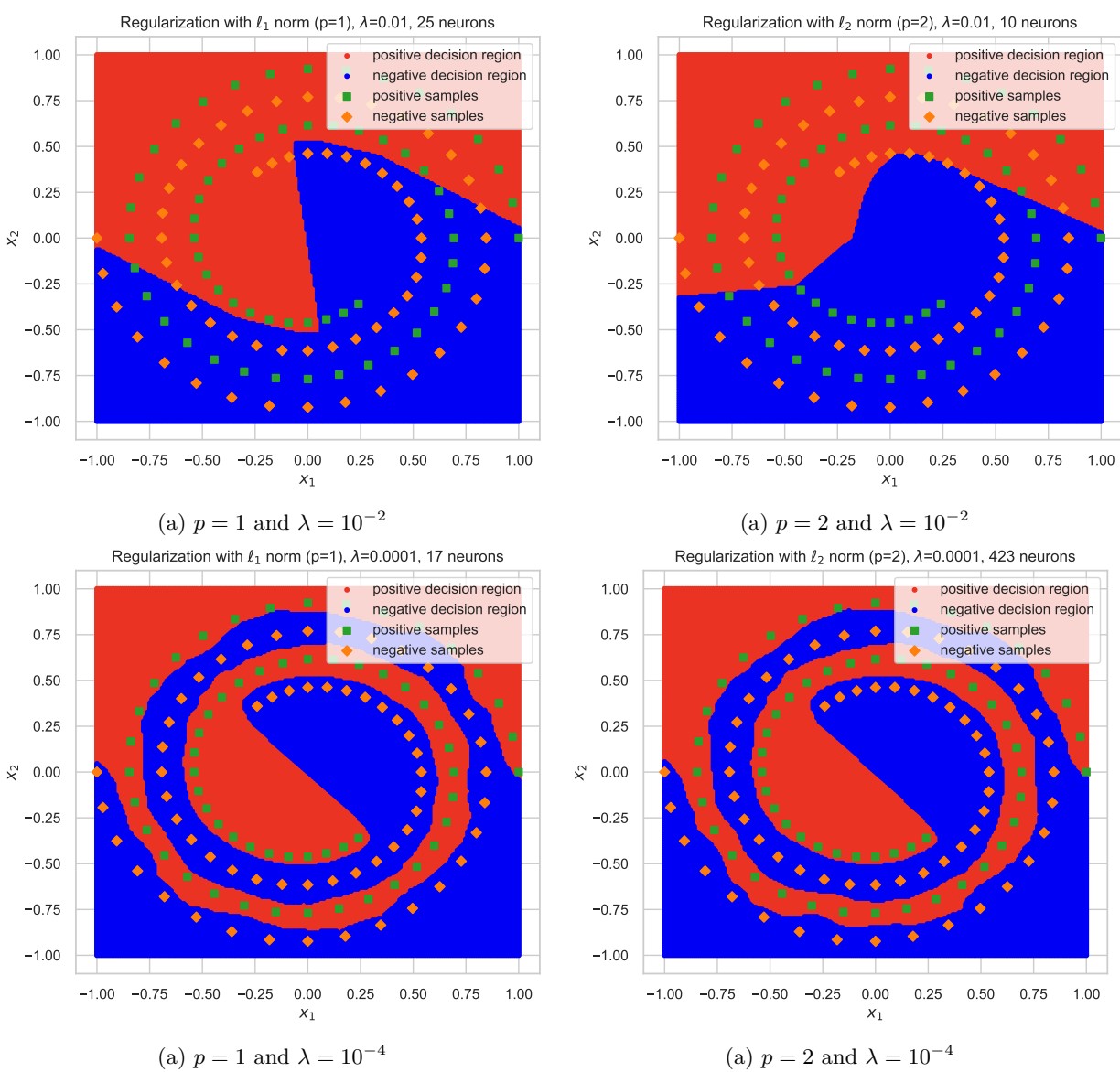

(a) $p = 1$ and $\lambda = 10^{-2}$

(a) $p = 2$ and $\lambda = 10^{-2}$

(a) $p = 1$ and $\lambda = 10^{-4}$

(a) $p = 2$ and $\lambda = 10^{-4}$

Figure 12: Comparison of $\ell_1$ and $\ell_2$ regularized neural networks on the spiral dataset. Note that the optimal decision boundaries are different for large values of $\lambda$ but become very similar for small values of $\lambda$.

(Wang et al., 2024) introduced randomized algorithms for geometric algebra, which is a promising direction to make progress in this area.

**Interpretability**

Our findings also contribute to the broader challenge of neural network interpretability. The polishing process is expected to improve the quality of the weights, leading to a significant improvement in the accuracy of the network while making the neurons fully interpretable as oriented distance functions via geometric algebra. More precisely, after polishing each ReLU neuron precisely outputs $(x_i^T w)_+ = \mathbf{dist}_+\big(x_i, \mathbf{Span}(x_{j_1}, ..., x_{j_{d-1}})\big)$. This representation is similar to that of Support Vector Machines, where the model is defined by a weighted combination of training samples. By elucidating the roles hidden layers play in encoding geometric information of training data points through signed volumes, we have taken a step towards a more transparent and foundational theory of deep learning.

**Uniqueness**

We note that the optimal weights of a ReLU neural network are not unique, and permutation, merging and splitting operations on the neurons can lead to equivalent networks. However, all globally optimal solutions can be recovered via the set of optimal solutions of the convex program (5) by considering these three operations (Mishkin and Pilanci, 2023; Wang et al., 2021). Moreover, under certain assumptions, the convex program for univariate data admits a unique solution (Boursier and Flammarion, 2023). In addition, all stationary points of the non-convex training objective can be recovered via the convex program when certain variables are constrained to be zero (Wang et al., 2021), up to permutation, merging and splitting. An important open question is characterizig the entire optimal set of the convex programs via geometric algebra, which we leave as future work.

**Other architectures**

Our findings can be extended to several other widely used network architectures. For instance, convolutional neural networks can be transformed into fully connected networks by reshuffling the data matrix, as shown in Ergen and Pilanci (2021d). Some of our results can be directly applied by redefining the training data vectors. This can be extended to deep CNNs with short receptive fields (Brendel and Bethge, 2019). For other generic CNNs, the results can be extended by employing the same approach. Additionally, we believe that our results can be extended to transformer architectures employing linear or ReLU attention, as convex formulations of these networks have been analyzed in Sahiner et al. (2022). Further results for various other neural network architectures are provided in Section 8 of the Appendix.

There are many other open questions for further research. Exploring how these insights apply to state-of-the-art network architectures, or in the context of different regularization techniques and variations of activation functions, such as the ones in attention layers, could be of significant interest. While our techniques allow for the interpretation of layer weights in popular pretrained network models, we leave this for further research. Additionally, practical implications of our results, including potential improvements to the polishing process remain to be fully explored. Our results also underlines the potential and utility of integrating geometric algebra into the theory of deep learning.

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

## 6 Appendix

### 6.1 Generalized cross products

**Definition 4.** Let $x_1, \ldots, x_{d-1} \in \mathbb{R}^d$ be a collection of $d-1$ vectors and let $A = \begin{bmatrix} x_1, \ldots, x_{d-1} \end{bmatrix} \in \mathbb{R}^{d \times (d-1)}$ be the matrix whose columns are the vectors $\{x_i\}_{i=1}^{d-1}$. The *generalized cross product* (Berger, 2009) of $x_1, \ldots, x_{d-1}$ is defined as

$$x_1 \times \ldots \times x_{d-1} = \times(x_1, \ldots, x_{d-1}) \triangleq \sum_{i=1}^{d} (-1)^{i-1} |A_i| e_i, \tag{42}$$

where $A_i$ is the square matrix obtained from $A$ by deleting the $i$-th row and $\{e_i\}_i^d$ is the standard basis of $\mathbb{R}^d$.

We next list some properties of the generalized cross product.

- The cross product $\times(x_1, \ldots, x_{d-1})$ is orthogonal to all vectors $x_1, \ldots, x_{d-1}$.

- The cross product $\times(x_1, \ldots, x_{d-1})$ equals zero if and only if $x_1, \ldots, x_{d-1}$ are linearly dependent.

- The norm of the cross product is given by the $(d-1)$-volume of the parallelotope spanned by $x_1, \ldots, x_{d-1}$, which is defined as

$$P(x_1, \ldots, x_{d-1}) \triangleq \Big\{ \sum_{i=1}^{d-1} t_i x_i \mid 0 \le t_i \le 1 \, \forall i \in [d-1] \Big\},$$

  where $x_1, \ldots, x_{d-1}$ are linearly independent.

- The cross product changes its sign when the order of two vectors is interchanged, i.e., $x_1 \times x_2 \times \ldots \times x_d = -x_2 \times x_1 \times \ldots \times x_d$ due to the determinant expansion of the cross product in (42).

- Inner-product with a vector gives the $d$-volume of the parallelotope spanned by $x_1, \ldots, x_{d-1}$ and the vector, i.e.,

$$\times(x_1, \ldots, x_{d-1})^T x = \det \begin{bmatrix} x & x_1 & \ldots & x_{d-1} \end{bmatrix} = \mathbf{Vol}\big(P(x, x_1, \ldots, x_{d-1})\big).$$

- Distance of a vector $x$ to the linear span of a collection of vectors $x_1, \ldots, x_{d-1}$ is given by

$$\mathbf{dist}(x, \mathbf{Span}(x_1, \ldots, x_{d-1})) = \frac{\mathbf{Vol}\big(P(x, x_1, \ldots, x_{d-1})\big)}{\mathbf{Vol}\big(P(x_1, \ldots, x_{d-1})\big)} = \frac{\times(x_1, \ldots, x_{d-1})^T x}{\| \times (x_1, \ldots, x_{d-1})\|_2}.$$

- Distance of a vector $x$ to the affine hull of a collection of vectors $x_1, \ldots, x_d$ is given by

$$\begin{aligned} \mathbf{dist}(x, \mathbf{Aff}(x_1, \ldots, x_{d-1})) &= \frac{\mathbf{Vol}\big(P(x - x_d, x_1 - x_d, \ldots, x_{d-1})\big)}{\mathbf{Vol}\big(P(x_1 - x_d, \ldots, x_{d-1} - x_d)\big)} \\ &= \frac{\times(x_1 - x_d, \ldots, x_{d-1} - x_d)^T (x - x_d)}{\| \times (x_1 - x_d, \ldots, x_{d-1} - x_d)\|_2} \\ &= \frac{\star\big((x_1 - x_d) \wedge \ldots \wedge (x_{d-1} - x_d) \wedge (x - x_d)\big)}{\|(x_1 - x_d) \wedge \ldots \wedge (x_{d-1} - x_d)\|_2}. \end{aligned}$$

- The rejection of a vector $x$ from the affine hull of a collection of vectors $x_1, \ldots, x_d$ can be written using Geometric Algebra $\mathbb{G}^d$ as

$$x - \mathbf{Proj}_{\mathbf{Aff}(x_1, \ldots, x_d)}(x) = \frac{(x_1 - x_d) \wedge \ldots \wedge (x_{d-1} - x_d) \wedge (x - x_d)}{(x_1 - x_d) \wedge \ldots \wedge (x_{d-1} - x_d)}.$$

When $d = 3$, the cross product reduces to the usual cross product of three vectors in $\mathbb{R}^3$. For example,

$$\begin{bmatrix} a \\ b \\ c \end{bmatrix} \times \begin{bmatrix} a' \\ b' \\ c' \end{bmatrix} = \begin{bmatrix} bc' - b'c \\ ca' - c'a \\ ab' - a'b \end{bmatrix}.$$

When $d = 2$, the cross product of a vector $\times x$ is rotation by a right angle in the clockwise direction in the plane. For example,

$$\times \left( \begin{bmatrix} a \\ b \end{bmatrix} \right) = \begin{bmatrix} b \\ -a \end{bmatrix}.$$

Detailed derivations of these results as well as further properties of generalized cross products can be found in Section 8.11 of Berger (2009), and their connections to the volumes of parallelotopes and zonotopes can be found in Gover and Krikorian (2010).

# 7   Supplementary Material I - Additional Numerical Results

## 7.1   Additional Numerical Results

In this section, we provide additional experiments showcasing the effectiveness of the proposed *polishing* process in enhancing the performance of neural networks. We consider a variety of datasets and architectures, including a three-layer fully connected ReLU network, a four-layer convolutional neural network, and a two-layer ReLU network for an autoregressive character-level text prediction task. We compare the performance of the *polishing* process with the standard methods such as SGD, Adam and AdamW, and experiment with all hyperparameters, including the learning rate, momentum parameters, number of epochs, and batch size. We also provide a comparison of the decision regions pre and post-polishing, highlighting the enhanced data distribution fit due to the polishing. It can be observed that the polishing process significantly improves the performance of the neural networks, leading to a marked improvement in the objective value and the decision regions, for a variety of datasets and architectures under different optimization hyperparameters.

### 7.1.1 Image classification and text prediction

We now consider the settings in Figure 9 and vary the optimization hyperparameters, including the momentum parameter $\beta_1$ of AdamW, the number of epochs, batch sizes for the four-layer convolutional neural network trained on the CIFAR dataset. We consider three different binary classification tasks on the CIFAR benchmark, and a character level text prediction task on the same subset of Wikipedia.

**7.1. Four-layer convolutional neural network for CIFAR class 0 vs class 1**
**(a) Varying AdamW momentum parameter $\beta_1$ and number of epochs**

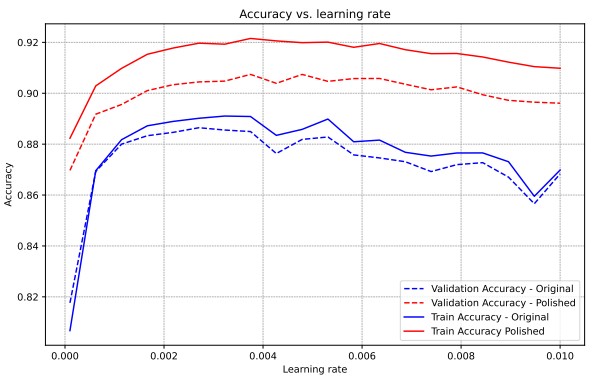

Figure 13: AdamW, $\beta_1 = 0.9$, bs=2048, epochs=10

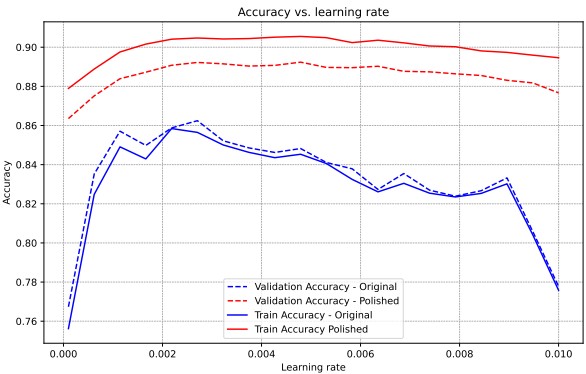

Figure 14: AdamW $\beta_1 = 0.8$, bs=2048, epochs=10

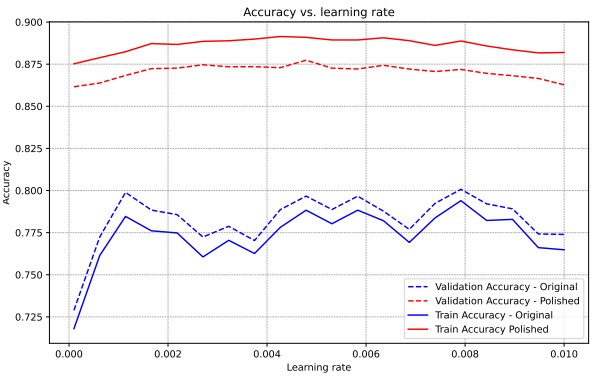

Figure 15: AdamW $\beta_1 = 0.6$, bs=2048, epochs=10

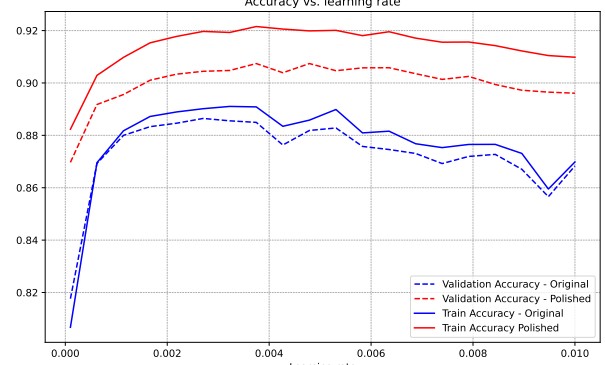

Figure 16: AdamW $\beta_1 = 0.9$, bs=1024, epochs=10

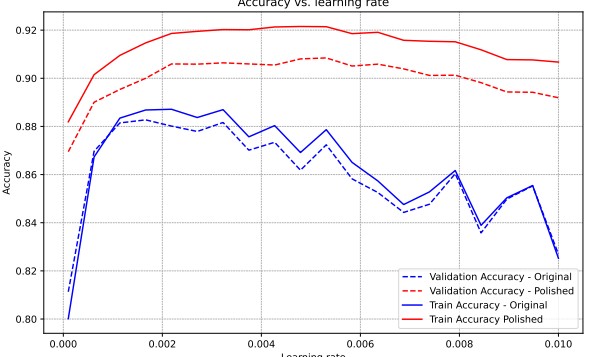

Figure 17: AdamW $\beta_1 = 0.6$, bs=2048, epochs=5

Figure 18: AdamW $\beta_1 = 0.9$, bs=2048, epochs=20

## 7.2. Four-layer convolutional network for CIFAR class 0 vs class 2

### (a) Varying AdamW momentum parameter $\beta_1$ and batch size (bs)

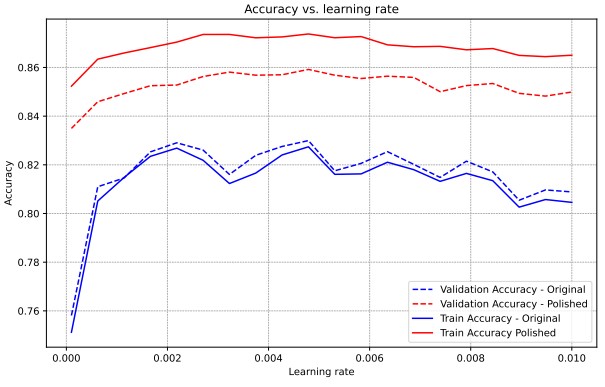

Figure 19: AdamW, $\beta_1 = 0.9$, bs=2048, epochs=10

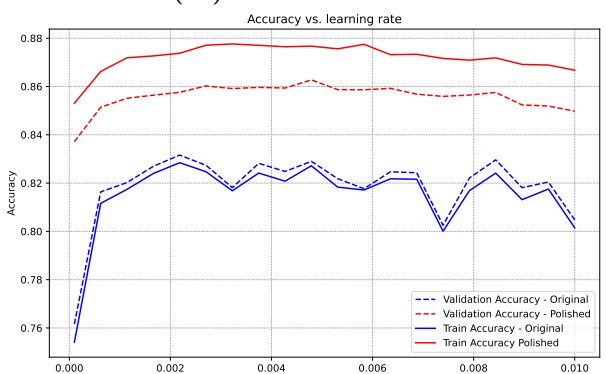

Figure 20: AdamW $\beta_1 = 0.8$, bs=2048, epochs=10

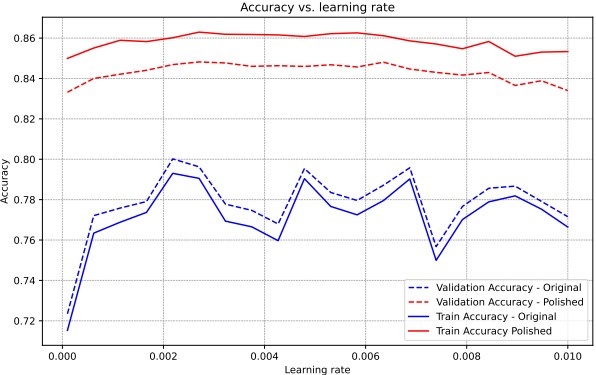

Figure 21: AdamW $\beta_1 = 0.6$, bs=2048, epochs=10

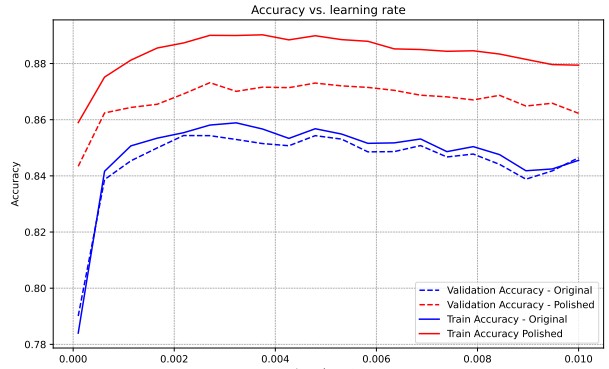

Figure 22: AdamW $\beta_1 = 0.9$, bs=1024, epochs=10

### (b) Varying number of epochs

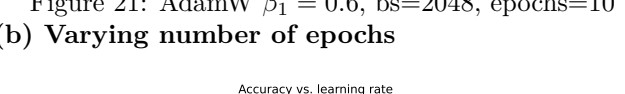

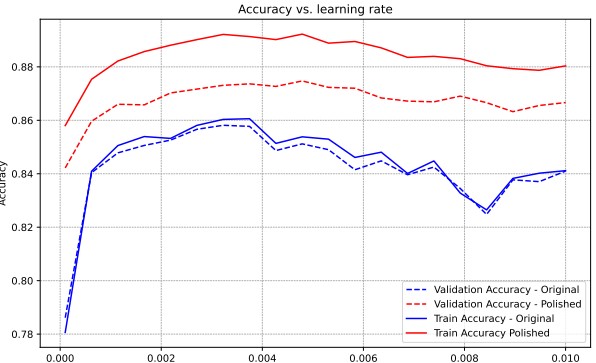

Figure 23: AdamW $\beta_1 = 0.6$, bs=2048, epochs=5

Figure 24: AdamW $\beta_1 = 0.9$, bs=2048, epochs=20

## 7.3. Four-layer convolutional neural network for CIFAR class 0 vs class 3

**(a) Varying AdamW momentum parameter $\beta_1$ and batch size (bs)**

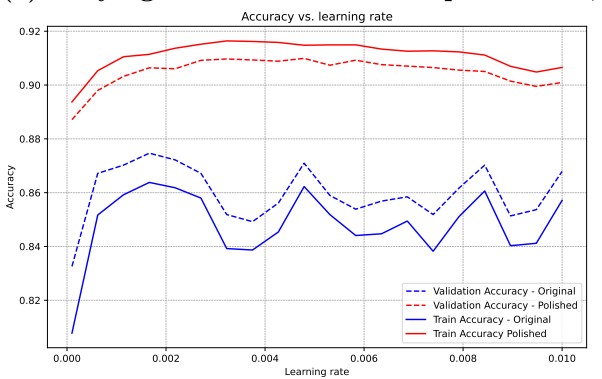

Figure 25: AdamW, $\beta_1 = 0.9$, bs=2048, epochs=10

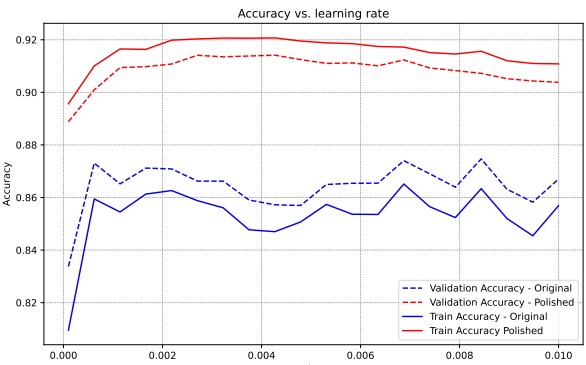

Figure 26: AdamW $\beta_1 = 0.8$, bs=2048, epochs=10

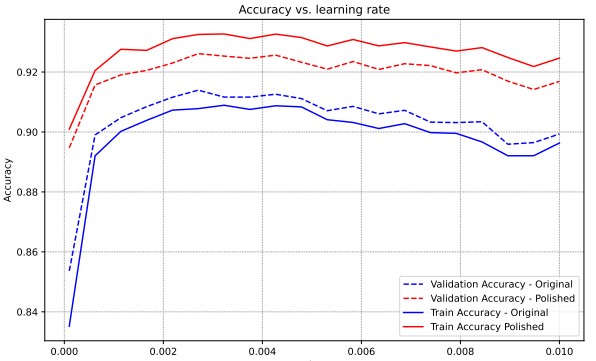

Figure 27: AdamW $\beta_1 = 0.6$, bs=2048, epochs=10

Figure 28: AdamW $\beta_1 = 0.9$, bs=1024, epochs=10

**(b) Varying number of epochs**

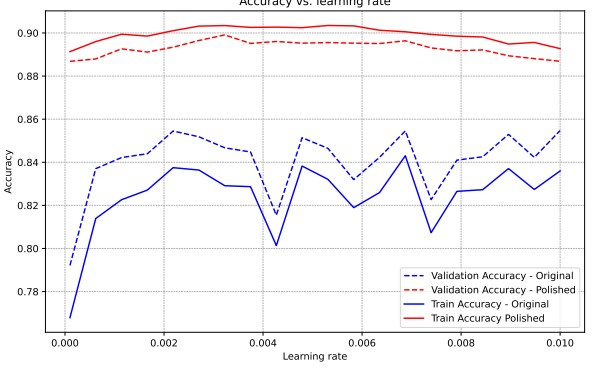

Figure 29: AdamW $\beta_1 = 0.6$, bs=2048, epochs=5

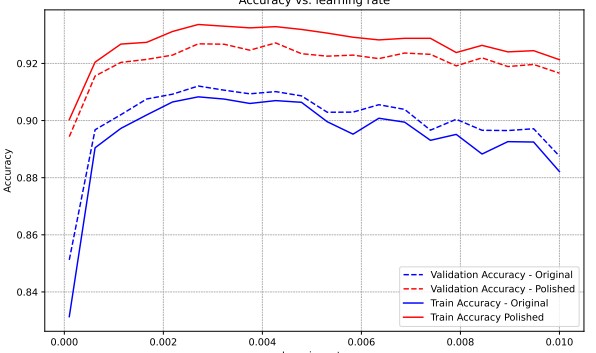

Figure 30: AdamW $\beta_1 = 0.9$, bs=2048, epochs=20

In this section, we vary the optimization hyperparameters, including the momentum parameters $\beta_1, \beta_2$ of AdamW, the number of epochs, batch sizes for a three-layer fully-connected ReLU network with 512 in each hidden layer, trained for binary classification CIFAR dataset. We consider the task of distinguishing class 1 (automobile) from 8 (ship). We display the average accuracies over 5 independent trials for each learning rate.

## 7.4. Three-layer fully connected ReLU network for CIFAR class 1 vs class 8
**(a) Varying AdamW momentum parameter $\beta_1$ when $\beta_2 = 0.999$ (default), batch size=2048, epochs=5**

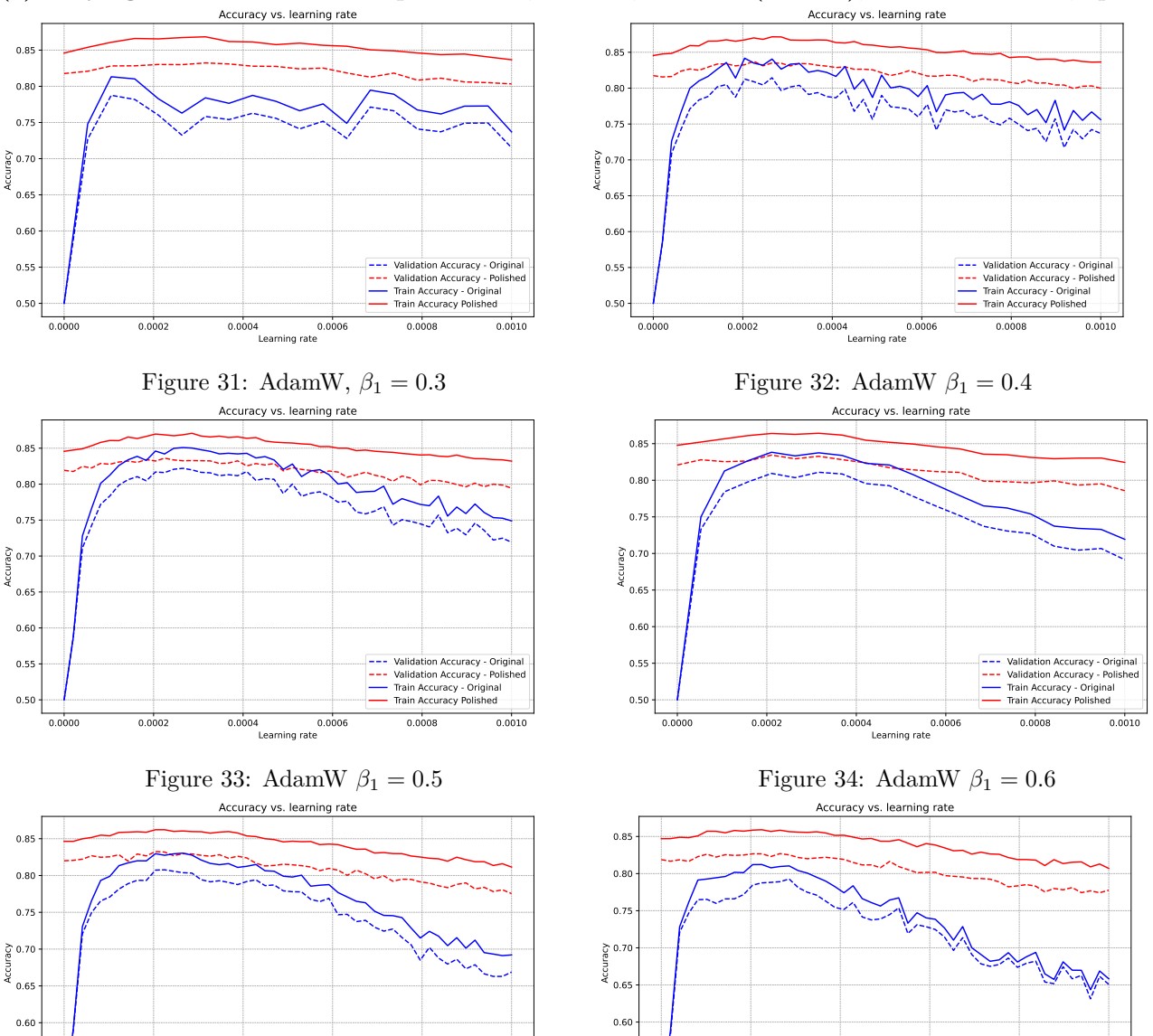

Figure 31: AdamW, $\beta_1 = 0.3$

Figure 32: AdamW $\beta_1 = 0.4$

Figure 33: AdamW $\beta_1 = 0.5$

Figure 34: AdamW $\beta_1 = 0.6$

Figure 35: AdamW $\beta_1 = 0.7$

Figure 36: AdamW $\beta_1 = 0.9$

## 7.5. Character based language model

In this section, we present additional numerical results for the character-based autoregressive language model. We compare AdamW with polishing in a small subset of text from Wikipedia, consisting 52000 characters from the article titled 'Neural network (machine learning)'. The block size is set to 16 characters and the batch size and learning rate is varied.

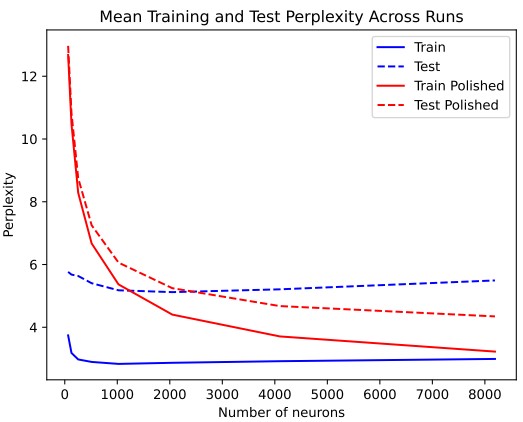
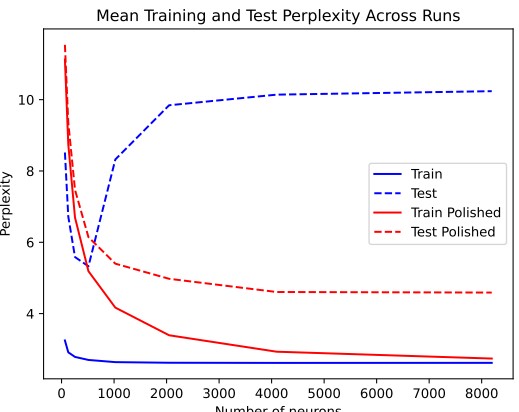

Figure 37: Character-level MLP (left) Adam with learning rate $10^{-4}$, batch size 2048 and (right) AdamW with learning rate $10^{-3}$, batch size 2048.

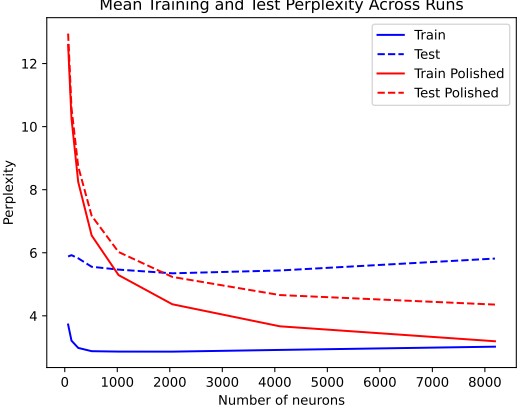
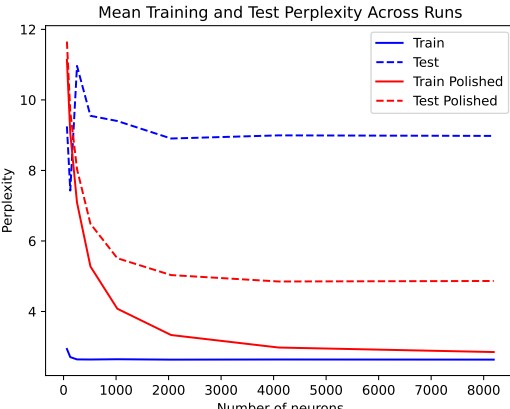

Figure 38: Character-level MLP (left) Adam with learning rate $10^{-4}$, batch size 512 and (right) AdamW with learning rate $10^{-2}$, batch size 2048.

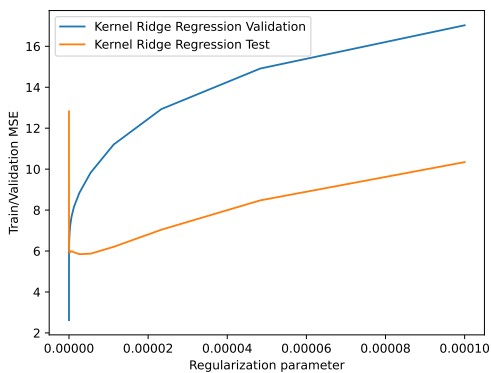

(a) Kernel ridge regression (Gaussian kernel)

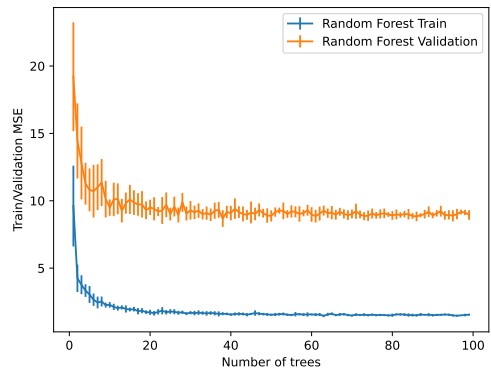

(b) Random forest regression

Figure 39: Comparison of regression models

### 7.6. Tabular Dataset

In this section, we provide further numerical results using the Boston Housing dataset. In Figure 39, we present the training and validation of MSE of kernel ridge regression using the Gaussian kernel and random forests. We provide one standard deviation error bars for the random forest over the randomness of the construction of trees. It can be seen that the polishing process shown in Figure 10 provides significantly lower validation MSE. In addition, we observed that linear regression and decision trees underperform compared to the kernel ridge regression.

## 8 Supplementary Material II - Variations of the Network Architecture

### 8.1 Two-dimensional two-layer networks with no bias

**Theorem 15.** *Suppose that the training set $\{x_1, \ldots, x_n\}$ is $\epsilon$-dispersed. For $p = 2$ and $d = 2$, an $\epsilon$-optimal network can be found via the following convex optimization problem:*

$$\min_{z \in \mathbb{R}^n} \ell\big(Kz, y\big) + \lambda \|z\|_1. \tag{43}$$

*Here, the matrix $K \in \mathbb{R}^{n \times n}$ is defined as $K_{ij} = \kappa(x_i, x_j)$, where*

$$\kappa(x, x') = \frac{(x \wedge x')_+}{\|x'\|_2} = \frac{2\mathbf{Vol}_+(\triangle(0, x, x'))}{\|x'\|_2} = \mathbf{dist}_+(x, \mathbf{Span}(x')),$$

*and the number of neurons obeys $m \geq \|z^*\|_0$. Here, $\mathbf{Span}(x_j)$ denotes the linear span of the vector $x_j$.*

*The $\epsilon$-optimal neural network can be constructed as follows:*

$$f(x) = \sum_{j=1}^{n} z_j^* \kappa(x, x_j), \tag{44}$$

*where $z^*$ is an optimal solution to (43). The optimal hidden neurons are given by scalar multiples of the Hodge duals of 1-blades formed by data points, $\star x_j = \times x_j$, corresponding to non-zero $z_j^*$.*

### 8.2 Two-dimensional two-layer networks with bias

We now augment the dataset by adding ordinary basis vectors with training points. Let us define $x_i^{(j)} = x_i$ for $i \in [n]$ $\forall j$ and $x_{n+k}^{(j)} = x_j + e_k$ for $j \in [n]$ $\forall k \in [d]$ where $e_1, \ldots, e_d$ are the canonical basis vectors. We note that this augmentation is only needed in the case of $p = 1$.

**Theorem 16.** *For $p = 1$ and $d = 2$, the two-layer neural network problem with biases is equivalent to the following convex $\ell_1$-regularized optimization problem:*

$$\min_{\substack{z \in \mathbb{R}^{\binom{n}{2}+nd} \\ t \in \mathbb{R}}} \ell\big(Kz + \mathbf{1}t, y\big) + \lambda\|z\|_1, \tag{45}$$

*provided that the number of neurons obeys $m \geq \|z^*\|_0$.*

*Here, the matrix $K \in \mathbb{R}^{n \times (\binom{n}{2}+d)}$ is defined as $K_{ij} := \kappa(x_i, x_{j_1}, x_{j_2}^{(j_1)})$ for $j = (j_1, j_2)$, where*

$$\kappa(x, x', x'') = \frac{2\mathbf{Vol}_+\big(\triangle(x, x', x'')\big)}{\|x - x'\|_1}, \tag{46}$$

*$j = (j_1, j_2)$ is a multi-index[1], $\triangle(x, x', x'')$ denotes the triangle formed by the points $x, x', x''$,*

$$\begin{aligned}
\mathbf{Vol}_+(\triangle(x, x', x'')) &= \frac{1}{2}\big((x - x') \wedge (x'' - x')\big)_+ \\
&= \frac{1}{2}\big(x \wedge x' + x' \wedge x'' + x'' \wedge x\big)_+,
\end{aligned}$$

*denotes the positive part of the signed volume of this triangle. An optimal neural network can be constructed as follows:*

$$f(x) = \sum_{j=(j_1, j_2)} z_j^* \kappa(x, x_{j_1}, x_{j_2}^{(j_1)}), \tag{47}$$

*where $z^*$ is an optimal solution to (45). The optimal hidden neurons are given by scalar multiples of the generalized cross product $\times(x_{j_1} - x_{j_2}^{(j_1)}) = \star(x_{j_1} - x_{j_2}^{(j_1)})$, which are Hodge duals of 1-blades formed by differences of data points, corresponding to non-zero $z_j^*$ for $j = (j_1, j_2)$.*

*Remark.* We note that each ReLU neuron in the optimal network has a breakline passing through a pair of training samples $x_{j_1}$ and $x_{j_2}$.

### 8.3 Two-layer networks with inputs of arbitrary dimension without biases

**Theorem 17.** *Consider the following convex optimization problem*

$$\hat{p}_\lambda := \min_z \ell\big(Kz, y\big) + \lambda\|z\|_1. \tag{48}$$

*The matrix $K$ is defined as $K_{ij} = \kappa(x_i, x_{j_1}, ..., x_{j_{d-1}})$ for $j = (j_1, ..., j_{d-1})$, where*

$$\kappa(x, u_1, ..., u_{d-1}) = \frac{\big(x \wedge u_1 \wedge \cdots \wedge u_{d-1}\big)_+}{\|u_1 \wedge ... \wedge u_{d-1}\|_2} = \mathbf{dist}_+\big(x, \mathbf{Span}(u_1, ..., u_{d-1})\big),$$

*the multi-index $j = (j_1, ..., j_{d-1})$ is over all combinations of $d-1$ rows $x_{j_1}, ..., x_{j_{d-1}} \in \mathbb{R}^d$ of $X \in \mathbb{R}^{n \times d}$. When the maximum chamber diameter satisfies $\mathscr{D}(X) \leq \epsilon$ for some $\epsilon \in (0, 1)$, we have the following approximation bounds*

$$p^* \leq \hat{p}_\lambda \leq \frac{1}{1-\epsilon}p^*, \tag{49}$$

$$\hat{p}_{(1-\epsilon)\lambda} \leq p^* \leq \hat{p}_\lambda \leq p^* + \frac{\epsilon}{1-\epsilon}\lambda R^*, \tag{50}$$

*Here, $p^*$ is the value of the optimal NN objective in (2) and $R^*$ is the corresponding weight decay regularization term of an optimal NN. A network that achieves the cost $\hat{p}_\lambda$ in (2) is*

$$f(x) = \sum_{j=(j_1, ..., j_{d-1})} z_j^* \kappa(x_{j_1}, ..., x_{j_{d-1}}),$$

*where $z^*$ is an optimal solution to (48).*

---

[1]For a multi-index $j = (j_1, j_2)$ where $j_1 \in [n], j_2 \in [n+d]$, the entry $K_{i,j}$ of the matrix $K$ is given by $K_{i,j_1+(n+d)(j_2-1)}$.

**Corollary 18.** *By taking the limit $\lambda \to 0$, we obtain the following interpolation variant of the convex NN problem* (48)

$$\hat{p}_0 := \min_{z\,:\,\ell(Kz,y)=0} \|z\|_1\,, \tag{51}$$

*provided that the constraint $\ell(Kz,y) = 0$ is feasible. In this case, we have*

$$p_0^* \le \hat{p}_0 \le \frac{1}{1-\epsilon} p_0^*, \tag{52}$$

*given that $\mathscr{D}(X) \le \epsilon$, where $p_0^* \triangleq \min_{\theta\,:\,\ell(f_\theta(X),y)=0} \|\theta\|_2^2$.*

## 9 Supplementary Material III - Mathematical Proofs

### 9.1 Proof of Theorem 1

Consider the dual constraint in (5), which can be rewritten as

$$\sup_{w,\,b\in\mathbb{R}\,:\,\|w\|_p\le 1} |\sum_i v_i(x_i w + b)_+| \le \lambda. \tag{53}$$

Since $w$ is a scalar, the constraint $\|w\|_p \le 1$ is equivalent to $|w| \le 1$ for all $p \in (0,\infty)$. An optimal $w$ satisfies $w \in \{-1,+1\}$ since either this constraint is tight or the objective is constant with respect to $w$. Next, observe that an optimal $b$ is achieved when $b \in \pm\{x_i\}_{i=1}^n$ or the objective goes to infinity when $b \to \infty$. This is because the objective is a piecewise linear function of $b$. In the latter case, the objective value is infinite as long as $\sum_i v_i \neq 0$. Therefore, we can rewrite (53) as

$$\max\left\{|\sum_i v_i(x_i - x_j)_+|, |\sum_i v_i(x_j - x_i)_+|\right\} \le \lambda \quad \text{and} \quad \sum_i v_i = 0 \tag{54}$$

We use strong Lagrangian duality to obtain the claimed convex program using Lemma 26. It can be seen that the network given in (8) achieves the optimal objective on the training dataset since $f(X) = [f(x_1), \cdots, f(x_n))]^T = Kz^* + 1t^*$.

### 9.2 Proof of Theorem 2 and Theorem 4

We now present the proof of Theorem 4, and the special case Theorem 2, which shows the equivalence of the $\ell_1$-regularized neural network and the Lasso problem in (14). Suppose that the training matrix is of rank $k$. Denote the compact Singular Value Decomposition (SVD) of $X$ as follows $X = U\Sigma V^T$ where $U \in \mathbb{R}^{n\times d}$ and $V \in \mathbb{R}^{d\times d}$ are orthonormal matrices and $\Sigma \in \mathbb{R}^{d\times d}$ is a diagonal matrix with non-negative diagonal entries. Only $k$ diagonal entries of $\Sigma$ are non-zero and the rest are zero. We denote the non-zero diagonal entries of $\Sigma$ as $\sigma_1, \cdots, \sigma_k$ and the corresponding columns of $U$ and $V$ as $u_1, \cdots, u_k$ and $v_1, \cdots, v_k$, respectively.

Consider the NN objective (2). It can be seen that the projection $v_i^T W_{1j}$ of the hidden neuron weights $j \in [m]$ does not affect the objective value for $i = k+1, ..., d$. Therefore, the optimal hidden weights are zero in the subspace $v_{k+1}, ..., v_d$ due to the norm regularization. In the remaining, we assume that the training matrix $X$ is full column rank and of size $n$ by $d = r$. Otherwise, we can remove the zero singular value subspaces of $X$ via the above argument.

We consider the convex dual of the $\ell_1$-regularized NN problem (14), and focus on the dual problem given in (5) when $\sigma$ is the ReLU activation function.

$$\max_{v\in\mathbb{R}^n} -\ell^*(v,y) \quad \text{s.t.} \quad \max_{w\,:\,\|w\|_p\le 1} |v^T(Xw)_+| \le \lambda. \tag{55}$$

The above optimization problem is a convex semi-infinite program which has a finite number of variables but infinitely many constraints. Our main strategy is to show that the constraint set can be described by a finite number of fixed points by analyzing the extreme points of certain convex subsets.

As shown in Pilanci and Ergen (2020), the constraint $|v^T(Xw)_+| \leq \lambda$ can be analyzed by enumerating the hyperplane arrangement patterns

$$\{D_k\}_{k=1}^P = \left\{\text{diag}(1[Xw \geq 0]) : w \in \mathbb{R}^d\right\},$$

where $P$ is the number of distinct hyperplane arrangement patterns and $D_k \in \{0,1\}^{n \times n}$ is the indicator matrix of the $k$-th pattern. The number $P$ is finite and satisfies the upper-bound by $P \leq 2 \sum_{j=0}^{d-1} \binom{n-1}{j}$ where $r = \mathbf{rank}(X)$. Note that we have the identity

$$(Xw)_+ = D_k Xw \quad \forall w : D_k Xw \geq 0, \ (I - D_k)Xw \leq 0,$$

which shows that the ReLU activation applied to the vector $Xw$ can be expressed as a linear function whenever $w$ is in the cone $(2D_k - I)Xw \geq 0$. Using the above parameterization, we write the subproblem in the constraint of (55) when $p = 1$ as follows

$$d_{\text{sub}} \triangleq \max_w |v^T D_k Xw| \tag{56}$$
$$\text{s.t. } \|w\|_1 \leq 1, \ (2D_k - I)Xw \geq 0.$$

We next claim that the constraint $\|w\|_1 \leq 1$ is active at the optimum, assuming the objective value is positive. Note that $w = 0$ is a feasible point, which achieves the objective value of zero. Otherwise, we can scale $w$ to satisfy the constraint and increase the objective value. Defining the set

$$\mathcal{C} \triangleq \{(2D_i - I)Xw \geq 0, \ \|w\|_1 = 1\},$$

we can express the dual subproblem $d_{\text{sub}}$ using the set $\mathcal{C}$ as follows

$$d_{\text{sub}} = \max \left\{0, \ \max_{w \in \mathcal{C}} v^T D_k Xw, \ \max_{w \in \mathcal{C}} -v^T D_k Xw\right\}.$$

Note that the maxima in the last two problems are achieved at extreme points of $\mathcal{C}$ since $\mathcal{C}$ is a bounded polytope and the objectives are linear functions.

The following lemma provides a characterization of the extreme points of the constraint set of the dual subproblem.

**Lemma 19.** *A vector $w$ is an extreme point of the set $\mathcal{C}$ if and only if $w \in \mathcal{C}$ and*

$$\begin{bmatrix} X \\ I \end{bmatrix}_S w = 0 \quad and \quad \|w\|_1 = 1 \tag{57}$$

*where $S$ is a subset of $d - 1$ linearly independent rows of $\begin{bmatrix} X \\ I \end{bmatrix}$.*

The proof of this lemma is given at the end of this subsection. We identify the extreme points via the cross product as follows. After defining the augmented data rows $x_{n+i} = e_i$ where $e_i$ is the $i$-th standard basis vector, a vector $w \in \mathcal{C}$ is an extreme point of $\mathcal{C}$ if $w \in \mathbf{Span}(\times_{j \in S} x_j)$ for some subset $S$ of size $d - 1$ such that $\{x_j\}_{j \in S}$ are linearly independent and $\|w\|_1 = 1$. Since this span is a $d - 1$ dimensional subspace, we can write $w$ as $\frac{\times_{j \in S} x_j}{\left\|\times_{j \in S} x_j\right\|_1}$. Note that the denominator is non-zero since $\{x_j\}_{j \in S}$ are linearly independent and $\|\times_{j \in S} x_j\|_1 \geq \|\times_{j \in S} x_j\|_2$, which is the non-zero absolute volume of the parallelotope spanned by $\{x_j\}_{j \in S}$. Consecutively, we apply this characterization of extreme points to the expression for $d_{\text{sub}}$ to obtain

$$d_{\text{sub}} = \max \left\{0, \ \max_{w \in V} v^T D_k Xw\right\},$$

where $\mathcal{V} \triangleq \left\{\pm \frac{\times_{j \in S} x_j}{\|\times_{j \in S} x_j\|_1} : \{x_j\}_{j \in S} \text{ linearly independent}, \ |S| = d - 1\right\}$.

Next we note that $D_k X w = (Xw)_+$ for $w \in \mathcal{V} \subseteq \mathcal{C}$, which shows that the dual problem $d^*$ in (55) is equivalent to the following problem

$$\max_{v \in \mathbb{R}^n} -\ell^*(v, y) \quad \text{s.t.} \quad |v^T (Xw)_+| \leq \lambda \ \forall w \in \mathcal{V}. \tag{58}$$

By applying the characterization of the extreme points, we reduce the semi-infinite problem in (55) which has an infinite number of constraints to a problem with finite constraints.

Taking the Lagrangian dual of the above convex problem by introducing dual variables for each of the constraints, we arrive at the $\ell_1$ penalized convex problem

$$\min_z \ \ell(Kz, y) + \lambda \sum_j |z_j|.$$

Here, $K_{ij} = \left( j_0 x_i^T \frac{\times (x_{j_1}, \ldots, x_{j_{d-1}})}{\|\times (x_{j_1}, \ldots, x_{j_{d-1}})\|_1} \right)_+ = \frac{\left( j_0 \mathbf{Vol}(\mathcal{P}(x_j : j \in S)) \right)_+}{\|\times (x_{j_1}, \ldots, x_{j_{d-1}})\|_1}$ using the multi-index notation $j = (j_0, j_1, \ldots, j_{d-1})$ where $j_0$ ranges over $\{-1, +1\}$ to represent the choice of sign in the expression $\pm x_i^T \frac{\times (x_{j_1}, \ldots, x_{j_{d-1}})}{\|\times (x_{j_1}, \ldots, x_{j_{d-1}})\|_1}$. Rescaling the dual variables $z_j$ by $\|\times (x_{j_1}, \ldots, x_{j_{d-1}})\|_1$, we obtain the problem

$$\min_z \ \ell(Kz, y) + \lambda \sum_j w_j |z_j|.$$

where $K_{ij} = \left( j_0 \mathbf{Vol}(\mathcal{P}(x_{j_1}, \ldots, x_{j_{d-1}})) \right)_+$ and $w_j = \|\times (x_{j_1}, \ldots, x_{j_{d-1}})\|_1$.

Finally, in order to simplify the notation, we take the index set $(j_1, \ldots, j_{d-1})$ over all subsets of $\{1, \ldots, n\}$ of size $d-1$ and redefine the matrix as $K_{ij} = \mathbf{Vol}_+(\mathcal{P}(x_{j_1}, \ldots, x_{j_{d-1}}))$ and $w_j = \|\times (x_{j_1}, \ldots, x_{j_{d-1}})\|_1$. This follows from the fact that the Euclidean norm of the cross product, and hence the volume of the parallelotope is zero the whenever the subset of vectors are linearly dependent. Moreover, the index $j_0 \in \{-1, +1\}$ is absorbed into the ordering of the vectors $x_{j_1}, \ldots, x_{j_{d-1}}$ in the parallelotope, by noting that the sign of the cross product is flipped whenever the ordering of two vectors is flipped. This completes the proof of Theorem 4.

Next, we provide the proofs of the lemmas used in the above proof.

*Proof of Lemma 19.* We express $w = w^+ - w^-$ where $w^+ \geq 0$ and $w^- \geq 0$ represent the positive and negative part of $w$ respectively. Then, we express the extreme points of the set $\mathcal{C}$ using this lifted representation as follows

$$\mathcal{C}' \triangleq \left\{ \begin{bmatrix} w^+ \\ w^- \end{bmatrix} : (2D_i - I)(Xw^+ - Xw^-) \geq 0, \ w_+, w_- \geq 0, \ 1^T w^+ + 1^T w^- = 1 \right\}.$$

We next argue that the extreme points of $\mathcal{C}$ can be analyzed via the extreme points of $\mathcal{C}'$. In particular, it holds that

$$\max_{w \in \mathcal{C}} x^T w = \max_{w \in \mathcal{C}'} x^T (w^+ - w^-)$$

for all $x \in \mathbb{R}^d$ and a maximizer $w$ to the former maximization problem can be obtained from a maximizer $(w^+, w^-)$ to the latter problem by setting $w^* = w^+ - w^-$. On the other hand, a maximizer $(w^+, w^-)$ to the latter problem can be obtained from a maximizer $w$ to the former problem by setting $w^+ = \max\{w, 0\}$ and $w^- = \max\{-w, 0\}$. Therefore, characterizing the extreme points of $\mathcal{C}$ is equivalent to characterizing the extreme points of $\mathcal{C}'$.

This can be written in matrix notation as follows

$$\underbrace{\begin{bmatrix} (2D_i - I)X & -(2D_i - I)X \\ I & 0 \\ 0 & I \end{bmatrix}}_{M} \begin{bmatrix} w^+ \\ w^- \end{bmatrix} \geq 0 \quad \text{and} \quad 1^T \begin{bmatrix} w^+ \\ w^- \end{bmatrix} = 1.$$

Using Lemma 22, the extreme points of the above set are given by the unique solutions of the linear system

$$\begin{bmatrix} M_S \\ 1^T \end{bmatrix} \begin{bmatrix} w^+ \\ w^- \end{bmatrix} = \begin{bmatrix} 0 \\ 1 \end{bmatrix}$$

where $M_S$ is a submatrix of $M$ such that $\begin{bmatrix} M_S \\ 1^T \end{bmatrix} \in \mathbb{R}$ is full rank. For any extreme point $(w^+, w^-)$ of $\mathcal{C}'$, the vector $w = w^+ - w^-$ is an extreme point $\mathcal{C}$. It can be seen that at least $d$ coordinates of $\begin{bmatrix} w^+ \\ w^- \end{bmatrix}$ are zero since $w = w^+ - w^-$ is the decomposition of $w$ into positive and negative parts. In order to characterize non-zero extreme points, we may assume that not all of the constraints $\begin{bmatrix} w^+ \\ w^- \end{bmatrix} \geq 0$ are active, since otherwise $w^+ = w^- = 0$ implying $w = 0$. We next show that the rows of $M_S$ are linearly independent from the row vector $1^T$ due to the structure of $M$. Specifically, the top block $\begin{bmatrix} (2D_i - I)X & -(2D_i - I)X \end{bmatrix}$ has row span orthogonal to $1^T$, and the bottom block $\begin{bmatrix} I & 0 \\ 0 & I \end{bmatrix}$ consists of canonical basis vectors, for which not all constraints are active (otherwise this implies $w = 0$), which makes the subset of active rows linearly independent from $1^T$.

We conclude that the non-zero extreme points of $\mathcal{C}$ are given by $w = w^+ - w^- \in \mathcal{C}$ such that

$$\begin{bmatrix} M_S \\ 1^T \end{bmatrix} \begin{bmatrix} w^+ \\ w^- \end{bmatrix} = \begin{bmatrix} 0 \\ 1 \end{bmatrix},$$

where $S$ is a subset of $2d - 1$ linearly independent rows of $M$ and the above linear system has a unique solution. Recall that at least $d$ coordinates of $\begin{bmatrix} w^+ \\ w^- \end{bmatrix}$ are zero at any extreme point due to the decomposition $w = w^+ - w^-$ where $w^+ = (w)_+$ and $w^- = (w)_-$ are the positive and negative parts of $w$ respectively. Suppose that the vector $\begin{bmatrix} w^+ \\ w^- \end{bmatrix}$ is a non-zero extreme point and has $d + k$ zero entries for some $k \in \mathbb{Z}_+$, then $2d - 1 - (d + k) = d - 1 - k$ entries of $\begin{bmatrix} (2D_i - I)X & -(2D_i - I)X \end{bmatrix} \begin{bmatrix} w^+ \\ w^- \end{bmatrix}$ are zero. Mapping this decomposition back to the usual representation, this implies that an extreme point $w \in \mathcal{C}$ has $k$ zero entries and the vector $(2D_i - I)Xw$ has $d - 1 - k$ zero entries. Therefore, $d - 1$ of the constraints $\begin{bmatrix} (2D_i - I)X \\ I \end{bmatrix} w \geq 0$ are active at a non-zero extreme point $w \in \mathcal{C}$. We conclude that $w \in \mathcal{C}$ is a non-zero extreme point of $\mathcal{C}$ if and only if

$$\begin{bmatrix} (2D_i - I)X \\ I \end{bmatrix}_R w = 0 \quad \text{and} \quad \|w\|_1 = 1,$$

where $R$ is a subset of $d - 1$ linearly independent rows of the matrix $\begin{bmatrix} (2D_i - I)X \\ I \end{bmatrix}$. Finally, we observe that the constraint $\begin{bmatrix} (2D_i - I)X \\ I \end{bmatrix}_R w = 0$ can be substituted by $\begin{bmatrix} X \\ I \end{bmatrix}_R w = 0$ since $(2D_i - I)$ is a diagonal matrix containing $\pm 1$ values on the diagonal, and note that linear independence is invariant to this diagonal sign multiplication. $\qquad\square$

### 9.3 Proof of Theorem 17

We now present the proof of Theorem 17 which shows the approximate equivalence of the $\ell_2$-regularized neural network and the Lasso problem in (48). We consider the dual problem in (5) and analyze its constraints, which are as follows:

$$\max_{i \in [P]} \max_w |v^T D_i X w| \leq \lambda \tag{59}$$

$$\text{s.t. } \|w\|_2 \leq 1, \ (2D_i - I)Xw \geq 0.$$

Here

$$\{D_i\}_{i=1}^P = \Big\{\mathrm{diag}(1[Xw \geq 0]) : w \in \mathbb{R}^d\Big\},$$

are the diagonal hyperplane arrangement patterns.

As in Section 9.2, we assume that the training matrix $X$ is full column rank and of size $n$ by $d = r$, without loss of generality.

Consider the sub-problem arising in the above constraint given by

$$d_{sub}^+(i) \triangleq \max_w v^T D_i X w$$
$$\text{s.t. } \|w\|_2 \leq 1, (2D_i - I)Xw \geq 0.$$

We define the set

$$\mathcal{C}_2(i) \triangleq \Big\{w : \|w\|_2 \leq 1, (2D_i - I)Xw \geq 0\Big\},$$

and let $d_{sub}^-(i) \triangleq \max_{w \in \mathcal{C}_2(i)} -v^T D_i X w$. Note that the constraint in (5) is precisely $d_{sub}^+(i) \leq \lambda$, $d_{sub}^-(i) \leq \lambda \ \forall i$. Our strategy is to show that $d_{sub}^+(i)$ and $d_{sub}^-(i)$ can be tightly approximated by a polyhedral approximation constructed using the extreme rays of the cone $\{w : (2D_i - I)Xw \geq 0\}$. Consequently, we will be able to obtain an approximation of the dual problem since the dual constraint is

$$d_{const}^* = \max_{i \in \{1,...,P\}} \max_{w \in \mathcal{C}_2} |v^T D_i X w| = \max \Big(\max_{i \in \{1,...,P\}} d_{sub}^+(i), \max_{i \in \{1,...,P\}} d_{sub}^-(i)\Big). \tag{60}$$

We first focus on a generic instance of the problem $d_{sub}^+ = d_{sub}^+(i)$ and $\mathcal{C}_2 = \mathcal{C}_2(i)$ for a certain value of $i$. Suppose that the extreme rays of the convex polyhedral cone

$$\{w : (2D_i - I)Xw \geq 0\} \subseteq \mathbb{R}^d$$

are $R_1, ..., R_k \subseteq \mathbb{R}^d$ for some $k \in \mathbb{Z}_+$. Note that by our assumption that $X$ is full column rank, this cone is pointed since it does not contain any nontrivial linear subspace. Let $p_1, \ldots, p_k \in \mathbb{S}^{d-1}$ denote the unit norm generators of the extreme rays $R_1, ..., R_k$ normalized to unit Euclidean norm. We define the convex sets

$$\mathcal{P} \triangleq \mathbf{Conv}(p_1, ..., p_k) \quad \text{and} \quad \mathcal{P}_0 \triangleq \mathbf{Conv}(0, p_1, ..., p_k).$$

We will use the convex set $\mathcal{P}_0$ as an inner polyhedral approximation of $\mathcal{C}_2$. We define the distance of the convex set $\mathcal{P}$ from the origin as

$$d_{\min} \triangleq \mathbf{dist}(0, \mathcal{P}) = \min_{w \in \mathcal{P}} \|w\|_2.$$

We have $d_{\min} \leq 1$ since $\|\sum_i \alpha_i p_i\|_2 \leq \sum_i |\alpha_i| \|p_i\|_2 = \sum_i |\alpha_i| = 1$ for any $\alpha_i \geq 0$ and $\sum_i \alpha_i = 1$, noting that $\|p_i\|_2 = 1$.

We apply Lemma 20 to obtain the following lower and upper bounds on $d_{sub}^+$:

$$\max_{w \in \mathcal{P}_0} v^T D_k X w \leq \max_{w \in \mathcal{C}_2} v^T D_k X w \leq \max_{w \in d_{\min}^{-1} \mathcal{P}_0} v^T D_k X w \tag{61}$$
$$= d_{\min}^{-1} \max_{w \in \mathcal{P}_0} v^T D_k X w.$$

Next, we simplify the polyhedral approximation $\max_{w \in \mathcal{P}_0} v^T D_k X w$. Applying Weyl's Facet Lemma (see Lemma 24) to the cone $K = \{w : (2D_i - I)Xw \geq 0\}$, we see that a vector $v \in K$ belongs to an extreme ray of $K$ if and only if there are $d - 1$ linearly independent rows of the matrix $(2D_i - I)X$ orthogonal to $v$. Then, each of the generators of the extreme rays $p_1, ..., p_k$ satisfy $p_i^T x_j = 0$, $j \in S_i$ where $S_i$ is a subset of $d - 1$ linearly independent rows of the matrix $X$. Using the properties of the generalized cross product (see 6.1), we identify each generator as a scalar multiple of the cross product $\times_{j \in S_i} x_j$ of the $d - 1$ linearly

independent rows $x_j$ of $X$. Note that the cross product of $d-1$ linearly independent vectors is orthogonal to each vector and lies on a one-dimensional linear subspace. Therefore the unit-norm vectors $p_i$ can be identified as $p_i = \pm \frac{\times_{j \in S_i} x_j}{\|\times_{j \in S_i} x_j\|_2}$, where the sign is chosen so that $p_i \in K$, i.e., $p_i^T D_{k,jj} x_j \geq 0$ for all $j \in S_i$.

Next, note that Weyl's Facet Lemma also guarantees that any vector $p \in K$ of the form $p = \frac{\times_{j \in S} x_j}{\|\times_{j \in S} x_j\|_2}$ is on an extreme ray of $K$ when $S$ is a subset of $d-1$ linearly independent rows of $X$. When $d \geq 2$, the collection of vectors $0 \cup \{\pm \times_{j \in S} x_j : S \subseteq [n], \dim \mathbf{Span}\{x_j : j \in S\} = d-1\}$ is equivalent to $0 \cup \{\times_{j \in S} x_j : S \subseteq P_{d-1}([n])\}$, where $P_{d-1}([n])$ denotes all permutations of subsets of $[n]$ with $d-1$ elements. In order to see this equivalence, observe that the $\pm 1$ multiplier can be removed when we consider permutations since exchanging the order of the two vectors in the cross product changes the sign of the resulting vector. Moreover, we can only consider subsets $S$ of size $d-1$, otherwise the vectors are linearly dependent and the cross product is zero. In addition, when the vectors are linearly dependent, the cross product is zero, which is already included in the collection of vectors.

Using this characterization, we can simplify the first optimization problem in (61) by enumerating all unit-norm generators of the extreme rays of the cone $K$, which can be done by considering all subsets $S$ of $d-1$ linearly independent rows of $X$ and taking the cross product of the rows in each subset. Define the set $\mathcal{D}_S$ as follows

$$\mathcal{D}_S = \left\{ \times_{j \in S} x_j \right\} \cup 0.$$

Then, we have

$$\max_{w \in \mathcal{P}_0} v^T D_k X w = \max_{w \in \{0, p_1, \dots, p_k\}} v^T D_k X w = \max_{S \subseteq P_{d-1}([n])} \max_{w \in K \cap \mathcal{D}_S} v^T \left( \frac{Xw}{\|w\|_2} \right)_+, \tag{62}$$

where the last maximization is over $d-1$ permutations $P_{d-1}([n])$ of subsets of $[n]$, and $w \in K \cap \mathcal{D}_S$. The justification of (62) is as follows: In the first equality in (62), we can drop the convex hull of $\{0, p_1, \dots, p_k\}$ since the objective is linear. In the second equality, we replace the unit norm generators by their expressions given by the cross product and use the fact that $D_i X w = (Xw)_+$ for $w \in K$. Note that the cross product is zero when the vectors are linearly dependent to simplify the expression of $\mathcal{D}_S$ in the last equality above.

Next, we plug-in the expression for $d_{sub}^+$ given in the last equality of (62) for $d_{sub}^+(i)$ in (60), repeat the same argument for $d_{sub}^-(i)$ which is identical, and use the approximation bound in (61)

$$d_{\text{const}}^* = \max_{i \in \{1, \dots, P\}, \ s \in \{+1, -1\}} \max_{w \in \mathcal{C}_2(i)} s v^T D_i X w \tag{63}$$

$$\leq d_{\min}^{-1} \max_{i \in \{1, \dots, P\}, \ s \in \{+1, -1\}} \max_{S \subseteq P_{d-1}([n])} \max_{w \in K(i) \cap \mathcal{D}_S} s v^T \left( X \frac{w}{\|w\|_2} \right)_+ \tag{64}$$

where $K(i) := \{w : (2D_i - I)Xw \geq 0\}$. Noting that

$$\cup_{i=1}^P K(i) = \cup_{i=1}^P \{w : (2D_i - I)Xw \geq 0\} = \mathbb{R}^d,$$

since the union of the chambers of the arrangement is the entire $d$-dimensional space, maximizing over the index $i \in \{1, \dots, P\}$ and $w \in K(i) \cap \mathcal{D}_S$ is equivalent to maximizing $w$ over $\cup_{i=1}^P K(i) \cap \mathcal{D}_S = \mathcal{D}_S$. Therefore, we obtain

$$\hat{d} \leq d_{\text{const}}^* \leq d_{\min}^{-1} \hat{d}, \tag{65}$$

where

$$\hat{d} \triangleq \max_{S \subseteq P_{d-1}([n])} \max_{w \in \mathcal{D}_S} \left| v^T \left( X \frac{w}{\|w\|_2} \right)_+ \right|. \tag{66}$$

Using the expression of the dual in (5), we have

$$p^* = \max_{d_{\text{const}}^*(v) \leq \lambda} -\ell^*(v) \tag{67}$$

where $d_{\text{const}}^* = d_{\text{const}}^*(v)$ is as defined in (63). We then use the inequalities in (65) to get the lower and upper-bounds on the optimal objective as follows

$$\max_{d_{\min}^{-1}\hat{d}(v)\leq\lambda} -\ell^*(v) \leq p^* \leq \max_{\hat{d}(v)\leq\lambda} -\ell^*(v)\,, \tag{68}$$

where $\hat{d}(v) = \hat{d}$ is as defined in (66). Replacing the duals of the maximization problems in the above equation using Lemma 25, we obtain

$$\min_z \ell(\tilde{K}z, y) + d_{\min}\lambda\|z\|_1 \leq p^* \leq \hat{p}_\lambda := \min_z \ell(\tilde{K}z, y) + \lambda\|z\|_1\,. \tag{69}$$

Here, we defined $\hat{p}_\lambda$ as the optimal value of the Lasso objective and $\tilde{K}$ is the matrix defined as follows

$$K_{i,j} = x_i^T \frac{\times(x_1, ..., x_{d-1})}{\|\times(x_1, ..., x_{d-1})\|_2}\,, \tag{70}$$

where $j = (j_1, ..., j_d)$ is a multi-index. The above shows that the optimal value $p^*$ corresponding to the NN objective in (1) when $p = 2$ is bounded between the convex Lasso program in the right-hand side of (69) and the same convex program with a slightly smaller regularization coefficient as follows

$$\hat{p}_{d_{\min}\lambda} \leq p^* \leq \hat{p}_\lambda.$$

Noting that

$$\ell(\tilde{K}\hat{z}_{d_{\min}\lambda}, y) + d_{\min}\lambda\|\hat{z}_{d_{\min}\lambda}\|_1 \geq \min_z \ell(\tilde{K}z, y) + \lambda\|z\|_1 + (d_{\min} - 1)\lambda\|\hat{z}_{d_{\min}\lambda}\|_1$$
$$= \hat{p}_\lambda - (1 - d_{\min})\lambda\|\hat{z}_{d_{\min}\lambda}\|_1\,,$$

where $\hat{z}_{d_{\min}\lambda}$ is a minimizer of the Lasso problem corresponding to $\hat{p}_{d_{\min}\lambda}$. we get the inequalities

$$\hat{p}_\lambda - (1 - d_{\min})\|\hat{z}_{d_{\min}\lambda}\|_1 \leq \hat{p}_{d_{\min}\lambda} \leq p^* \leq \hat{p}_\lambda\,. \tag{71}$$

This shows that the optimality gap is at most $(1 - d_{\min})\lambda\|\hat{z}_{d_{\min}\lambda}\|_1$, where $\hat{z}_{d_{\min}\lambda}$ is an optimal solution of the Lasso problem in the left-hand side of (69) with regularization coefficient $d_{\min}\lambda$.

We now obtain upper and lower-bounds on $\hat{p}_\lambda$ in terms of $p^*$ by noting that (65) implies

$$d_{\text{const}}^* \leq d_{\min}^{-1}\hat{d} \leq d_{\min}^{-1}d_{\text{const}}^*.$$

Multiplying both sides by $d_{\min}$ we obtain

$$d_{\min}d_{\text{const}}^* \leq \hat{d} \leq d_{\text{const}}^*.$$

Using the above inequalities in the dual programs, we get

$$\max_{d^*(v)\leq\lambda} -\ell^*(v) \leq \max_{\hat{d}(v)\leq\lambda} -\ell^*(v) \leq \max_{d_{\min}d_{\text{const}}^*(v)\leq\lambda} -\ell^*(v). \tag{72}$$

Using the dual of the Lasso program again from Lemma (5), we obtain

$$p^* \leq \min_z \ell(\tilde{K}z, y) + \lambda\|z\|_1 \leq p_{d_{\min}^{-1}\lambda}^*\,, \tag{73}$$

where $p_{\lambda'}^*$ in the right-hand side is defined as

$$p_{\lambda'}^* \triangleq \min_{W_1, W_2} \ell\Big(\sum_{j=1}^m \sigma(XW_{1j} + 1b_{1j})W_{2j}, y\Big) + \lambda'\sum_{j=1}^m \|W_{1j}\|_2^2 + \|W_{2j}\|_2^2, \tag{74}$$

for $m \geq m^*$ where $m^*$ is the number of neurons in an optimal solution of the bidual of (55) when $p = 2$. Here, $p^*_{d^{-1}_{\min}\lambda}$ is the value of the optimal NN objective when the regularization coefficient is set to to the larger value $d^{-1}_{\min}\lambda$. This value can be upper-bounded as follows

$$p^*_{d^{-1}_{\min}\lambda} \leq F(W^*_1, W^*_2) + d^{-1}_{\min}\lambda R(W^*_1, W^*_2) \tag{75}$$

$$= F(W^*_1, W^*_2) + \lambda R(W^*_1, W^*_2) + (d^{-1}_{\min} - 1)\lambda R(W^*_1, W^*_2) \tag{76}$$

$$= p^*_\lambda + (d^{-1}_{\min} - 1)\lambda R(W^*_1, W^*_2), \tag{77}$$

where $F(\cdot, \cdot)$ and $R(\cdot, \cdot)$ are the functions in the first (loss) and second (regularization) terms in the objective (74) respectively, and $(W^*_1, W^*_2)$ is an optimal solution of (74), i.e., we have $p^*_\lambda = F(W^*_1, W^*_2) + \lambda R(W^*_1, W^*_2)$. Therefore, (73) implies that

$$p^* \leq \min_z \ell(\tilde{K}z, y) + \lambda\|z\|_1 \leq p^* + (d^{-1}_{\min} - 1)\lambda R(W^*_1, W^*_2) \leq d^{-1}_{\min}p^*, \tag{78}$$

where $R(W^*_1, W^*_2) := \sum_{j=1}^m \|W^*_{1j}\|_2^2 + \|W^*_{2j}\|_2^2$ is the regularization term corresponding to any optimal solution of (74). The final inequality follows from $\lambda R(W^*_1, W^*_2) \leq p^*$. This shows that the optimality gap of the Lasso solution is at most $\frac{1-d_{\min}}{d_{\min}}\lambda R(W^*_1, W^*_2)$.

Finally, to simplify the expression of the matrix $\tilde{K}$ given in (70), we use the following cross product representation of the distance to an affine hull (see Supplementary Material)

$$x^T \frac{\times(x_1, ..., x_{d-1})}{\|\times(x_1, ..., x_{d-1})\|_2} = \mathbf{dist}(x, \mathbf{Aff}(0, x_1, ..., x_{d-1}))$$

$$= \mathbf{dist}(x, \mathbf{Span}(x_1, ..., x_{d-1})),$$

for any $x, x_1, ..., x_{d-1} \in \mathbb{R}^d$ where $\times(x_1, ..., x_{d-1})$ is the cross product of the vectors $x_1, ..., x_{d-1}$. Combining the bounds (71) and (78) with the lower bound on the minimum distance $d_{\min}$ in terms of the maximum chamber diameter $r$ from Lemma 21 completes the proof.

## 9.4 Proof of Theorem 14

The proof parallels the proof of Theorems 4 and 17. Here, we only highlight the main differences.

The convex dual of the problem (40) is given by

$$p^*_v \geq d^*_v \triangleq \max_{V \in \mathbb{R}^n} -\ell^*(V, Y) \quad \text{s.t.} \quad \|V^T\sigma(Xw)\|_2 \leq \lambda, \forall w \in \mathcal{B}_p^d. \tag{79}$$

This dual convex program was derived in Sahiner et al. (2020), where it was also shown that strong duality holds if the number of neurons, $m$, exceed the critical threshold $m^*$, which is the number of neurons in the convex bidual program.

We focus on the dual constraint subproblem in (79) given by

$$d_{\text{v-sub}} \triangleq \max_w \|V^T\mathrm{D}_kXw\|_2$$

$$\text{s.t.} \quad \|w\|_p \leq 1, (2\mathrm{D}_k - \mathrm{I})Xw \geq 0.$$

$$= \max_w \max_{u:\ \|u\|_2 \leq 1} |u^TV^T\mathrm{D}_kXw|$$

$$\text{s.t.} \quad \|w\|_p \leq 1, (2\mathrm{D}_k - \mathrm{I})Xw \geq 0.$$

We note that the above problem has the same form of equation (56) and (59) for $p = 1$ and $p = 2$ respectively, where the vector $v$ plays the role of $Vu$. The rest of the proof is identical for both cases. □

**Lemma 20** (Archimedean approximation of spherical sections via polytopes). *Suppose that* $\mathbf{Cone}(P) \cap -\mathbf{Cone}(P) = \{0\}$, *i.e.,* $\mathbf{Cone}(P)$ *does not contain a non-trivial (non-zero dimensional) linear subspace. Then, it holds that*

$$\mathcal{P}_0 \subseteq \mathcal{C}_2 \subseteq d^{-1}_{\min}\mathcal{P}_0.$$

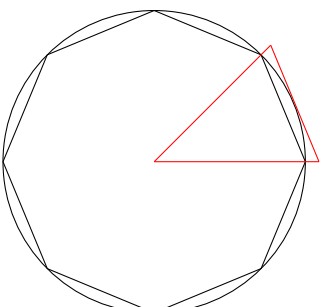

Figure 40: Illustration of the Archimedean approximation in Lemma 20.

*Proof.* The first inclusion follows by noting that $\mathcal{P}_0$ is a polytope and it is contained in $\mathcal{C}_2$ since all of its extreme points $0, p_1, ..., p_k$ are contained in the convex set $\mathcal{C}_2$.

In order to prove the second inclusion, we first show that $\mathcal{C}_2 = \mathbf{Cone}(P) \cap \{\|w\|_2 \le 1\}$. Since $\mathbf{Cone}(P)$ is pointed, i.e., $\mathbf{Cone}(P) \cap -\mathbf{Cone}(P) = \{0\}$, it is equal to the convex hull of its extreme rays (Theorem 13 of Fenchel and Blackett (1953)). Since the extreme rays of $\mathbf{Cone}(P)$ and the extreme rays of the cone $\{w : (2D_i - I)Xw \ge 0\}$, are identical and the latter cone is also pointed, these two cones are identical. Therefore, we have $\mathcal{C}_2 = \mathbf{Cone}(P) \cap \{\|w\|_2 \le 1\}$.

Next, suppose that there exists some $w^* \in \mathcal{C}_2$. Then, $\exists t \in \mathbb{R}_+, \alpha_i \ge 0, \sum_i \alpha_i = 1$ such that $w^* = t \sum_i \alpha_i p_i$ since $w^* \in \mathbf{Cone}(P)$. Since $\|w^*\|_2 \le 1$, we have $1 \ge \|w^*\|_2 = \|t \sum_i \alpha_i p_i\|_2 \ge t \min_{\alpha_i} \|\sum_i \alpha_i p_i\|_2 \ge t \min_{w \in \mathcal{P}} \|w\|_2 = t\, d_{\min}$. From this chain of inequalities, we obtain the upper-bound $t \le d_{\min}^{-1}$. We have $w^* \in d_{\min}^{-1} \mathcal{P}_0$ since

$$w^* = t \sum_i \alpha_i p_i \in t\mathbf{Conv}(p_1, ..., p_k) \in \mathbf{Conv}(0, p_1, ..., p_k) \in d_{\min}^{-1} \mathcal{P}_0.$$

$\square$

**Lemma 21** (Diameter of a chamber bounds its distance to the origin)**.** *Consider the set $\tilde{\mathcal{C}}_2 : \{w : (2D_i - I)Xw \ge 0, \|w\|_2 = 1\}$. Suppose that $r \in \mathbb{R}$ is the $\ell_2$-diameter of $\tilde{\mathcal{C}}_2$, which is defined as the smallest $r \ge 0$ such that $\|w_1 - w_2\|_2 \le r$ for all $w_1, w_2 \in \tilde{\mathcal{C}}_2$. Then,*

$$d_{\min} := \min_{v \in \mathbf{Conv}(v_1, ..., v_k)} \|v\|_2 \ge 1 - r,$$

*for any set of $k$ vectors $\{v_1, ..., v_k\} \in \tilde{\mathcal{C}}_2$.*

*Proof.* Since the $\ell_2$-diameter of the set $\tilde{\mathcal{C}}_2$ is upper bounded by $r$, we first argue that there exists a Euclidean ball of radius $r$ that contains $\tilde{\mathcal{C}}_2$ as follows. Suppose that $w_1, w_2$ are two vectors in $\tilde{\mathcal{C}}_2$ that achieve the $\ell_2$-diameter $r$, i.e., $\|w_1 - w_2\|_2 = r$. Then, the Euclidean ball of radius $r$ centered at $\frac{w_1 + w_2}{2}$ contains $\tilde{\mathcal{C}}_2$ since for all $w \in \tilde{\mathcal{C}}_2$ it holds that

$$\|w - \frac{w_1 + w_2}{2}\|_2 = \frac{1}{2}\|(w - w_1) + (w - w_2)\|_2 \le \frac{1}{2}\|w - w_1\|_2 + \frac{1}{2}\|w - w_2\| \le r.$$

Next, note that $\mathbf{Conv}(v_1, ..., v_k)$ is contained in this ball due to the convexity of the Euclidean ball and since $v_1, ..., v_k \in \tilde{\mathcal{C}}_2$ are contained in this set. Therefore, the minimum norm point

$$v^* := \arg \min_{v \in \mathbf{Conv}(v_1, ..., v_k)} \|v\|_2,$$

satisfies $\|v^*\| \ge \|v_1\|_2 - \|v^* - v_1\|_2 = 1 - \|v^* - v_1\|_2 \ge 1 - r.$ $\square$

*Proof of Lemma 7.* We use Lemma 2.1 of Plan and Vershynin (2014) by taking $K$ equal to the unit Euclidean sphere in $\mathbb{R}^d$. Since the Gaussian width of this set is bounded by $\sqrt{d}$, Lemma 2.1 implies that

$$\left| \frac{1}{m} \|Xz\|_1 - \alpha \right| \leq \varepsilon \quad \forall z \ : \ \|z\|_2 = 1,$$

with probability at least $1 - 2e^{-nu^2/2}$, where $\alpha = \sqrt{2/\pi}$ and $\varepsilon = 4\sqrt{\frac{d}{n}} + u$. Rewriting the above inequality as $-\varepsilon \leq \frac{1}{m}\|X\frac{w}{\|w\|_2}\|_1 - \alpha \leq \varepsilon \ \forall w$, multiplying each side by $\|w\|_2$ and letting $u = \sqrt{\frac{d}{n}}$, we observe that the $\ell_1$ isometry condition (20) where $\epsilon = 5\sqrt{\frac{d}{n}}$ holds with probability at least $1 - 2e^{-d/2}$. Invoking Lemma 6, we obtain that the maximum chamber radius is bounded by $4\sqrt{\epsilon} = 4\sqrt{5\sqrt{\frac{d}{n}}} \leq 9(\frac{d}{n})^{1/4}$, which concludes the proof. $\square$

*Proof of Lemma 6.* Suppose that $w_1, w_2 \in \tilde{\mathcal{C}}_2$. Then, $w_a := \frac{1}{2}(w_1 + w_2)$ satisfies $(2D_i - I)Xw_a \geq 0$ since $w_1, w_2$ belongs to the set $(2D_i - I)Xw \geq 0$, which is convex. Therefore, we have

$$\begin{aligned}
|x_i^T w_a| &= \frac{1}{2}|x_i^T w_1 + x_i^T w_2| \\
&= \frac{1}{2}\text{sign}(x_i^T w_1 + x_i^T w_2)(x_i^T w_1 + x_i^T w_2) \\
&= \frac{1}{2}\left(\text{sign}(x_i^T w_1)(x_i^T w_1) + \text{sign}(x_i^T w_2)(x_i^T w_2)\right) \\
&= \frac{1}{2}|x_i^T w_1| + \frac{1}{2}|x_i^T w_2|.
\end{aligned}$$

where the third equality follows since $\text{sign}(x_i^T w_1) = \text{sign}(x_i^T w_2) \ \forall i$ due to the constraints $(2D_i - I)Xw_1 \geq 0$ and $(2D_i - I)Xw_2 \geq 0$. Note that since $\|Xw\|_1 = \sum_{i=1}^n |x_i^T w|$, this implies the identity

$$\|Xw_a\|_1 = \frac{1}{2}\|Xw_1\|_1 + \frac{1}{2}\|Xw_2\|_1.$$

Applying the isometry condition given in (20) to $w_a$ and using the above identity, we get

$$\begin{aligned}
\alpha(1+\epsilon)\|w_a\|_2 &\geq \frac{1}{n}\|Xw_a\|_1 \\
&= \frac{1}{2n}\|Xw_1\|_1 + \frac{1}{2n}\|Xw_2\|_1 \\
&\geq \frac{1}{2}\alpha(1-\epsilon)\|w_1\|_2 + \frac{1}{2}\alpha(1-\epsilon)\|w_2\|_2 \\
&= \alpha(1-\epsilon),
\end{aligned} \tag{80}$$

where the inequality (80) follows from the concentration inequality (20) applied to $w_1$ and $w_2$. Therefore, we have $\|w_a\|_2 \geq (1+\epsilon)^{-1}(1-\epsilon)$ and

$$\|w_a\|_2^2 = \left\|\frac{1}{2}(w_1 + w_2)\right\|_2^2 = \frac{1}{2} + \frac{w_1^T w_2}{2} \geq \left(\frac{1-\epsilon}{1+\epsilon}\right)^2 \geq 1 - 4\epsilon.$$

The last inequality holds since

$$(1 - \epsilon^2) = 1 - 2\epsilon + \epsilon^2 \geq 1 - 3\epsilon - 4\epsilon^2 = (1+\epsilon)(1-4\epsilon),$$

for $\epsilon \geq 0$. This implies $w_1^T w_2 \geq 1 - 8\epsilon$. Since $w_1$ and $w_2$ are unit Euclidean norm, we get $\|w_1 - w_2\|_2^2 = 2 - 2w_1^T w_2 \leq 16\epsilon$. Therefore $\|w_1 - w_2\|_2 \leq 4\sqrt{\epsilon}$.

$\square$

### 9.5 Proofs for two-dimensional networks

*Proof of Theorem 15.* The proof follows by specializing the proof of Theorem 17 to $d = 2$ and noting that the chamber diameter is controlled by the angle $\epsilon\pi$. Consider the triangle corresponding the the chamber where the angular dispersion bound is achieved. Consider the bisector of this angle and observe that the chamber diameter is equal to $2\sin(\epsilon\pi/2)$, which is upper-bounded by $\epsilon\pi$. Although this provides the approximation factor $1/(1 - \epsilon\pi)$ as long as $\epsilon < \pi^{-1}$, we can improve this bound by considering a direct Archimedean approximation of the circular section as shown in Figure 40 to replace Lemma (20). Specifically, using the bisector of the angle we also have

$$\mathcal{P}_0 \subseteq \mathcal{C}_2 \subseteq (\cos(\epsilon\pi/2))^{-1}\mathcal{P}_0,$$

which improves Lemma (20). Finally, note that

$$(\cos(\epsilon\pi/2))^{-1} \leq 1 + \epsilon \ ,$$

for $\epsilon \in [0, 1/2]$, which completes the proof. $\qquad\square$

*Proof sketch of Theorem 3.* The proof follows the same steps as in Theorem 17 and 5. $\qquad\square$

### 9.6 Proofs for deep neural networks

*Proof of Theorem 10.* Suppose that an optimal solution of (24) is given by

$$\theta^* = \left(W^{*(1)}, \cdots, W^{*(L)}\right).$$

Define optimal activations of the first two-layer block based on the optimal $W^{*(1)}, W^{*(2)}$ above as the matrix $Y^{*(1)}$ by defining

$$Y^{*(1)} := \sigma(XW^{*(1)})W^{*(2)}.$$

Now, we focus on the first two-layer NN block and consider the following auxiliary problem

$$(\hat{W}^{(1)}, \hat{W}^{(2)}) = \arg\min_{W_1^{(1)}, W_2^{(2)}} \sum_{j=1}^{m} \|W_{:j}^{(1)}\|_p^2 + \|W_{j:}^{(2)}\|_p^2 \tag{81}$$

$$\text{s.t.} \qquad \sigma(XW^{(1)})W^{(2)} = Y^{*(1)}.$$

The above problem is in the form of a two-layer neural network optimization problem. We now show that $\hat{W}^{(1)}, \hat{W}^{(2)}$ combined with the rest of the optimal weights $W^{*(3)}, W^{*(4)} \cdots, W^{*(L)}$ are also optimal in the optimization problem (24). First, note that the outputs of the network, $f_{\theta^*}(X)$ and $f_{\hat{\theta}_1}(X)$, are identical where $\theta^*$ is an optimal solution and

$$\hat{\theta}_1 := \left(\hat{W}^{(1)}, \hat{W}^{(2)}, W^{*(3)}, W^{*(4)} \cdots, W^{*(L)}\right),$$

since the activations of the first two-layer block are identical. Second, note that the regularization terms $R_\theta^{(1)} := \sum_{j=1}^{m} \|W_{:j}^{(1)}\|_p^2 + \|W_{j:}^{(2)}\|_p^2$ are also equal since $R_{\hat{\theta}_1}^{(1)} < R_{\theta^*}^{(1)}$ contradicts the optimality of $\theta^*$ in (24) and $R_{\hat{\theta}_1}^{(1)} > R_{\theta^*}^{(1)}$ contradicts the optimality of $(\hat{W}^{(1)}, \hat{W}^{(2)})$ in (81).

Next, we analyze the remaining layers. We define the optimal activations before and after the $\ell$-th two-layer NN block as follows

$$X^{*\ell} = \sigma(\cdots\sigma(XW^{*(1)})W^{*(2)} \cdots W^{*(\ell-1)})$$

$$Y^{*\ell} = \sigma(\cdots\sigma(XW^{*(1)})W^{*(2)} \cdots W^{*(\ell-1)})W^{*(\ell+1)}.$$

Consider the auxiliary problem for the $\ell$-th two-layer block

$$(\hat{W}^{(\ell)}, \hat{W}^{(\ell+1)}) = \arg\min_{W^{(\ell)}, W^{(\ell+1)}} \sum_{j=1}^{m} \|W_{:j}^{(\ell)}\|_p^2 + \|W_{j:}^{(\ell+1)}\|_p^2 \tag{82}$$

$$\text{s.t.} \qquad \sigma(X^{*\ell}W^{(\ell)})W^{(\ell+1)} = Y^{*(\ell)}.$$

We show that $\hat{\theta}_\ell := \left(W^{*(1)}, W^{*(2)}, \cdots \hat{W}^{(\ell)}, \hat{W}^{(\ell+1)} \cdots, W^{*(L-1)}, W^{*(L)}\right)$, which equals $\theta^*$ optimal in (24) except the $\ell$-th and $(\ell+1)$-th layer weights are replaced with an optimal solution of (82), is an optimal solution of (24). As in the above case, the outputs of the network, $f_{\theta^*}(X)$ and $f_{\hat{\theta}_\ell}(X)$, are identical by construction. Moreover, the regularization terms $R_\theta^{(\ell)} := \sum_{j=1}^m \|W_{\cdot j}^{(\ell)}\|_p^2 + \|W_{j\cdot}^{(\ell+1)}\|_p^2$ are also equal since $R_{\hat{\theta}_\ell}^{(\ell)} < R_{\theta^*}^{(\ell)}$ contradicts the optimality of $\theta^*$ in (24) and $R_{\hat{\theta}_\ell}^{(\ell)} > R_{\theta^*}^{(\ell)}$ contradicts the optimality of $(\hat{W}^{(\ell)}, \hat{W}^{(\ell+1)})$ in (82) since $(\hat{W}^{(\ell)}, \hat{W}^{(\ell+1)})$ is feasible in (82), i.e., $\sigma(X^{*\ell}W^{*(\ell)})W^{*(\ell+1)} = Y^{*(\ell)}$.

Finally, we finish the proof by iteratively applying the above claims to invoke the two-layer result given in Theorem 14. Given an optimal solution $\theta^*$ of (24), we replace the first two-layer NN block with $(\hat{W}^{(1)}, \hat{W}^{(2)})$ from (81) while maintaining global optimality with respect to (24). For $\ell = 2, 3, ..., L$ we repeat this process in pairs of consecutive layers to reach the claimed result. $\qquad\square$

*Proof of Theorem 11.* The proof proceeds similar to the proof of Theorem 10. We first show that given an optimal solution $\theta^*$, we can replace the first two layers $(W^{*(1)}, W^{*(2)})$ with $(\hat{W}^{(1)}, \hat{W}^{(2)})$, where

$$(\hat{W}^{(1)}, \hat{W}^{(2)}) = \arg \min_{W_1^{(1)}, W_2^{(2)}} \sum_{j=1}^m \|W_{\cdot j}^{(1)}\|_2^2 + \|W_{j\cdot}^{(2)}\|_2^2 \tag{83}$$
$$\text{s.t.} \qquad \sigma(XW^{(1)})W^{(2)} = Y^{*(1)},$$

and $Y^{*(1)} := \sigma(XW^{*(1)})W^{*(2)}$. By Corollary 18, the weights $(\hat{W}^{(1)}, \hat{W}^{(2)})$ only increase the regularization $\sum_{j=1}^m \|W_{\cdot j}^{(1)}\|_2^2 + \|W_{j\cdot}^{(2)}\|_2^2$ by a factor of $1/(1 - \mathscr{D}(X))$, while producing the identical network output $f_{\hat{\theta}}(X) = f_{\theta^*}(X)$.

Next, we analyze the remaining layers one by one. We define the optimal activations before and after the $\ell$-th two-layer NN block as follows

$$X^{*\ell} = \sigma(\cdots \sigma(XW^{*(1)})W^{*(2)} \cdots W^{*(\ell-1)})$$
$$Y^{*\ell} = \sigma(\cdots \sigma(XW^{*(1)})W^{*(2)} \cdots W^{*(\ell-1)})W^{*(\ell+1)}.$$

Consider the auxiliary problem for the $\ell$-th two-layer block

$$(\hat{W}^{(\ell)}, \hat{W}^{(\ell+1)}) = \arg \min_{W^{(\ell)}, W^{(\ell+1)}} \sum_{j=1}^m \|W_{\cdot j}^{(\ell)}\|_2^2 + \|W_{j\cdot}^{(\ell+1)}\|_2^2 \tag{84}$$
$$\text{s.t.} \qquad \sigma(X^{*\ell}W^{(\ell)})W^{(\ell+1)} = Y^{*(\ell)}.$$

By Theorem 14, replacing the weights $\hat{W}^{(\ell)}, \hat{W}^{(\ell+1)}$ only increases the regularization term for those weights by a factor of $1/(1 - \mathscr{D}(X^{*\ell}))$ while maintaining the network output $f_{\hat{\theta}}(X) = f_{\theta^*}(X)$. We can repeat this process for all $\ell = 2, 3, ..., L$ one by one, i.e., replacing layers 1 and 2, 2 and 3,.... (as opposed to pairs of consecutive weights as in the proof of Theorem 10), we replace all the weights by increasing the total regularization term by a factor of at most $2/(1-\epsilon)$ assuming $\mathscr{D}(X) \leq \epsilon$ and $\mathscr{D}(X^{*\ell}) \leq \epsilon$ for $\ell \in \{1, ..., L-2\}$. The factor 2 is needed to account for the overlapping weights in the replacement process. $\qquad\square$

### 9.7 Proof of Theorem 16 and Theorem 5

The proof closely parallels the proofs of Theorems 17 and 4. The dual problem in (5) when an additional bias term is present takes the form

$$p^* \geq d^* \triangleq \max_{v \in \mathbb{R}^n} -\ell^*(v, y) \quad \text{s.t.} \quad |v^T \sigma(Xw + 1b)| \leq \lambda, \forall w \in \mathcal{B}_p^d, b \in \mathbb{R}. \tag{85}$$

As in the proof of Theorem 1, we focus on the dual constraint subproblem

$$\sup_{w \in \mathcal{B}_p^d, b \in \mathbb{R}} |v^T \sigma(Xw + 1b)| \leq \lambda \tag{86}$$

and observe that the above objective is piecewise linear in $b$. Moreover, the objective tends to infinity as $b \to \infty$ as long as $\sum_i v_i \neq 0$. Otherwise, an optimal choice for $b$ is $-x_j^T w^*$ for some index $j \in [n]$ for some optimal $w^*$.

Therefore, the constraint (86) can be rewritten as two constraints:

$$\max_{j \in [n]} \sup_{w \in \mathcal{B}_p^d} |\sum_i v_i \sigma((x_i - x_j)^T w^*)| \leq \lambda \quad \text{and} \quad \sum_{i=1}^n v_i = 0. \tag{87}$$

Now, we define $\tilde{X} := X - 1_n x_j^T$ and observe that the form of the dual constraint is identical to the ones analyzed in Theorems 17 and 4 for each fixed $j \in [n]$. We repeat the argument for each $j \in [n]$, and use the same steps that to obtain the convex programs and approximation result. In particular, when the chamber diameters $\mathscr{D}(X - 1_n x_j^T) \leq \epsilon$ uniformly for all $j \in [n]$, $\qquad\square$

## 9.8 Proof of Lemma 8

Lemma 7 immediately implies that the random dataset is $\epsilon$-dispersed with high probability. We aim to prove that for any $j \in \{1, \dots, n\}$, the modified dataset $X' = X - 1x_j^T$ satisfies $\mathscr{D}(X') \leq \epsilon$, $\forall j \in [n]$ with high probability. Note that $X - 1x_j^T$ is also a set of i.i.d. Gaussian, except the $j$-th row, which is all-zeros. Using the same derivation in the proof of Lemma 6 and 7, we have

$$(1 - \epsilon')\|w\|_2 \leq \frac{1}{\alpha(n-1)}\|(X - 1x_j^T)w\|_1 \leq (1 + \epsilon')\|w\|_2, \quad \forall w \in \mathbb{R}^d, \tag{88}$$

for $\epsilon' = 5\sqrt{d/(n-1)}$ and some $\alpha$ with probability at least $1 - 2e^{-d/2}$. Taking a union bound over $j \in [n]$, we observe that the above inequality holds simultaneously $\forall j \in [n]$ with high probability at least $1 - 2ne^{-d/2}$. We set $\epsilon' = \epsilon/2$ and obtain that the dataset $X$ is locally $\epsilon$-dispersed with high probability as long as $n \gtrsim \epsilon^{-4}d$.

## 9.9 Additional Lemmas

**Lemma 22** (Extreme points of polytopes restricted to an affine set). *Suppose that $A \in \mathbb{R}^{n \times d}$, $b \in \mathbb{R}^b$, $c \in \mathbb{R}^d$ and $d \in \mathbb{R}$ are given. A vector $x \in \mathbb{R}^d$ is an extreme point of $\mathcal{C} \triangleq \{Ax \leq b, c^T x = d\}$ if and only if there exists a subset $S$ of $n$ linearly independent rows of $A$ such that $x$ is the unique solution of the linear system*

$$\begin{bmatrix} A_S \\ c^T \end{bmatrix} x = \begin{bmatrix} 0 \\ d \end{bmatrix}, \tag{89}$$

*where the matrix $\begin{bmatrix} A_S \\ c^T \end{bmatrix} \in \mathbb{R}^{d \times d}$ is full rank. Here, $A_S \in \mathbb{R}^{d-1 \times d}$ is the submatrix of $A$ obtained by selecting the $d - 1$ rows indexed by $S$.*

*Proof.* We first provide a proof of the first direction (extreme point $\implies$ full rank) by contradiction. Let $x^*$ be an extreme point of $C$. We define $S$ as the subset of inequality constraints active at $x^*$ and $S^c$ as the subset of inequality constraints inactive. More precisely, we have $A_S x^* = 0$ and $A_{S^c} x^* < 0$ for some subset $S$ of $n$ rows of $A$ and its complement $S^c$. Suppose that the matrix $\begin{bmatrix} A_S \\ c^T \end{bmatrix} \in \mathbb{R}^{d \times d}$ is not full rank. Then we may pick $v \neq 0$ in the nullspace of this matrix. Define $x_1 \triangleq x^* + \epsilon v$ and $x_2 \triangleq x^* - \epsilon v$, which are feasible for $\mathcal{C}$ for any sufficiently small $\epsilon > 0$ since $\begin{bmatrix} A_S \\ c^T \end{bmatrix} v = 0$ and $A_{S^c} x^* < 0$. However, this is a contradiction since $\frac{1}{2}x_1 + \frac{1}{2}x_2 = x^*$ and $x_1, x_2 \neq x^*$ implies that $x^*$ is not an extreme point of $\mathcal{C}$.

We next provide a proof of the second direction (full rank $\implies$ extreme point) by contradiction. Suppose that $x^* \in \mathcal{C}$ is a point such that $A_S x^* = 0$ and $A_{S^c} x^* < 0$ for some subset $S$ its complement $S^c$, $c^T x^* = d$, and the matrix $\begin{bmatrix} A_S \\ c^T \end{bmatrix} \in \mathbb{R}^{d \times d}$ is full row rank. It follows that $x^*$ is the unique solution of the linear system

$\begin{bmatrix} A_S \\ c^T \end{bmatrix} x = \begin{bmatrix} 0 \\ d \end{bmatrix}$. We will show that $x^*$ is an extreme point. Suppose that this is not the case and there exists $x_1, x_2 \in \mathcal{C}$ distinct from $x^*$ such that $x^* = \frac{1}{2}x_1 + \frac{1}{2}x_2$. Then, we have $Ax_1 \leq 0$, $Ax_2 \leq 0$ and $\frac{1}{2}A_S x_1 + \frac{1}{2}A_S x_2 = 0$. Note that $A_S x_1 \leq 0$ and $A_S x_2 \leq 0$ together with $\frac{1}{2}A_S x_1 + \frac{1}{2}A_S x_2 = 0$ imply that $A_S x_1 = A_S x_2 = 0$ (In order to see this, suppose that $a_j^T x_1 < 0$ for some row $a_j$ of $A$ such that $j \in S$. Then we have $\frac{1}{2}a_j^T x_1 + \frac{1}{2}a_j^T x_2 < 0$ which is a contradiction.). Finally, noting $c^T x_1 = c^T x_2 = d$ implies that $\begin{bmatrix} A_S \\ c^T \end{bmatrix} x_1 = \begin{bmatrix} 0 \\ b \end{bmatrix}$ and $\begin{bmatrix} A_S \\ c^T \end{bmatrix} x_2 = \begin{bmatrix} 0 \\ b \end{bmatrix}$, which is a contradiction since $x^*$ is the unique solution of this linear system. $\qquad\square$

**Lemma 23** (Rank reduction). *Let us denote the Singular Value Decomposition (SVD) of $X$ as $X = U\Sigma V^T$ in compact form, where $U \in \mathbb{R}^{n \times r}$, $\Sigma \in \mathbb{R}^{r \times r}$ and $V \in \mathbb{R}^{r \times d}$ and let $X = [U\ U^\perp] \begin{bmatrix} \Sigma & 0 \\ 0 & 0 \end{bmatrix} [V\ V^\perp]^T$ denote the full SVD of $X$. Then, the following optimization problems are equivalent*

$$p_2^* = \min_{W^{(1)}, W^{(2)}, b} \ell\Big( \sum_{j=1}^{m} \sigma(XW_j^{(1)})W_j^{(2)}, y \Big) + \lambda \sum_{j=1}^{m} \|W_j^{(1)}\|_2^2 + \|W_j^{(2)}\|_2^2, \tag{90}$$

$$p_2^r = \min_{\tilde{W}^{(1)}, \tilde{W}^{(2)}, b} \ell\Big( \sum_{j=1}^{m} \sigma(U\Sigma\tilde{W}_j^{(1)})\tilde{W}_j^{(2)}, y \Big) + \lambda \sum_{j=1}^{m} \|\tilde{W}_j^{(1)}\|_2^2 + \|\tilde{W}_j^{(2)}\|_2^2, \tag{91}$$

*i.e., $p^* = p^r$. Optimal solutions of (90) and (91) satisfy $(V^\perp)^T W_j^{(1)} = 0$, $V^T W_j^{(1)} = \tilde{W}_j^{(1)}$ and $W_j^{(2)} = \tilde{W}_j^{(2)}$ $\forall j \in [m]$.*

*Proof of Lemma 23.* We define $[\tilde{W}_j^{(1)} \hat{W}_j^{(1)}] \triangleq [V\ V^\perp] W_j^{(1)}$ and plug-in the compact SVD of $X$ in the expression (90)

$$\min_{W^{(1)}, W^{(2)}, b} \ell\Big( \sum_{j=1}^{m} \sigma(U\Sigma\tilde{W}_j^{(1)})W_j^{(2)}, y \Big) + \lambda \sum_{j=1}^{m} \|\tilde{W}_j^{(1)}\|_2^2 + \|\hat{W}_j^{(1)}\|_2^2 + \|W_j^{(2)}\|_2^2, \tag{92}$$

where we have $\|W_j^{(1)}\|_2^2 = \|[V\ V^\perp]W_j^{(1)}\|_2^2 = \|\tilde{W}_j^{(1)}\|_2^2 + \|\hat{W}_j^{(1)}\|_2^2$. It can be seen that $\hat{W}_j^{(1)} = 0$ $\forall j$ and $V^T W_j^{(1)} = \tilde{W}_j^{(1)}$ $\forall j$. $\qquad\square$

### 9.10 Extreme rays of convex cones

In this subsection, we present some definitions and lemmas related to extreme rays of convex cones (Fenchel and Blackett, 1953; Weyl, 1950).

**Definition 5** (Rays). Let $K$ be a convex cone in $\mathbb{R}^d$. A cone $R \subseteq K$ is called a *ray* of $K$ if $R = \{\lambda x : \lambda \geq 0\}$, for some $x \in K$.

**Definition 6** (Extreme rays). Let $K$ be a convex cone in $\mathbb{R}^d$. A ray $R = \{\lambda x : \lambda \geq 0\} \subseteq K$ generated by some vector $x$ is called an *extreme ray* of $K$ if $x$ is not a positive linear combination of two linearly independent vectors of $K$.

**Lemma 24** (Weyl's Facet Lemma (Weyl, 1950)). *Define the convex cone $K = \{w \in \mathbb{R}^d : w^T p_i \leq 0, i \in [k]\} \subseteq \mathbb{R}^d$, where $p_1, ..., p_k \in \mathbb{R}^d$ are a collection of vectors. Then, a non-zero vector $x \in K$ is an element of an extreme ray of $K$ if and only if $\dim \mathbf{Span}(p_i : x^T p_i = 0, i \in [k]) = d - 1$.*

### 9.11 Convex Duality and Lasso

**Lemma 25** (Lagrange Dual of Lasso). *Suppose that $K \in \mathbb{R}^{n \times P}$ is a matrix whose columns are given by the vectors $k_1, ..., k_p \in \mathbb{R}^n$ and $\lambda > 0$ is a regularization parameter. Then, the primal and dual optimization*

*problems for Lasso are given by*

$$\min_{z} \ell(Kz, y) + \lambda\|z\|_1 \quad = \quad \max_{v \ :\ |v^T k_j| \leq \lambda\, \forall j \in [p]} -\ell^*(v) \tag{93}$$

**Lemma 26** (Lagrange Dual of Lasso with a bias term). *Consider the problem in Lemma 25 with an additional bias term. Then, the primal and dual optimization problems for Lasso are given by*

$$\min_{z,\, b} \ell(Kz + 1b, y) + \lambda\|z\|_1 \quad = \quad \max_{\substack{v \ :\ |v^T k_j| \leq \lambda\, \forall j \in [p] \\ \sum_i v_i = 0}} -\ell^*(v) \tag{94}$$

*Proof.* Lemma 25 and 26 are based on a standard application of Lagrangian duality (Boyd and Vandenberghe, 2004). Strong duality holds, i.e., the primal and dual problems have the same value which are both achieved, in both problems since $v = 0$ in the dual problem is strictly feasible for the inequalities due to $\lambda > 0$. □

### 9.12 Optimizing scaling coefficients

Our characterization of the hidden neurons using geometric algebra in Theorem 4 and 17 determine the direction of optimal neurons. The magnitude of the optimal neuron can be analytically determined. In this section, we show that the optimal scaling coefficients are given by the ratio of the norms of the corresponding weights.

Consider the two-layer neural network training problem given below

$$p^* \triangleq \min_{W^{(1)}, W^{(2)}, b} \ell\Big( \sum_{j=1}^{m} \sigma(XW_j^{(1)} + 1b_j)W_j^{(2)}, y \Big) + \lambda \sum_{j=1}^{m} \|W_j^{(1)}\|_p^2 + \|W_j^{(2)}\|_p^2, \tag{95}$$

where the activation function is positively homogeneous, i.e., $\sigma(\alpha x) = \alpha\sigma(x)$ for $\alpha > 0$. Introducing non-negative scaling coefficients $\alpha_1, ..., \alpha_m$ to scale hidden neuron weight and bias, and dividing the corresponding second layer weight by the same scalar we obtain

$$p^* \triangleq \min_{\substack{W^{(1)}, W^{(2)}, b \\ \alpha_1, ..., \alpha_m \geq 0}} \ell\Big( \sum_{j=1}^{m} \sigma(XW_j^{(1)}\alpha_j + 1b_j\alpha_j)W_j^{(2)}\alpha_j^{-1}, y \Big) + \lambda \sum_{j=1}^{m} \|W_j^{(1)}\|_p^2\alpha_j^2 + \|W_j^{(2)}\|_p^2\alpha_j^{-2}, \tag{96}$$

$$= \min_{\substack{W^{(1)}, W^{(2)}, b \\ \alpha_1, ..., \alpha_m \geq 0}} \ell\Big( \sum_{j=1}^{m} \sigma(XW_j^{(1)} + 1b_j)W_j^{(2)}, y \Big) + \lambda \sum_{j=1}^{m} \|W_j^{(1)}\|_p^2\alpha_j^2 + \|W_j^{(2)}\|_p^2\alpha_j^{-2}, \tag{97}$$

where the last equality follows from the fact that $\sigma(\alpha x) = \alpha\sigma(x)$ for $\alpha > 0$. Through simple differentiation, it is straightforward show that the optimal scaling coefficients are given by

$$\alpha_j^* = \Big( \frac{\|W_j^{(2)}\|_p}{\|W_j^{(1)}\|_p} \Big)^{1/2} \quad \text{for } j \in [m]. \tag{98}$$

### 9.13 Proof for three-layer networks with inputs of arbitrary dimension

We consider the scalar output three-layer network

$$f(x) = \sum_{j=1}^{m} ((x^T W_j^{(1)} + b^{(1)})_+ W_j^{(2)} + b^{(2)})_+ W_j^{(3)} + b^{(3)}, \tag{99}$$

where $W_j^{(1)} \in \mathbb{R}^{d \times q}$, $W_j^{(2)} \in \mathbb{R}^{q \times q}$, $W_j^{(3)} \in \mathbb{R}$, $b^{(1)} \in \mathbb{R}^p$, $b^{(2)} \in \mathbb{R}^q$ and $b^{(3)} \in \mathbb{R}$ are trainable weights.

### 9.13.1 Optimal scaling

We now consider optimizing the scaling coefficients for three-layer neural networks.

**Lemma 27.** *Consider the three-layer neural network training problem given below*

$$p^* \triangleq \min_{W^{(1)}, W^{(2)}, W^{(3)}, b^{(1)}, b^{(2)}} \ell\Big(\sum_{j=1}^{m}((XW_j^{(1)} + b^{(1)})_+ W_j^{(2)} + b^{(2)})_+ W_j^{(3)} + b^{(3)}, y\Big) \tag{100}$$

$$+ \frac{\lambda}{3} \sum_{j=1}^{m} \|W_j^{(1)}\|_p^3 + \|W_j^{(2)}\|_p^3 + |W_j^{(3)}|^3. \tag{101}$$

*The above problem is equivalent to*

$$\min_{\|W_j^{(1)}\|_p=1, \|W_j^{(2)}\|_p=1 \,\forall j, W^{(3)}, b^{(1)}, b^{(2)}} \ell\Big(\sum_{j=1}^{m}((XW_j^{(1)} + b^{(1)})_+ W_j^{(2)} + b^{(2)})_+ W_j^{(3)} + b^{(3)}, y\Big) + \lambda \sum_{j=1}^{m} |W_j^{(3)}|. \tag{102}$$

*Proof.* Consider scalar non-negative weights $\alpha_1, \alpha_2, \alpha_3$ satisfying $\alpha_{1j}\alpha_{2j}\alpha_{3j} = 1$ for $j \in [m]$. Applying the scaling $W_j^{(1)} \leftarrow \alpha_{1j} W_j^{(1)}$, $W_j^{(2)} \leftarrow \alpha_{2j} W_j^{(2)}$, and $W_j^{(3)} \leftarrow \alpha_{3j} W_j^{(3)}$ for all $j \in [m]$, we observe that the loss term $\ell(\cdot)$ does not change due to the positive homogeneity of ReLU. Optimizing over these scalars, we obtain

$$p^* \triangleq \min_{\substack{\alpha_{1j},\alpha_{2j},\alpha_{3j} \in \mathbb{R}_+ \\ \alpha_{1j}\alpha_{2j}\alpha_{3j}=1, j\in[m]}} \min_{W^{(k)}, b^{(k)}, k\in[3]} \ell\Big(\sum_{j=1}^{m}((XW_j^{(1)} + b^{(1)})_+ W_j^{(2)} + b^{(2)})_+ W_j^{(3)} + b^{(3)}, y\Big) \tag{103}$$

$$+ \frac{\lambda}{3} \sum_{j=1}^{m} \alpha_{1j}^3 \|W_j^{(1)}\|_p^3 + \alpha_{2j}^3 \|W_j^{(2)}\|_p^3 + \alpha_{3j}^3 |W_j^{(3)}|^3 \tag{104}$$

$$\geq \min_{\substack{\alpha_{1j},\alpha_{2j},\alpha_{3j} \in \mathbb{R}_+ \\ \alpha_{1j}\alpha_{2x}\alpha_{3j}=1, \forall j \in=[m]}} \min_{W^{(k)}, b^{(k)}, k\in[3]} \ell\Big(\sum_{j=1}^{m}((XW_j^{(1)} + b^{(1)})_+ W_j^{(2)} + b^{(2)})_+ W_j^{(3)} + b^{(3)}, y\Big) \tag{105}$$

$$+ \lambda \sum_{j=1}^{m} \Big(\alpha_{1j}^3 \alpha_{2j}^3 \alpha_{3j}^3 \|W_j^{(1)}\|_p^3 \|W_j^{(2)}\|_p^3 |W_j^{(3)}|^3\Big)^{1/3} \tag{106}$$

$$= \min_{W^{(k)}, b^{(k)}, k\in[3]} \ell\Big(\sum_{j=1}^{m}((XW_j^{(1)} + b^{(1)})_+ W_j^{(2)} + b^{(2)})_+ W_j^{(3)} + b^{(3)}, y\Big) \tag{107}$$

$$+ \lambda \sum_{j=1}^{m} \|W_j^{(1)}\|_p \|W_j^{(2)}\|_p |W_j^{(3)}|, \tag{108}$$

where the inequality follows from the AM-GM inequality, which holds with equality when $\alpha_{kj} = \|W_j^{(k)}\|_p^{-1} \prod_{k=1}^{3} \|W_j^{(k)}\|_p$ for all $j \in [m]$, $k \in [3]$. Finally, we scale $W_j^3$ by $\|W_j^{(1)}\|_p^{-1} \|W_j^{(2)}\|_p^{-1}$ for all $j \in [m]$ and obtain the claimed result. $\qquad\square$

Next, we present the convex dual of the problem in Lemma 27.

**Lemma 28.** *The dual of the problem in* (100) *is given by*

$$d^* = \max_{\substack{v \in \mathbb{R}^n \\ 1_n^T v = 0}} -\ell^*(v) \tag{109}$$

$$s.t. \max_{\|W^{(1)}\|_p=1, \|W^{(2)}\|_p=1, b^{(1)}, b^{(2)}} \Big|v^T\big((XW^{(1)} + b^{(1)})_+ W^{(2)} + b^{(2)}\big)_+\Big| \leq \lambda. \tag{110}$$

The proof of this lemma parallels the proof of Proposition 5 in Wang et al. (2022), where it was also shown that strong duality holds, i.e., $p^* = d^*$, as long as the number of neurons, $m$, is sufficiently large.

### 9.13.2 Proofs for three-layer networks without bias terms

*Proof of Theorem 12.* We focus on the maximization subproblem in the constraints of the dual problem in Lemma 28. We use the same steps in the proof of Theorem 4 and 17 to maximize over the layer two and three weights while the first layer weights are fixed, and finally maximize over the first layer weights. This yields dual constraints analogous to a nested version of (87) given by $\{\mathcal{Z}(v) \leq \beta, \ 1_n^T v = 0\}$, where $\mathcal{Z} = \mathcal{Z}(v)$ is given by

$$\mathcal{Z} := \max_{j \in [n]} \max_{\|w\|_1 = 1} \max \left( \left| \sum_{i=1}^{n} \left( (x_i^T w)_+ - (x_j^T w)_+ \right)_+ v_i \right|, \left| \sum_{i=1}^{n} \left( (x_j^T w)_+ - (x_i^T w)_+ \right)_+ v_i \right| \right) \tag{111}$$

$$= \max(\mathcal{Z}^{(-1)}, \mathcal{Z}^{(1)}, \mathcal{X}^{(-1)}, \mathcal{X}^{(1)}). \tag{112}$$

Here,

$$\mathcal{Z}^{(s)} := \max_{j \in [n]} \max_{\|w\|_1 = 1} s \sum_{i=1}^{n} \left( (x_i^T w)_+ - (x_j^T w)_+ \right)_+ v_i \text{ for } s \in \{-1, +1\}. \tag{113}$$

and

$$\mathcal{X}^{(s)} := \max_{j \in [n]} \max_{\|w\|_1 = 1} s \sum_{i=1}^{n} \left( (x_j^T w)_+ - (x_i^T w)_+ \right)_+ v_i \text{ for } s \in \{-1, +1\}. \tag{114}$$

We focus on $\mathcal{Z}^{(1)}$ since the analysis for $\mathcal{Z}^{(-1)}, \mathcal{X}^{(-1)}, \mathcal{X}^{(1)}$ are identical except small changes. Using hyperplane arrangements of the matrix $X$, we can write the above problem as

$$\mathcal{Z}^{(1)} = \max_{j \in [n], k \in [P]} \max_{w \in \mathcal{C}_k, \|w\|_1 = 1} \sum_{i=1}^{n} \left( D_{kii} x_i^T w - D_{kjj} x_j^T w \right)_+ v_i, \tag{115}$$

where $D_1, ..., D_P$ are the $P$ diagonal matrices of the hyperplane arrangement of $X$ and $D_{kii}$ is the $i$-th diagonal element of the matrix $D_k$. Here $\mathcal{C}_k$ is defined as $\mathcal{C}_k := \{w \in \mathbb{R}^d : (2D_k - I)Xw \geq 0\}$. Now note that $D_{kii} = 0$ implies that

$$D_{kii} x_i^T w - D_{kjj} x_j^T w = -(x_j^T w)_+ \leq 0.$$

Therefore, we can write the above problem as

$$\mathcal{Z}^{(1)} = \max_{j \in [n], k \in [P]} \max_{w \in \mathcal{C}_k, \|w\|_1 = 1} \sum_{i=1}^{n} D_{kii} \left( x_i^T w - D_{kjj} x_j^T w \right)_+ v_i, \tag{116}$$

$$= \max_{j \in [n], k \in [P]} \max_{w \in \mathcal{C}_k, \|w\|_1 = 1} v^T D_k \left( Xw - D_{kjj} 1 x_j^T w \right)_+ \tag{117}$$

Now we split the above problem into two subproblems by considering the values of $D_{kj}$.

$$\mathcal{Z}^{(1)} = \max(\mathcal{M}_1, \mathcal{M}_2), \tag{118}$$

where

$$\mathcal{M}_1 = \max_{j \in [n], k \in [P], D_{kj} = 1} \max_{w \in \mathcal{C}_k, \|w\|_1 = 1} v^T D_k \left( (X - 1 x_j^T) w \right)_+ \tag{119}$$

and

$$\mathcal{M}_2 = \max_{j \in [n], k \in [P], D_{kj} = 0} \max_{w \in \mathcal{C}_k, \|w\|_1 = 1} v^T D_k X w. \tag{120}$$

Note that $(D_k X w)_+ = D_k X w$, which we used to simplify the expression for $\mathcal{M}_2$. Now we focus on the subproblem $\mathcal{M}_1$, by considering the hyperplane arrangements of the matrix $X - 1 x_j^T$. Suppose that the

diagonal arrangement patterns of the matrix $X - 1x_j^T$ are given by $H_{j1}, ..., H_{jG}$, and the corresponding convex cones are given by $\mathcal{F}_{j1}, ..., \mathcal{F}_{jG}$. Then, we can write the subproblem $\mathcal{M}_1$ as

$$\mathcal{M}_1 = \max_{j\in[n],k\in[P],D_{kj}=1,\ell\in[G]} \max_{w\in\mathcal{F}_{j\ell}\cap\mathcal{C}_k,\|w\|_1=1} v^T H_{j\ell}(X - 1x_j^T)w. \tag{121}$$

We also rewrite the subproblem $\mathcal{M}_2$ using the same hyperplane arrangements for reasons that will become clear later.

$$\mathcal{M}_2 = \max_{j\in[n],k\in[P],\ell\in[G],D_{kj}=0} \max_{w\in\mathcal{F}_{j\ell}\cap\mathcal{C}_k,\|w\|_1=1} v^T D_k Xw. \tag{122}$$

Using Lemma 19, we can rewrite the subproblems using the extreme points

$$\mathcal{E}_{jk\ell} := \mathbf{Ext}(\mathcal{T}_{jk\ell}) \text{ where } \mathcal{T}_{jk\ell} := \mathcal{F}_{j\ell} \cap \mathcal{C}_k \cap \{w : \|w\|_1 = 1\}, \tag{123}$$

as follows

$$\mathcal{M}_1 = \max_{j\in[n],k\in[P],\ell\in[G],D_{kj}=1} \max_{w\in\mathcal{E}_{jk\ell}} v^T D_k \left(Xw - D_{kjj}1x_j^T w\right)_+, \tag{124}$$

and

$$\mathcal{M}_2 = \max_{j\in[n],k\in[P],\ell\in[G],D_{kj}=0} \max_{w\in\mathcal{E}_{jk\ell}} v^T D_k \left(Xw - D_{kjj}1x_j^T w\right)_+. \tag{125}$$

Then, we can write

$$\mathcal{Z}^{(1)} = \max(\mathcal{M}_1, \mathcal{M}_2) = \max_{j\in[n],k\in[P],\ell\in[G]} \max_{w\in\mathcal{E}_{jk\ell}} v^T D_k \left(Xw - D_{kjj}1x_j^T w\right)_+$$

$$= \max_{w\in\cup_{j\in[n],k\in[P],\ell\in[G]}\mathcal{E}_{jk\ell}} \sum_{i=1}^{n} \left((x_i^T w)_+ - (x_j^T w)_+\right)_+ v_i.$$

We now apply the characterization of extreme points from Lemma 19, which shows that

$$\mathcal{E}_{jk\ell} = \left\{ w \in \mathcal{T}_{jk\ell} \subseteq \mathbb{R}^d \mid \exists S \subseteq [n], |S| = d-1, \ Mw = 0, \ M := \begin{bmatrix} X \\ X - 1x_j^T \\ I \end{bmatrix}_S, \ \mathrm{rank}(M) = d-1 \right\},$$

where $\mathcal{T}_{jk\ell}$ is defined in (123). Now we note that for each fixed $j \in [n]$, the sets $\{F_{jl} \cap C_k\}_{k\in[P],l\in[G]}$ are chambers of hyperplane arrangements that exhaustively partition $\mathbb{R}^d$. Taking a union over all chambers as the tuple $(k, l)$ range over $[P] \times [G]$ gives the entire space $\mathbb{R}^d$. Therefore, we have

$$\mathcal{Z}^{(1)} = \max_{j\in[n], w\in\mathcal{E}'_j\subseteq\mathbb{R}^d} \sum_{i=1}^{n} \left((x_i^T w)_+ - (x_j^T w)_+\right)_+ v_i.$$

where $\mathcal{E}'_j$ is defined as

$$\mathcal{E}'_j = \left\{ w \in \mathbb{R}^d : \exists S \subseteq [n], |S| = d-1, \ Mw = 0, \ M := \begin{bmatrix} X \\ X - 1x_j^T \\ I \end{bmatrix}_S, \ \mathrm{rank}(M) = d-1 \right\}.$$

As in the proof of Theorem 4, we identify the elements of the set $\mathcal{E}'_j$ using the generalized cross product and obtain the following characterization of $\mathcal{Z}$

$$\mathcal{Z} = \max_{j\in[n], w\in\mathcal{W}\subseteq\mathbb{R}^d} \sum_{i=1}^{n} \left((x_i^T w)_+ - (x_j^T w)_+\right)_+ v_i.$$

where $\mathcal{W}_j$ is defined as $\mathcal{W}_j \triangleq \left\{ \pm \frac{\times_{\ell\in S}\tilde{x}_j^{(j)}}{\|\times_{\ell\in S}\tilde{x}_j^{(j)}\|_1} : \{\tilde{x}_j^{(j)}\}_{j\in S} \text{ linearly independent}, |S| = d-1 \right\}$ and the set of vectors $\{\tilde{x}_i^{(j)}\}_{i=1}^{2n+d}$ are the union of $\{x_i\}_{i=1}^{n}$, $\{x_i - x_j\}_{i=1}^{n}$, and $\{e_i\}_{i=1}^{d}$.

The arguments for $\mathcal{Z}^{(-1)}, \mathcal{X}^{(-1)}, \mathcal{X}^{(1)}$ hold in the same way, and we obtain the following characterization of the dual problem. Using the convex duality of Lasso from Lemma 25, we obtain the claimed convex bidual problem, and the optimal solution is given by the extreme points of the set $\mathcal{W}_j$. □

### 9.13.3 Proofs for depth three networks with biases

*Proof of Theorem 13.* Consider the dual constraint subproblem

$$\mathcal{Z}^{(b)} := \max_{j\in[n]} \max_{\|w\|_1=1,\, b\in\mathbb{R}} \left| \sum_{i=1}^{n} \left( (x_i^T w + b)_+ - (x_j^T w + b)_+ \right)_+ v_i \right|. \tag{126}$$

We now focus on the maximization with respect to the scalar $b$ when the vector $w$ and the index $j$ are fixed. Observe that the objective is a piecewise linear function of $b$, when all other variables are fixed. Therefore, it suffices to check the break points and when $\beta \to -\infty$ or $\beta \to \infty$. The break points are when $b$ such that $x_i^T w + b = 0$ for some $i \in [n]$. When $\beta \to \infty$ we have

$$(x_i^T w + b)_+ - (x_j^T w + b)_+ = x_i^T w + b - x_j^T w - b = (x_i - x_j)^T w$$

for some $i, j \in [n]$. Then, we rewrite the above problem as

$$\mathcal{Z}^{(b)} = \max(\mathcal{Z}_1^{(b)}, \mathcal{Z}_2^{(b)}),$$

where $\mathcal{Z}_1^{(b)}$ are the maximum value over the break points $b = -x_i^T w$ for each $i \in [n]$ and $\mathcal{Z}_2^{(b)}$ is the maximum value when $\beta \to \infty$ as given below.

$$\mathcal{Z}_1^{(b)} := \max_{j\in[n],\ell\in[n]} \max_{\|w\|_1=1} \left| \sum_{i=1}^{n} \left( ((x_i - x_\ell)^T w)_+ - ((x_j - x_\ell)^T w)_+ \right)_+ v_i \right| \tag{127}$$

and

$$\mathcal{Z}_2^{(b)} = \max_{j\in[n]} \max_{\|w\|_1=1} \left| \sum_{i=1}^{n} \left( (x_i - x_j)^T w \right)_+ v_i \right|,$$

Finally we note that by re-labeling the data points as $x_i \leftarrow x_i - x_\ell$ for some $\ell \in [n]$, the problem $\mathcal{Z}_1^{(b)}$ is equivalent to the maximization problem $\mathcal{Z}$ analyzed in the proof of Theorem 12. The rest of the proof closely follows the proof of Theorem 12 and we obtain the claimed result. $\qquad\square$

