# OpenReview forum: "From Complexity to Clarity: Analytical Expressions of Deep Neural Network Weights via Clifford Algebra and Convexity"
_TMLR — Accepted by TMLR_

### Review · Reviewer_Zaq9 · 2024-06-19

**Summary Of Contributions:**

This paper presents an analytical characterization of optimal weights of neural networks using mathematical tools in geometric algebra and optimization.
Analyzing ReLU networks with arbitrary depth and width, the paper shows that the optimal weights of a neuron can be expressed by a wedge product of a subset of training samples.
This in turn provides an interpretation of neural networks - each neuron calculates the positive distance of the input sample to an affine hull generated by a subset of training points (when using $\ell_2$ regularization).

**Audience:**

Yes

**Broader Impact Concerns:**

None.

**Claims And Evidence:**

Yes

**Requested Changes:**

Below are some questions and suggestions for the authors to consider.

Major comments:

- Eq. (2) and (20):
The regularization term is the squared sum of the $p$-norm of the weights with some special partition, which seems not a common regularization term in machine learning (except for when $p=2$, which reduces to the $\ell_2$ regularization).
Could the authors provide some insights on this?

- The results in the paper show an interesting resemblance to support vector machines, where the optimal solution is defined by a subset of training samples.
In SVM, the support vectors, a subset of training samples, define the decision boundary.
In this paper, the optimal weights and breaklines of each neuron are also defined by a subset of training samples.
Could the authors provide some insights on the connection between the two?

- Section 4.1, CIFAR experiment:
"After training, we implement the proposed polishing process on the first layer, and re-train the second-layer weights via convex optimization."
How is the retraining via convex optimization performed? Does the first/second layer refer to the first/second fully-connected layer?
In addition, I'm wondering why retraining is necessary. How's the network's performance without retraining?

- How would the analytical results generalize to convolutional layers? Convolutional layer weights are shared across spatial locations, and are sparse if written as a matrix. It's hard to imagine how the optimal weights can be written as a wedge product of training samples in this case.

- In the numerical experiments, I'm curious about the distribution of inner product magnitude between training points and trained weights.
The theory says that the optimal weight is orthogonal to a subset of training samples.
In the experiments, are the values of the inner product magnitude much smaller for the first $r-1$ samples than the rest?

- I'm curious about the distribution of the subsets of training points defining the optimal weights of each neuron.
For a single neuron, are the defining samples all from the same class?
Do all training points appear equally frequently as defining samples?
Could the author provide some insights on this?


Other minor comments:

- Theorems 6 and 7:
Are some assumptions missing? Is there any assumption on $m$, $d$, or $c$?

- Below Eq. (17):
The definition of $\kappa$ and $\kappa_b$ is a bit confusing, because in the text, the subscript ends at $d-1$, but in the definition, it ends at $d$.

- Sec 2.3: Why is $w \in \R^d$? If $\ell$ is a loss function and we have $\ell(w, y)$, should it be the same dim as $y \in \R^n$?

- In Eq (13): It seems that $Z \in \R^{\tilde{n} \times \tilde{n}}$ should be $z \in \R^{\tilde{n} \choose 2}$.

- It might be helpful to explain in Sec 2.2 how $\ell1$ norm is defined for a multivector.

- Theorem 6: "The weights of an optimal solution of (22)" - The equation number refers to a future equation. Should be 20?

- Section 3.5:
For the two-layer network, the optimal neurons in the first layer are shown. For the three-layer network, which layer's optimal neurons is being referred to?

- Fig 4(a):
Is the learning rate on x-axis for the pretraining stage or the re-training stage?

- Repitition:
"In our notation, lower-case letters are employed to represent vectors, while upper-case letters denote matrices. We use lower-case letters for vectors and upper-case letters for matrices."

**Strengths And Weaknesses:**

Strengths:
- The paper presents a clear and unified characterization of optimal networks, well-written and easy to follow.
- The derivation from low-dimensional to high-dimensional cases and from two-layer to deep networks is step-by-step and easy to follow.
- The interpretation of how neural networks extract representations of data is interesting, refreshing, and novel, based on insights from geometric Cliffor algebra and optimization theory.

Weaknesses:

The paper’s formulation of regularization term is a bit uncommon. Some experimental details are lacking. There are some errors or inconsistencies in notations.
See detailed comments below.

---

> ### Author Response · Authors · 2024-08-17
> **Author response**
>
> We would like to express our sincere gratitude for the thorough review and the constructive feedback provided. Your insights have been invaluable in refining our work. We are especially grateful for the acknowledgment of our novel application of geometric algebra within the context of neural networks.
>
> We have carefully revised the manuscript and uploaded a new version. All changes and updates are highlighted in blue for easy reference.
>
> Regarding the major comments:
>
> 1. Eq. (2) and (20): The regularization term is the squared sum of the
> -norm of the weights with some special partition, which seems not a common regularization term in machine learning (except for when , which reduces to the  regularization). Could the authors provide some insights on this?
>
> We thank the reviewer for this comment. We have added new figures to illustrate this point (please see Figures 11 and 12 in Section 4). We agree that p=2, which reduces to the standard weight decay regularization is more common in neural network. Our results apply for this case with a vanishing error as the number of samples grow large. However, our closed-form formulas apply exactly for p=1. Although this regularization is not commonly used, we would like to point out many of its interesting properties revealed by our analysis. In Figures 11 and 12, we show that the decision regions of optimal NNs regularized with p=1 and p=2 are nearly identical, especially for small values of lambda.
>
>
>
> 2. The results in the paper show an interesting resemblance to support vector machines, where the optimal solution is defined by a subset of training samples. In SVM, the support vectors, a subset of training samples, define the decision boundary. In this paper, the optimal weights and breaklines of each neuron are also defined by a subset of training samples. Could the authors provide some insights on the connection between the two?
>
> We thank the reviewer for this insightful question. In support vector machines (SVMs), the decision boundary is defined by a subset of training samples. This follows from convex duality, where the sparsity pattern of the SVM dual variables reveals this particular subset of training samples. This notion is closely paralleled in ReLU neural networks due to their dual convex formulation where each neuron is defined by a subset of training samples.  In our convex reformulations, the variable z plays the role of the SVM dual variables.  What is common between the two problems is that both dual formulations have polyhedral constraints. This is due to the piecewise-linear loss (hinge loss) in SVMs whereas in ReLU NNs, it’s due to the piecewise-linear activation and regularization. Furthermore, we also note that kernel SVMs also have a similar structure due to Reproducing Kernel Hilbert Space representation theorems. It may be possible to obtain a unified representer theorem for a wide class of models, but this is out of the scope of our current work. We added this discussion to Section 5 in the revision.
>
>
> 3. Section 4.1, CIFAR experiment: "After training, we implement the proposed polishing process on the first layer, and re-train the second-layer weights via convex optimization." How is the retraining via convex optimization performed? Does the first/second layer refer to the first/second fully-connected layer? In addition, I'm wondering why retraining is necessary. How's the network's performance without retraining?
>
> Thank you for this question. In the revision, we clarified and detailed this point. For example, consider a two-layer fully connected network. The polishing process applies the generalized cross product formula to the first layer neurons. Consequently, the second layer neurons need to be readjusted to maintain the accuracy, since the current values are sub-optimal according to the training loss. Therefore, we minimize the loss with respect to the second layer weights while keeping the first layer weights set via the cross product formula and unchanged (using requires_gradient False in Pytorch). This optimization problem is convex and may have a closed form solution (e.g., squared loss) and we take advantage of this. The first/second layer refers to first/second fully connected layers in a two-layer NN. For deeper NN experiments used in the paper, we apply this process to the last two fully connected layers.
>
> Regarding the necessity of this procedure, we have tried applying the polishing process without retraining the second layer weights and observed low accuracies (50-60% in CIFAR binary classification).

---

> ### Author Response · Authors · 2024-08-17
> **Author response cont'd**
>
> 4. How would the analytical results generalize to convolutional layers? Convolutional layer weights are shared across spatial locations, and are sparse if written as a matrix. It's hard to imagine how the optimal weights can be written as a wedge product of training samples in this case.
>
> Thank you for this question. We added a paragraph discussing CNNs and Transformers in Section 5. In short, a two layer standard convolutional network can be written as a fully connected NN with a reshaped data matrix containing vectorized patches extracted from the training images as its rows. This was shown in [R1]. Our results applies to this CNN and implies that the optimal neurons are given by wedge products of image patches.
> For deeper CNNs, the same result applies for short receptive fields using the BagNet architecture [R2]. We have added these discussions to the revised paper. For generic deep CNNs, it’s possible to redo our analysis using a similar strategy, however, it is beyond the scope of this work.
>
> [R1] T. Ergen, M. Pilanci  Implicit Convex Regularizers of CNN Architectures: Convex Optimization of Two- and Three-Layer Networks in Polynomial Time
> International Conference on Learning Representations, ICLR 2021
>
> [R2] Approximating cnns with bag-of-local-features models works surprisingly well on imagenet
> W Brendel, M Bethge
>
> 5. In the numerical experiments, I'm curious about the distribution of inner product magnitude between training points and trained weights. The theory says that the optimal weight is orthogonal to a subset of training samples. In the experiments, are the values of the inner product magnitude much smaller for the first samples than the rest?
>
> Thank you for this question. We have added a new numerical result (Section 4.1.3) and new plots (Figure 10) in the revised version to illustrate this point. The networks trained with conventional non-convex optimizers such as SGD and Adam only approximately minimize the training objective. However, we observe that for each neuron, many inner-products with training data are small (in [10^-2,  10^-3] when vectors are normalized to unit Euclidean norm). In Figure 10, we present the sorted inner-products for the Boston Housing dataset which confirms that the inner-products are numerically small for a subset of training points. After applying the polishing process, we force r-1 inner-products to be exactly zero and the accuracy of the model is maintained or improved.
>
> 6. I'm curious about the distribution of the subsets of training points defining the optimal weights of each neuron. For a single neuron, are the defining samples all from the same class? Do all training points appear equally frequently as defining samples? Could the author provide some insights on this?
>
> Thank you for this question. We have added a new numerical result to analyze this further. In Figure 11, we solve our convex programs to find the optimal L1 and L2 regularized NNs and their decision regions on the XOR dataset with four points. We also plot the optimal neuron breaklines, which pass through a special subset of training data vectors. It can be seen that these data vectors are of opposite class, and they define (a translated version of) the local decision boundary. Moreover, the optimal solutions with L1 regularization (p=1) and L2 regularization (p=2) are identical, leading to the same decision regions. In other datasets, we observed that the neuron-defining data points are from different classes. We believe that the examples near the decision boundary are more likely to be part of this subset. We consider this is an interesting and important question but we leave a detailed analysis of this structure to further work.
>
>
> Regarding the minor comments:
>
> 1. Theorems 6 and 7: Are some assumptions missing?
>
> Thank you for this question. There are no assumptions on d or c. There is an assumption that m (number of neurons) is not too small, in order to apply the characterization of Theorem 4. We have now made this assumption explicit in the beginning of Section 3.6.1 in the revision.
>
> 2. Below Eq. (17): The definition of \kappa is a bit confusing, because in the text, the subscript ends at d-1 , but in the definition, it ends at d.
>
> Thank you for pointing this out. There was a typo in the indices, which is fixed in the revision. The indices go from 1 to d-1 for \kappa and 1 to d for kappa_b. This is due to the addition of a bias term in kappa_b (b stands for bias). We are grateful for your careful review.
>
> 3. Sec 2.3: Why is w in R^d?
>
> Thank you for pointing this out. This was a typo: d needs to replaced with n, and there was a variable clash. We have fixed these in the revision. We are grateful for your careful review.
>
>
> 4. In Eq (13): It seems that z should be in R^(n choose 2).
>
>
> Thank you for pointing this out. You are right that z is of size (n \choose 2). We have fixed this in the revision. We are grateful for your careful review.

---

> ### Author Response · Authors · 2024-08-17
> **Author response cont'd**
>
> 5. It might be helpful to explain in Sec 2.2 how L1 norm is defined for a multivector.
>
> Thanks for pointing this out. We have added this in Section 2.2. We define the $\ell_1$ norm of a multivector as the sum of the absolute values of each component. In equation (15), this concides with the usual vector L1 norm since the multivector reduces to an ordinary vector.
>
>
> 6. Theorem 6: "The weights of an optimal solution of (22)" - The equation number refers to a future equation. Should be 20?
>
> Thanks for pointing this out. We have fixed this in the revision.
>
>
> 7. Section 3.5: For the two-layer network, the optimal neurons in the first layer are shown. For the three-layer network, which layer's optimal neurons is being referred to?
>
> Thank you for this question. We are referring to first layer neurons here. We have clarified this in the revision.
>
>
> 8. Fig 4(a): Is the learning rate on x-axis for the pretraining stage or the re-training stage?
>
> This learning rate is for the pretraining stage. We have clarified this in the revision.
>
>
> 9. Repetition: "In our notation, lower-case letters are employed to represent vectors, while upper-case letters denote matrices. We use lower-case letters for vectors and upper-case letters for matrices."
>
> Thanks for pointing this out. We have fixed this in the revision.

---

> > ### Comment · Reviewer_Zaq9 · 2024-08-27
> > **Reviewer comment**
> >
> > Thank the authors for their detailed responses, which addressed all my questions. I don't have further questions.

---

### Review · Reviewer_XQUN · 2024-07-02

**Summary Of Contributions:**

The paper proposes a novel approach to understanding deep neural networks using geometric algebra (specifically, Clifford algebra) and convex optimization. It demonstrates that optimal weights of deep ReLU neural networks can be derived through the wedge product of training samples. The study reframes the training of neural networks as a convex optimization problem over wedge product features, capturing the geometric structure of the dataset. This approach provides insights into the inner workings of deep networks, particularly in terms of hidden layers and neuron functionality.

**Audience:**

Yes

**Broader Impact Concerns:**

This paper presents a promising and novel approach to understanding neural networks through the lens of geometric algebra and convex optimization. With additional empirical validation and practical insights, it could significantly impact the field of neural network interpretability.

**Claims And Evidence:**

Yes

**Requested Changes:**

**Detailed comments**

1. The paper effectively sets up the problem and highlights the novelty of the approach. However, it could benefit from a more detailed discussion on the practical implications and potential applications of the proposed method.
2. The literature review is comprehensive, situating the paper within existing research on neural network interpretability, geometric analysis, and convex optimization. However, a more explicit comparison with other interpretability methods could be beneficial.
3. The introduction of wedge products and their application in neural networks is well-explained, but additional examples or visualizations could aid in understanding.
4. The reduction of neural network training to a convex optimization problem is innovative. However, the practical implications of this transformation, especially regarding computational efficiency, are not fully addressed.
5. The derivations and proofs are robust, but some sections, particularly involving higher-dimensional spaces, may be difficult for readers unfamiliar with the mathematical background.
6. The numerical examples provided are insightful, but the paper would benefit from more extensive experimentation on real-world datasets. This would help validate the theoretical findings and assess the method’s performance compared to traditional training methods.
7. The discussion around the geometric interpretation of neurons and the role of hidden layers is compelling and provides valuable insights.
8. The paper briefly mentions limitations but could elaborate on potential challenges and future research directions, particularly regarding scalability and integration with existing neural network frameworks.

I recommend acceptance of this paper if the authors can revise the manuscript with the following point.

1. Include more experiments on diverse datasets and architectures to substantiate the theoretical claims and assess practical performance.
2.  Discuss the computational efficiency and potential overhead of implementing the proposed convex optimization approach in practice.
3. Consider adding more supplementary materials or visual aids to help readers without a background in geometric algebra understand the concepts.

**Strengths And Weaknesses:**

**Strengths>:**
1. The use of geometric algebra to analyze neural networks is a novel and potentially transformative perspective, providing closed-form solutions for optimal weights.
2. The paper establishes a clear connection between neural networks, convex optimization, and geometric structures, offering new insights into the role of hidden layers and activation patterns.
3. The analysis is grounded in rigorous mathematical formulations, with detailed derivations and proofs provided for key theorems.
4. The study extends its analysis across various dimensions and neural network depths, making it broadly applicable.
5. The results offer a clear geometric interpretation of neural network operations, enhancing the understanding of neuron functionalities.

**Weaknesses:**

1. The use of Clifford algebra and the mathematical depth may be challenging for practitioners without a strong background in advanced mathematics.
2.  While theoretical insights are well-developed, the paper lacks extensive empirical validation on diverse datasets and architectures, which is crucial for demonstrating practical applicability.
3. The practical implications of implementing the proposed convex optimization approach on large-scale neural networks are not fully explored.
4. The assumptions and specific settings required for the theoretical results might limit the generalizability across different types of neural network architectures.

---

> ### Author Response · Authors · 2024-08-17
> **Author response**
>
> We would like to express our sincere gratitude for the thorough review and the constructive feedback provided. Your insights have been invaluable in refining our work. We are especially grateful for the acknowledgment of our novel application of geometric algebra within the context of neural networks.
>
> We have carefully revised the manuscript and uploaded a new version. All changes and updates are highlighted in blue for easy reference.
>
> Regarding the requested changes:
>
> 1. Include more experiments on diverse datasets and architectures to substantiate the theoretical claims and assess practical performance.
>
> Thank you for this suggestion. We have added two three new numerical results to Section 4.1.3 to further illustrate applications of our framework.
>                     (i)  In Figure 10, we consider the Boston housing dataset. We show that the polishing process yields similar performance gains within a tabular machine learning context as well. Similar to its effects on convolutional networks for CIFAR image classification and character-level autoregressive language modeling, the polishing process enhances both the training and test accuracy of the model, particularly when the number of neurons is increased.
>                     (ii)  In Figure 11, we compare L1 and L2 regularized optimal NNs and their optimal breaklines in the XOR dataset. It can be seen that the optimal breaklines pass through a subset of training points and both regularization strategies yield exactly the same decision region.
>                     (iii) In Figure 12, we compare L1 and L2 regularized optimal NNs for different values of the regularization parameter lambda. This figure illustrates that the predictions of the models are near identical for small lambda, but may deviate slightly for large lambda.
>
> 2. Discuss the computational efficiency and potential overhead of implementing the proposed convex optimization approach in practice.
>
> Thank you for this suggestion. We added two sections where we discuss computational complexity (page 19 and 22) of solving the convex programs exactly, and applying the polishing process. In summary, the complexity of the polishing process is very light. It’s not more expensive than running a standard optimizer such as SGD or Adam. On the other hand, the complexity of solving the convex programs exactly is polynomial in the number of samples but exponential in the rank of the training data. This is unavoidable due to the problem being NP hard. Nevertheless, we strongly believe that the Lasso characterization provides clarity to the inner-workings of ReLU neural networks.
>
> 3. Consider adding more supplementary materials or visual aids to help readers without a background in geometric algebra understand the concepts.
>
> Thank you for this suggestion. We have now added (i) a new section (Section 3.4 Illustrative Calculations in Geometric Algebra) containing two graphical illustrations of the affine hull distance formula and example applications of the geometric algebra formulas to calculate the optimal neurons, (ii) additional numerical illustrations in Figure 2 and Figure 3 to illustrate the triangular area via the wedge product formula.

---

### Review · Reviewer_tihS · 2024-08-07

**Summary Of Contributions:**

This paper presents a novel analysis of solutions to the neural network training problem using Clifford algebra and convex optimization. The paper provides closed form expressions for the optimal solutions of $\ell_1$ regularized neural networks and approximate solutions to $\ell_2$ regularized ReLU neural networks. This shows that the optimal neurons depend only on a subsets of the training data through their wedge products. The results hold for both shallow and deep networks of arbitrary dimension both with and without biases. Finally, the paper uses these insights to provide an algorithm for “polishing” the neurons of an approximately optimal neural network trained with SGD based on the analysis.

**Audience:**

Yes

**Broader Impact Concerns:**

This work is primarily theoretical and does not pose any ethical concerns.

**Claims And Evidence:**

Yes

**Requested Changes:**

I think the paper is good and I am willing to accept as is however, I do have a few suggestions which might strengthen it.

- It’s not clear how these results can be translated to the weights of neural network. For example in Theorem 2 the function is expressed as a linear combination of these generalized cross products. Not as ReLU ridge functions. (I think there is an explicit expression for the weights in the proof of Theorem 2 but it would be nice to state it in the main body).

- Some discussion on why the proof is limited to an approximate solution in the case of $\ell_2$ regularization.

- Fig. 3 shows the break points for the optimal neurons for different architectures. I think it would be interesting to see whether this is what is actually learned when training a neural network on a simple 2D problem.

- What is $\beta$ in the proof of Theorem 1? Should this be $b$?

**Strengths And Weaknesses:**

**Strengths**
- To the best of my knowledge the use of Clifford algebra in analyzing neural networks is novel and could pave the way for a new means of analyzing neural networks.
- The paper provides good evidence for the utility of this insight on real neural networks.
- The results in the paper hold for both shallow and deep neural networks with different types of regularization.

**Weaknesses**
- Optimal solutions are only obtained for the $\ell_1$ regularized case. This is not is not really standard in training neural networks. It would be nice if the paper could provides some insights on the difficulties of proving a similar result in the case $\ell_2$ regularization.
- While the optimal neurons are shown to be k-plane products of subsets of the training data it’s still not clear which subset is utilized.

---

> ### Author Response · Authors · 2024-08-17
> **Author response**
>
> We would like to express our sincere gratitude for the thorough review and the constructive feedback provided. Your insights have been invaluable in refining our work. We are especially grateful for the acknowledgment of our novel application of geometric algebra within the context of neural networks.
>
> We have carefully revised the manuscript and uploaded a new version. All changes and updates are highlighted in blue for easy reference.
>
> Regarding the requested changes:
>
> 1. It’s not clear how these results can be translated to the weights of neural network.
>
> Thank you for your insightful feedback. You are correct that the connection between the results and the neural network weights wasn't clearly articulated. In Theorem 2, the function is indeed expressed as a linear combination of generalized cross products, rather than directly as ReLU ridge functions. We have now added a remark following Theorem 2 to clarify this point. Specifically, the function $(x \wedge x')_+$ and positive part of the volume of the triangle formed by 0, x, x' are indeed ReLU ridge functions. We believe this connection between ReLU ridge functions and volume forms is novel in the literature.
>
> 2. Some discussion on why the proof is limited to an approximate solution in the case of
>  regularization.
>
> Thank you for this comment. We have added a remark after Theorem 2 to explain this difference that the l2 regularized NNs are near optimal while l1 regularized NNs are exactly optimal. This discrepancy is due to the polyhedral nature of the dual problem with $\ell_1$ regularization. In addition, we have added new numerical simulations in Section 4 to compare l1 and l2 regularized optimal NNs. We show that the decision regions of optimal NNs with $p=1$ and $p=2$ are near identical for small values of $\lambda$.
>
> 3. Fig. 3 shows the break points for the optimal neurons for different architectures. I think it would be interesting to see whether this is what is actually learned when training a neural network on a simple 2D problem.
>
> Thank you for this comment. We have added two new simulation results to confirm this result. In Figure 11, we illustrate the breaklines and the decision regions of the optimal L1 and L2 regularized NNs. As predicted by the theory, the breaklines pass through a subset of data points. In addition, in Figure 10 (b), we plot the sorted magnitudes of inner-products between the data vectors and neurons found via Adam. This plot confirm that the inner-products are in fact small (10^-2 to 10^-3 when vectors are normalized to unit norm) for a subset of training samples.
>
> 4. What is beta in the proof of Theorem 1? Should this be  b?
>
> Thank you for catching this typo. You are absolutely right that it should be $b$ instead of $\beta$. We've made the correction in the revised version. Your careful review is much appreciated!
>
>
>
> Regarding the weaknesses:
>
> It is correct that the results are exact for L1 regularization and near-exact for L2 regularization. We have added new figures to illustrate this point (please see Figures 11 and 12 in Section 4). We agree that p=2, which reduces to the standard weight decay regularization is more common in neural network. Our results apply for this case with a vanishing error as the number of samples grow large. However, our closed-form formulas apply exactly for p=1. Although this regularization is not commonly used, we would like to point out many of its interesting properties revealed by our analysis. In Figures 11 and 12, we show that the decision regions of optimal NNs regularized with p=1 and p=2 are nearly identical, especially for small values of lambda.
>
> You are correct that we do not know which specific subset of samples are required in the wedge product calculation before solving the problem. However, this subset can be found (i) exactly by enumerating all features and solving the Lasso formulation (ii) inexactly by using an initial guess for the NN (e.g., SGD/Adam trained) and applying the polishing process.

---

### Comment · Action_Editor_tVWL · 2024-07-11
**delay**

Dear Authors,

I apologize for the extended review period. One of the reviewers encountered unexpected issues and could not complete the review on time. Please anticipate an additional delay of several weeks. I appreciate your understanding and apologize for any inconvenience caused.

Best wishes,

Your AE

---

### Decision · Action_Editor_tVWL · 2024-09-23

**Recommendation:** Accept as is

**Comment:**

The paper proposes a novel approach to understanding deep neural networks by leveraging Clifford algebra and convex optimization techniques. All reviewers highlighted the significance and originality of the proposed methods. The use of Clifford algebra offers a new perspective that has not been explored before and can serve as a valuable tool for follow-up work. The main theorems are clearly presented and illustrated with good numerical examples. The results are derived for univariate, multivariate, shallow, and deep settings, providing a comprehensive analysis.

One reviewer noted that while some results, such as the univariate case, are not entirely new, and the optimal neurons for the multivariate case are derived for regularized neural networks, which are less common in practice, the overall contributions remain valuable. The polishing algorithm suggests potential practical advantages of this theory.

Based on the positive evaluations from the reviewers and the potential impact on the field, I recommend accepting the paper.

**Audience:**

The paper would be of interest to researchers in the foundational theory of deep learning and the broader learning theory community. By introducing novel ideas using geometric algebra and convex optimization, the work enriches the diversity within the community.

**Claims And Evidence:**

The reviewers agree that the claims made in this work are supported by accurate and clear evidence. The paper introduces a novel approach to analyzing neural networks using Clifford algebra, offering new perspectives not previously explored in the literature. The theorems are clearly explained and supported by numerical experiments, demonstrating the potential utility of this theory in practice. The overall contributions are significant and well-substantiated.